# Mucosal vaccination clears *Clostridioides difficile* colonization

Audrey K. Thomas[1,2], F. Christopher Peritore-Galve[1,2], Alyssa G. Ehni[1,2], Bruno B. C. Lança[1,2], Jonathan Coggin[1,2], Eric J. Brady[2,3], Sandra M. Yoder[2,3], Rebecca Shrem[2,4], Rubén Cano Rodríguez[2,4], Heather K. Kroh[1,2], Katherine N. Gibson-Corley[1], M. Kay Washington[1,2], Danyvid Olivares-Villagómez[1,2,5], C. Buddy Creech[2,3], Maribeth R. Nicholson[2,6], Benjamin W. Spiller[2,4] & D. Borden Lacy[1,2,7 ✉]

*Clostridioides difficile* infection (CDI) is the leading cause of healthcare- and antibiotic-associated infection and has a 30% recurrence rate[1–5]. Previous vaccine strategies against CDI failed to reduce pathogen burden, a prerequisite for preventing *C. difficile* transmission and recurrence[6–11]. These vaccines were administered parenterally, which induced a systemic immune response, rather than a mucosal response in the colon, the site of infection. Here we compare protection and colonization burden between mucosal (rectal) and parenteral (intraperitoneal) administration routes of a multivalent, adjuvanted vaccine combining inactivated *C. difficile* toxins and novel surface antigens. We found that mucosal immunization, but not parenteral, clears *C. difficile* from the host. Unique correlates of decolonization included faecal IgG responses to vegetative surface antigens and a colonic, T helper type 17 ($T_H17$)-skewed tissue-resident memory T cell response against spore antigen. Importantly, mucosal vaccination protected against morbidity, mortality, tissue damage and recurrence. Our results demarcate notable differences in correlates of protection and pathogen clearance between vaccine administration routes and highlight a mucosal immunization regimen that elicits sterilizing immunity against CDI.

*C. difficile* is a spore-forming anaerobic bacterium that is a leading cause of nosocomial infections and the primary cause of antibiotic-associated diarrhoea[1]. In the United States alone, *C. difficile* infection (CDI) leads to approximately 500,000 cases, 29,000 deaths and US$4.8 billion in healthcare costs each year[2–4]. As such, substantial efforts have been made to develop vaccines against CDI.

Previous vaccine strategies against CDI have targeted the primary virulence factors, toxins TcdA and TcdB[6–10]. A phase 3 clinical trial of a bivalent TcdA/TcdB vaccine from Pfizer protected against severe infection but did not meet its primary end point of preventing infection (ClinicalTrials.gov: NCT03090191), while one from Sanofi was discontinued after meeting criteria for futility (ClinicalTrials.gov: NCT01887812). Recently, a multivalent mRNA–lipid nanoparticle candidate vaccine against CDI showed promise in preclinical studies by protecting mice against severe infection and death[11]. However, these strategies and others demonstrated either minimal or no efficacy in clearing the bacterium from the colon[6–12]—a crucial end point when considering *C. difficile* spore transmission through the faecal–oral route. The 30% incidence of recurrent CDI[5] and the documented increase in community-acquired CDI cases among otherwise healthy adults[3] underscore the need for an immunization strategy that prioritizes *C. difficile* clearance.

We established a vaccination approach to potentiate *C. difficile* clearance while promoting protection against CDI symptoms. Our strategy combined (1) selection of novel vegetative and spore antigens to promote clearance of *C. difficile*; (2) inactivating point mutations of the *C. difficile* toxin antigens that retain native structure for broad epitope recognition; (3) the double mutant of *Escherichia coli* heat labile toxin (dmLT) as a mucosal adjuvant; (4) a rectal route of administration that was compared against parenteral vaccination; and (5) the assessment of humoral and cellular indicators of immune responses to identify correlates of symptom reduction and clearance. We demonstrate a protective mucosal vaccine formulation that provides sterilizing immunity to clear *C. difficile* from the host.

## NTAs induce non-canonical clearance

Sixteen candidate *C. difficile* non-toxin antigens (NTAs) associated with cell-surface functions were predicted to have low allergenicity, high antigenicity, B cell linear epitopes and MHC-II-binding sites, robust conservation across *C. difficile* strains, and low homology to host (mouse and human) and commensal-microbial proteins in silico[13]. We recombinantly expressed and purified 13 of these proteins and selected several for antigenicity testing based on the overall yield

[1]Department of Pathology, Microbiology & Immunology, Vanderbilt University Medical Center, Nashville, TN, USA. [2]Vanderbilt Institute for Infection, Immunology, and Inflammation, Vanderbilt University Medical Center, Nashville, TN, USA. [3]Vanderbilt Vaccine Research Program, Department of Pediatrics, Vanderbilt University Medical Center, Nashville, TN, USA. [4]Department of Pharmacology, Vanderbilt University, Nashville, TN, USA. [5]Department of Medicine, Division of Infectious Diseases, Vanderbilt University Medical Center, Nashville, TN, USA. [6]Department of Pediatrics, Division of Pediatric Gastroenterology, Hepatology, and Nutrition, Vanderbilt University Medical Center, Nashville, TN, USA. [7]Veterans Affairs Tennessee Valley Healthcare System, Nashville, TN, USA. ✉e-mail: borden.lacy@vanderbilt.edu

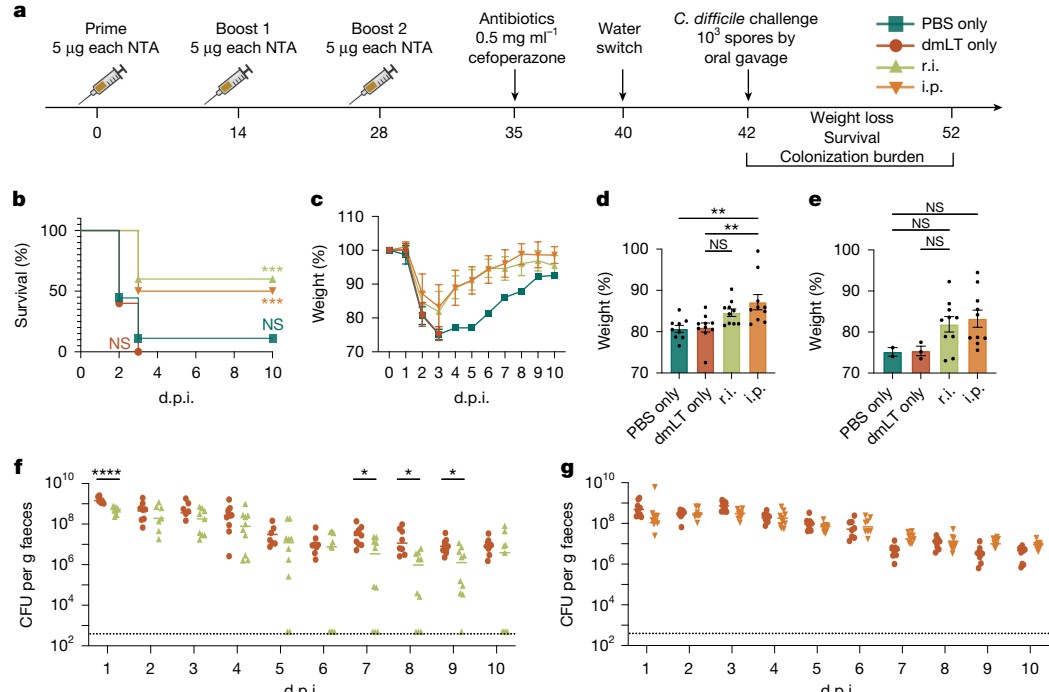

**Fig. 1 | Rectal instillation of the NTA cocktail promotes survival and reduces *C. difficile* colonization burden. a**, Experimental schematic. The diagram was created using BioRender. **b**–**e**, The survival (**b**) and percentage weight loss relative to day 0 (**c**), and quantification of weight at 2 days post-infection (d.p.i.) (**d**) and 3 d.p.i. (**e**). **f**,**g**, Enumeration of *C. difficile* bacteria in the faeces through CFU titrations after r.i. (**f**) and i.p. (**g**) vaccination. The limit of detection is indicated by the dotted lines: 500 CFU per g faeces. For **c**–**e**, data are mean ± s.e.m. Statistical significance was calculated using log-rank Mantel–Cox tests (**b**), one-way ANOVA with Tukey's correction (**d** and **e**) and two-sided Student's *t* tests (**f** and **g**), with each day analysed separately; *$P \le 0.05$, **$P \le 0.01$, ***$P \le 0.001$, ****$P \le 0.0001$; NS, not significant. For **b**–**g**, $n = 10$ per group; decreases in $n$ were due to animal mortality during infection. Individual datapoints are represented and were pooled from two independent experiments.

and solubility of each protein or protein complex (Supplementary Table 1). We prioritized FlgGEK, a ternary complex comprising the FlgG, FlgE and FlgK flagellar basal-body rod components (Supplementary Information); C40 peptidase 2, a cell-wall-modifying enzyme; polysaccharide deacetylase, a peptidoglycan deacetylase; and CspC, a spore-coat-bound germinant receptor[14].

Vaccination-induced bacterial clearance from the colon is mediated by mesenteric lymph node responses. We therefore reasoned that immunization with a mixture of the NTAs at the site of infection would induce a mucosal immune response capable of reducing the bacterial burden. We vaccinated mice 3 times over 28 days either by rectal instillation (r.i.; a proxy for mucosal immunization by enema) or intraperitoneal (i.p.) injection (a proxy for parenteral immunization) (Fig. 1a). All immunizations included dmLT as an adjuvant, which elicits systemic and mucosal humoral and $T_H 17$ responses in the gut[15]. Both r.i. and i.p. vaccination of the NTA mixture promoted survival against lethal challenge with wild-type (WT) *C. difficile* R20291 (Fig. 1b). However, only i.p.-vaccinated mice were significantly protected against weight loss at 2 days after infection (Fig. 1c–e). Separate cohorts of vaccinated mice were challenged with *C. difficile* R20291 ΔAΔB (hereafter ΔAΔB), a disrupted toxin strain, to enumerate colonization burden without animal mortality. r.i. of the NTA mixture significantly reduced ΔAΔB colonization burden compared with in the dmLT-only controls (Fig. 1f). Within the r.i.-vaccinated group, there were individual mice that decreased colonization to the limit of detection (500 colony-forming units (CFU) per g faeces) beginning at day 5 after infection and remained uncolonized until the experimental end point. By contrast, i.p. vaccination had no effect on colonization burden (Fig. 1g). This demonstrates that mucosal vaccination of NTAs reduces colonization burden while providing modest protection against severe disease and death.

To determine the individual contributions of each NTA to the protective and colonization-reduction effects from r.i. vaccination with the NTA mixture, we immunized mice through either r.i. or i.p. with a single NTA before challenging with either WT R20291 or ΔAΔB (Extended Data Fig. 1a). Each NTA administered by r.i., as well as C40 peptidase 2 administered by i.p., provided a survival benefit (Extended Data Fig. 1b,c). Importantly, r.i. vaccination of CspC, C40 peptidase 2 and FlgGEK significantly reduced the colonization burden in mice challenged with ΔAΔB compared with the dmLT-only controls (Extended Data Fig. 1d). i.p. vaccination with the individual NTAs did not affect colonization (Extended Data Fig. 1e).

Despite the ability of the vaccine to clear bacteria, r.i.-vaccinated mice did not exhibit increased systemic anti-NTA IgG or faecal IgA, discordant with typical dmLT-adjuvanted mucosal humoral responses[15]. Indeed, i.p.-vaccinated mice had significantly greater anti-NTA serum IgG (Extended Data Fig. 1f) and faecal IgA (Extended Data Fig. 1g) titres compared with the r.i.-vaccinated mice. By contrast, faecal IgG titres against FlgGEK, C40 peptidase 2 and polysaccharide deacetylase increased in r.i.-vaccinated mice (Extended Data Fig. 1h). While there were no statistically significant increases in anti-NTA faecal IgA or serum IgG in r.i.-vaccinated mice compared with i.p.-vaccinated mice, these titres did correlate positively with the ability of single-NTA r.i.-vaccinated mice to reduce *C. difficile* burden, but did not for i.p.-vaccinated mice (Extended Data Fig. 1i,j).

## r.i. vaccination provides sterilizing immunity

Previous *C. difficile* vaccines included either chemically crosslinked or formalin-inactivated toxoid antigens of *C. difficile* toxins TcdA and TcdB[6–10]. Although these methods allow for safe injection, they potentially disrupt pertinent neutralization epitopes on the toxins[16,17].

To avoid this, we minimally mutated TcdA and TcdB constructs to prevent glucosyltransferase activity of both toxins (GTX)[18]. To address residual TcdB toxicity, we included additional mutations to abrogate pore formation (L1106K)[19] and CSPG4 receptor binding (D1812G)[20]. These mutant toxins were verified as natively folded, inactive in a cell rounding assay and safe for immunization in vivo when coupled with dmLT (Supplementary Information).

As a third immunization by r.i. and i.p. did not increase mucosal humoral titres (Extended Data Fig. 1f–h), we immunized mice with a combination of NTAs and inactivated *C. difficile* toxins using a two-dose, prime–boost schedule (Fig. 2a). The amount of dmLT for r.i. and i.p. vaccinations was also optimized to maximize humoral longevity, reducing dmLT to 15 and 1 μg per dose, respectively (Extended Data Fig. 2).

Both r.i. and i.p. vaccination of the toxin and NTA formula promoted survival during challenge with WT *C. difficile* R20291 (Fig. 2b,c). Vaccination of the toxins alone or in combination with the NTAs by both administration routes protected against weight loss at critical disease timepoints (Fig. 2d,e). Importantly, r.i. vaccination protected against colonic and caecal epithelial injury (Extended Data Fig. 3a,b) on day 3 after infection in comparison to naive unvaccinated mice.

To determine whether the addition of the toxins to a NTA formula reduces colonization in r.i.- and i.p.-vaccinated mice, total CFUs (reflecting both spores and vegetative bacteria) or heat-resistant spores alone were enumerated in the faeces for 10 or 15 days after infection. Vaccination with the toxin and NTA formula by r.i. allowed mice to clear *C. difficile* by day 9 after infection (Fig. 2f–h) and spores by day 8 (Fig. 2i–k). No spores or vegetative bacteria were present in macerated colons and caeca from these animals (Extended Data Fig. 4a–d), nor did they contain *C. difficile* as determined by 16S PCR (Extended Data Fig. 4e) and quantitative PCR (qPCR; Extended Data Fig. 4f,g). These results contrasted with i.p.-vaccinated mice, which did not clear *C. difficile* (Fig. 2l–q and Extended Data Fig. 4a–g) and remained colonized at levels similar to naive infection (Extended Data Fig. 4h). These data demonstrate that r.i., but not i.p., delivery of the inactive toxin and NTA formula eliminates *C. difficile* at both the faecal and tissue level while protecting against weight loss, epithelial damage and death.

The ability of the inactive toxin + NTA formula to eliminate CDI in r.i.-vaccinated mice led us to next examine whether this administration route and formula could prevent recurrent infection in a vancomycin-induced relapse model (Fig. 3a). Mice r.i.-administered with the toxin + NTA formula were significantly protected against death during CDI relapse compared with the dmLT-only and toxin + dmLT formulas (Fig. 3b). Toxin + NTA-vaccinated mice were also significantly protected against weight loss and diarrhoea during primary and recurrent infection (Fig. 3c,d). These mice also cleared *C. difficile* during vancomycin treatment and did not relapse for 30 days after infection (Fig. 3i,j), in contrast to the control groups (Fig. 3e–h). Thus, r.i.-vaccination-induced clearance of *C. difficile* from the host protects against relapse by providing sterilizing immunity.

## Faecal IgG clears vegetative *C. difficile*

Mucosal and systemic antibody responses to the vaccines were assessed in the sera and faeces before *C. difficile* challenge to examine the humoral mechanisms of bacterial clearance. While both administration routes induced anti-TcdA serum IgG, systemic humoral responses for all other antigens were elicited only after i.p. injection (Fig. 4a–e). r.i. vaccination did not induce faecal IgA responses against the antigens, apart from TcdA (Fig. 4f–j). However, r.i. vaccination did significantly increase anti-TcdA, anti-C40 peptidase 2 and anti-FlgGEK faecal IgG (Fig. 4k,n,o).

Vaccinated mice were euthanized at 3 days after infection with *C. difficile* and assessed for cellular correlates of humoral immunity.

Notably, r.i. of the toxin + NTA formula increased anti-C40 peptidase 2 and anti-FlgGEK memory B cells (Extended Data Fig. 5b) and plasma cells (Extended Data Fig. 5c) in the gut-draining mesenteric lymph nodes. Thus, r.i. vaccination increases mucosal IgG responses and memory against the vegetative antigens C40 peptidase 2 and FlgGEK.

To further gauge the role of anti-vegetative mucosal IgG in *C. difficile* clearance, as well as determine whether these IgG can transudate from serum to the site of infection, we performed a passive transfer experiment. Donor mice were immunized through r.i. with either dmLT alone or C40 peptidase 2, FlgGEK and dmLT to elicit anti-vegetative antigen mucosal IgG. Faeces was collected 2 weeks after boost, and faecal IgG was isolated and filter-sterilized. Donor faecal IgG was administered through either i.p. or r.i. three times during *C. difficile* challenge to recipient mice that were previously vaccinated against the inactivated toxins and CspC (Extended Data Fig. 6a).

There were no differences in survival or weight loss between the groups administered control or anti-vegetative antigen faecal IgG from either route (Extended Data Fig. 6c,d). However, mice that had received anti-vegetative antigen faecal IgG through r.i. had significantly less severe diarrhoea than those that had received the control (Extended Data Fig. 6e). These mice also had significantly decreased total *C. difficile* burden compared with their respective r.i. dmLT-only controls (Extended Data Fig. 6f,g). Total CFU burden was unaffected in mice that were i.p. injected with either group's mucosal IgG (Extended Data Fig. 6j,k). None of the groups showed statistical differences in *C. difficile* spore counts, suggesting that the variances in total CFU burden were due to effects on vegetative *C. difficile* (Extended Data Fig. 6h,i,l,m). A functional in vitro analysis revealed that anti-C40 peptidase 2 and anti-FlgGEK mucosal IgG disrupted *C. difficile* swimming motility (Extended Data Fig. 6b). Taken together, these data illustrate that anti-C40 peptidase 2 and -FlgGEK faecal IgG reduce vegetative *C. difficile* burden and impede bacterial movement. These results further demonstrate that anti-vegetative faecal IgG administered through the i.p. route do not reduce colonization burden, suggesting that any contribution of systemic circulating IgG to *C. difficile* clearance is minimal.

## r.i. elicits anti-spore T$_H$17 responses

Although cellular mechanisms of protection against *C. difficile* have not been studied in the context of rectal immunization, other mucosal vaccines induce tissue-resident memory (T$_{RM}$; CD103$^+$CD69$^+$) cells at the site of infection[21,22]. As such, we hypothesized that the immune memory generated by r.i. would also enable T$_{RM}$ cell responses in intraepithelial (IEL) and lamina propria (LPL) lymphocytes.

There were significant increases in CD8$^+$ T$_{RM}$ cells in the IEL (Fig. 4p) and LPL (Fig. 4q) compartments of mice that were r.i. administered the toxin-only or toxin + NTA formulas. Moreover, r.i. administration significantly induced CD4$^+$ T$_{RM}$ cells in both the IEL and LPL compartments (Fig. 4r,s). To define the molecular T$_{RM}$ cell response to specific antigens, we co-cultured colonic CD4$^+$ or CD8$^+$ T$_{RM}$ cells isolated from i.p.- and r.i.-immunized mice with bone-marrow-derived dendritic cells (BMDCs) from naive mice that were primed with a single vaccine antigen. Cytokine profiles from co-cultures were analysed after 3 days. CspC induced a significant increase in T$_H$17-cell-response-associated cytokines, including IL-1β, IL-6 and IL-1α, in T cells from r.i.-vaccinated mice (Fig. 4t–v). These cytokine responses were not produced by CD4$^+$ and CD8$^+$ T cells co-cultured with BMDCs that were primed with the other vaccine antigens (Extended Data Fig. 7a–c). No other notable cytokine responses against the other antigens were observed, including those canonical in the T$_H$1 and T$_H$2 pathways (Extended Data Fig. 7).

To determine whether the inclusion of CspC in the vaccine formula is necessary to clear *C. difficile*, mice were r.i.-vaccinated with either a

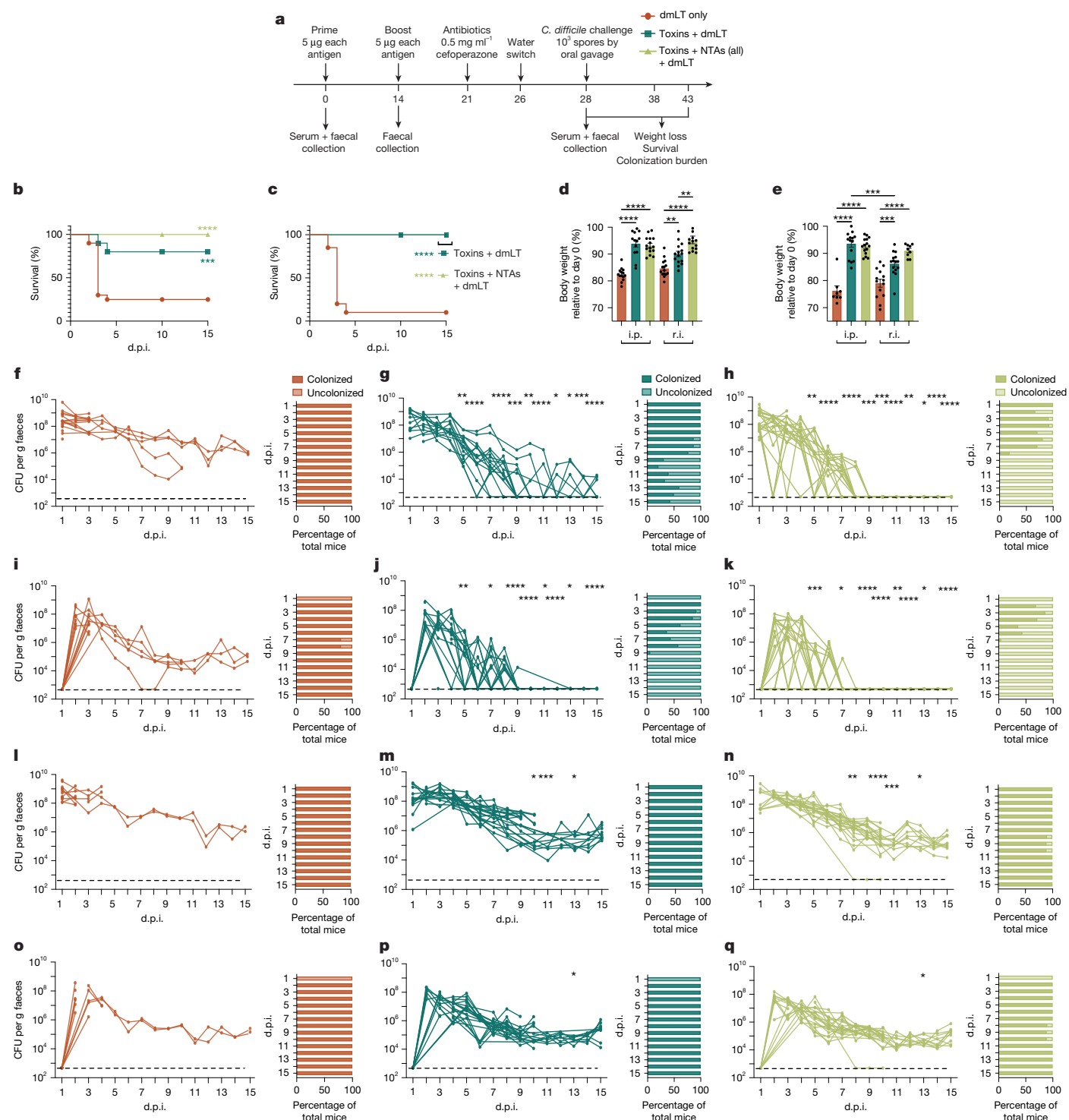

**Fig. 2 | r.i. of the toxin + NTA formula clears colonization while preventing weight loss, epithelial injury and mortality. a**, Experimental schematic. Mice were infected for 10 or 15 days, with no differences observed between cohorts. The diagram was created using BioRender. **b,c**, The survival of infected with WT *C. difficile* R20291 after r.i. (**b**) and i.p. (**c**) vaccination. The bracket refers to the lines of both the toxin + dmLT and toxin + NTA + dmLT cohorts. **d,e**, Weight loss at 2 d.p.i. (**d**) and 3 d.p.i. (**e**) relative to day 0. Data are mean ± s.e.m. **f–q**, Enumeration of total *C. difficile* R20291 bacteria (vegetative cells and spores (**f–h** and **l–n**)) and spores (**i–k** and **o–q**) in faeces after vaccination with dmLT only (**f,i,l,o**),

toxins + dmLT (**g,j,m,p**) and toxins + NTAs + dmLT (**h,k,n,q**) through the r.i. (**f–k**) and i.p. (**l–q**) routes. *P* values were calculated relative to the control at the same timepoint. Individual lines correspond to individual mice. The limit of detection is shown by the dotted line: 500 CFU per g faeces. Statistical significance was calculated using log-rank Mantel–Cox tests (**b** and **c**) and one-way ANOVA with Tukey's correction (**d–g** and **f–q** (each day analysed individually)). For **b–q**, *n* = 20 per group; decreases in *n* are due to animal mortality during infection. Individual datapoints are represented and were pooled from two independent experiments.

formula of inactivated toxins, CspC and dmLT, or a formula of inactivated toxins, C40 peptidase 2, FlgGEK and dmLT. While vaccination with the toxins and CspC significantly protected against death and

weight loss (Extended Data Fig. 8a–c), the formula without resulted in a 40% mortality rate and significant weight loss. The CFU burden in mice vaccinated without CspC was significantly increased compared with

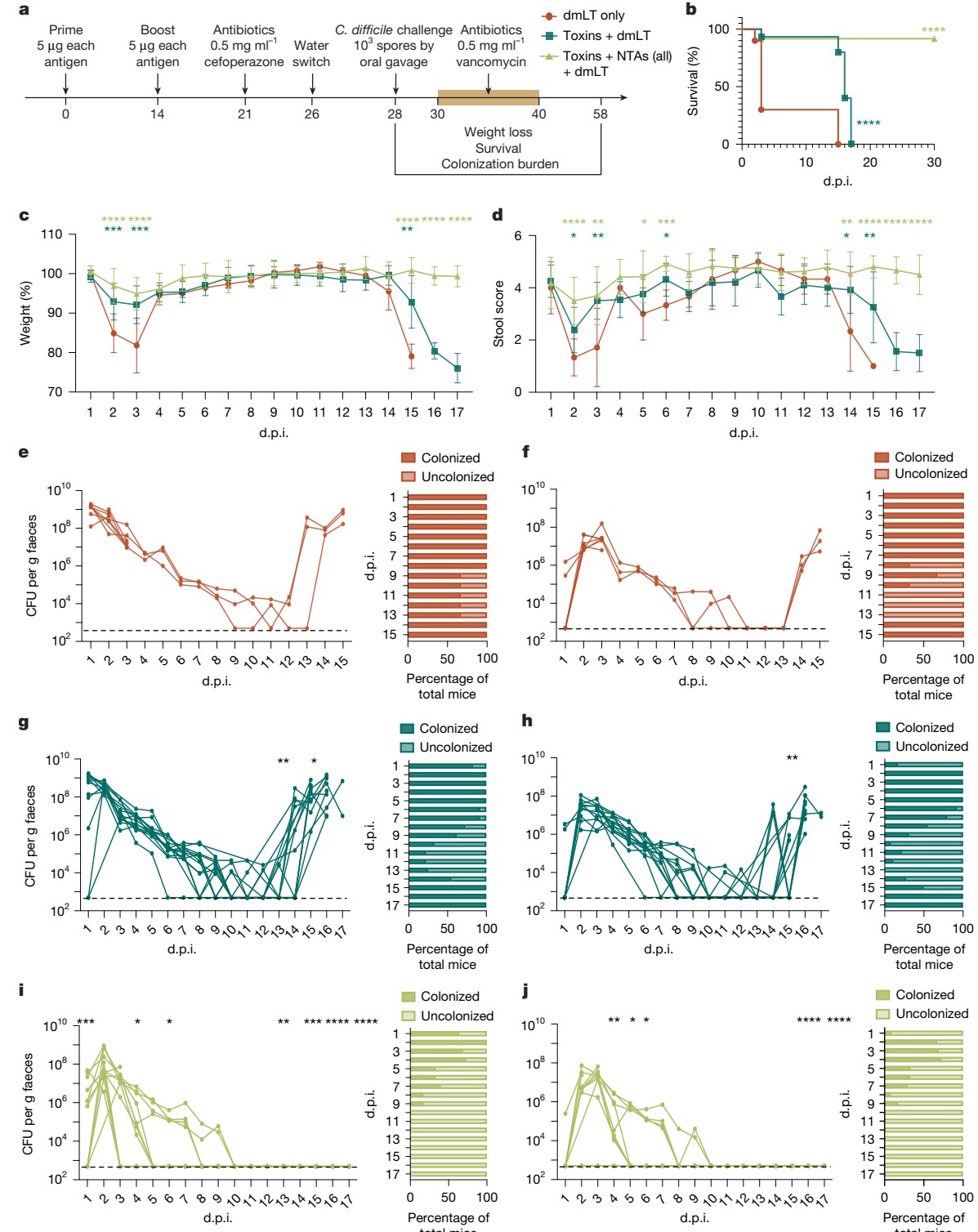

**Fig. 3 | r.i. of the toxin + NTA formula prevents relapsing CDI. a**, Experimental schematic. The diagram was created using BioRender. **b**, Survival of mice 30 days after infection and relapse with WT *C. difficile* R20291. **c**, Weight loss. **d**, Stool score. **e**–**j**, Enumeration of total *C. difficile* R20291 bacteria (vegetative cells and spores; **e,g,i**;) and spores in faeces (**f,h,j**) after vaccination with dmLT only (**e,f**), toxins + dmLT (**g,h**) and toxins + NTAs + dmLT (**i,j**). *P* values were calculated relative to the control at the same timepoint. Individual lines correspond to individual mice. The limit of detection is shown by the dotted line: 500 CFU per g faeces. For **c** and **d**, data are mean ± s.e.m. Statistical significance was calculated using log-rank Mantel–Cox tests (**b**) and one-way ANOVA with Tukey's correction (**c** and **d**, and **e**–**j**, with each day analysed individually). For **b**–**j**, *n* = 15 per group; decreases in *n* are due to animal mortality during infection.

in the dmLT-only controls (Extended Data Fig. 8e,g). Conversely, mice immunized with the toxin + CspC formula cleared *C. difficile* spores by day 10 after infection but did not clear total bacterial burden (Extended Data Fig. 8d,f).

## Mucosal vaccination has longevity

Although current parenteral *C. difficile* vaccines elicit long-term protective antitoxin immunity, the longevity of such responses in

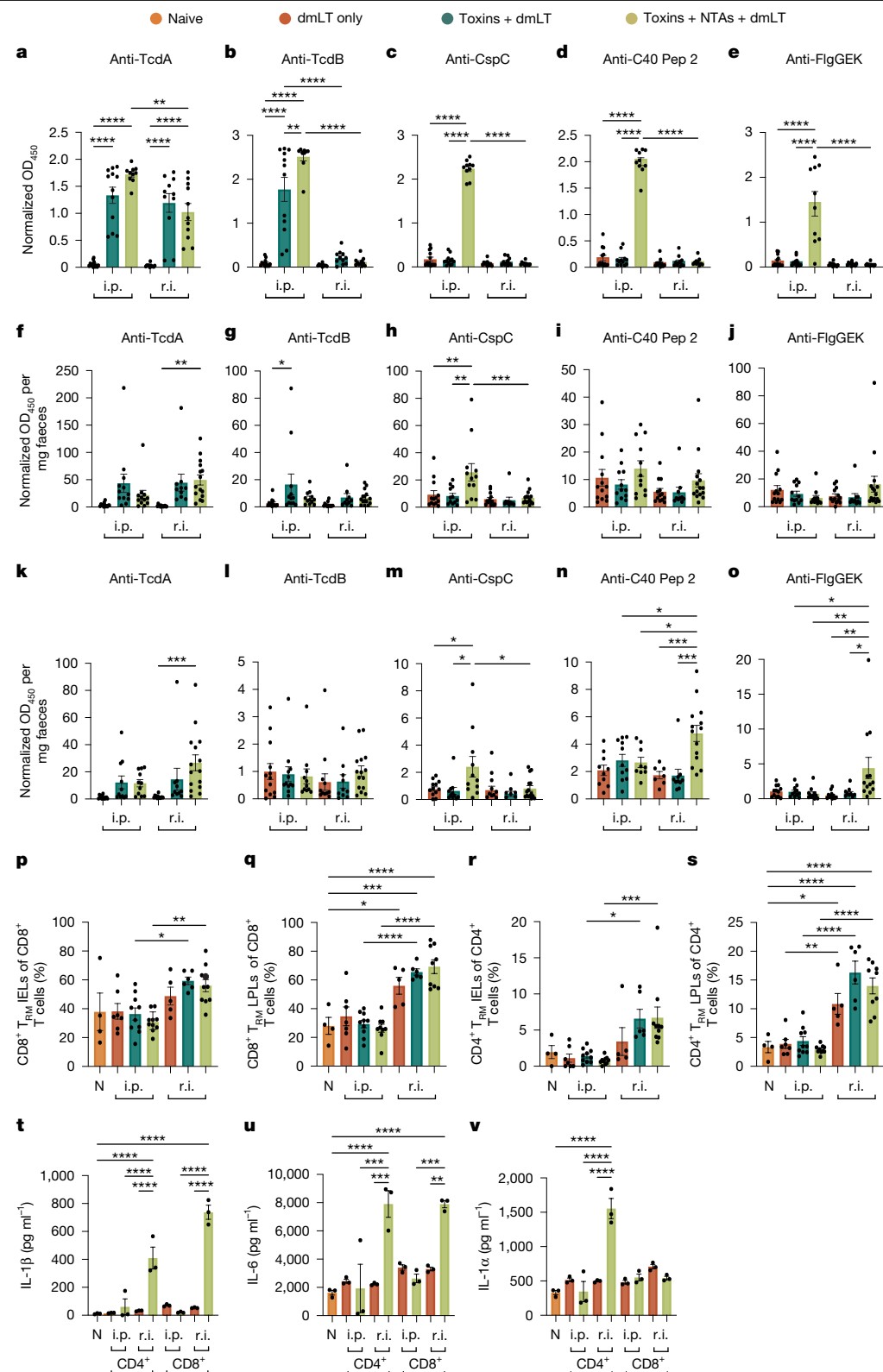

**Fig. 4 | r.i. of the toxin + NTA formula elicits robust anti-C40 peptidase 2 and anti-FlgGEK faecal IgG and a colonic T_H17-associated tissue-resident memory cell response against CspC. a–e**, Sera IgG responses against antigens before challenge at day 28. **f–j**, Faecal IgA responses against antigens before challenge at day 28. **k–o**, Faecal IgG responses against antigens before challenge at day 28. **p–s**, Percentages of T cells in the IEL and LPL compartments of the colon at day 3 after infection. **t–v**, Enumeration of cytokines produced by ex vivo T cells from

vaccinated mice that were co-cultured with CspC-primed dendritic cells in vitro. For **a–v**, data are mean ± s.e.m. Statistical significance was calculated using one-way ANOVA with Tukey's correction (**a–v**). Individual datapoints are represented and were pooled from two independent experiments. For **a–o**, $n = 5–15$ per group (dependent on ability to collect samples); **p–s**, $n = 4–10$ per group (dependent on whether there was death at day 3 after infection); **t–v**, $n = 3$ replicates per group/condition. C40 pep 2, C40 peptidase 2; N, naive/no antigen mice.

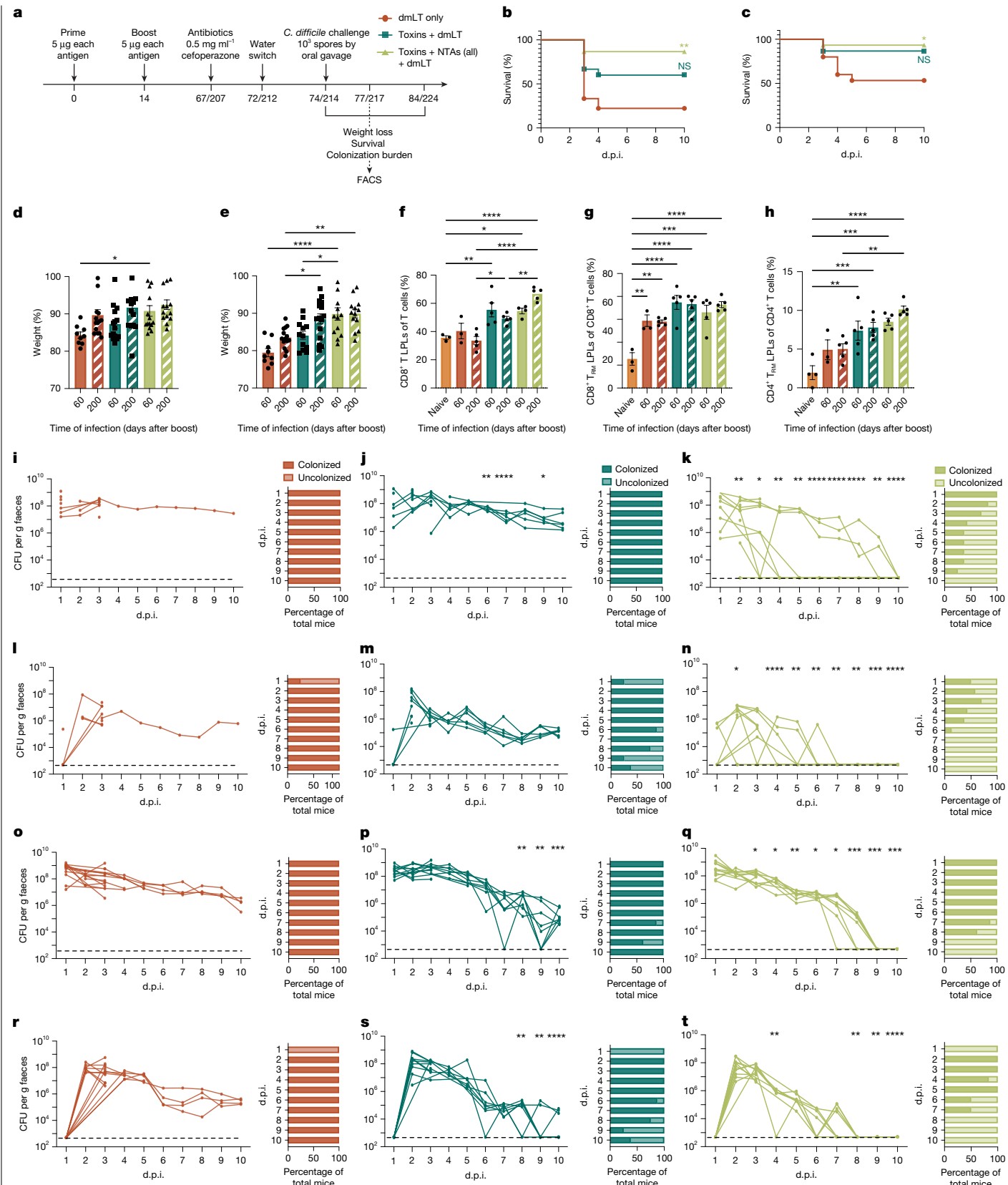

**Fig. 5** | See next page for caption.

a colonic mucosal vaccine model have not been investigated. To test whether r.i. administration would durably protect against disease and clear *C. difficile*, we vaccinated and challenged mice with WT *C. difficile* R20291 either 60 or 200 days after the final dose

(Fig. 5a). Immunization with the toxin and NTA formula protected against death and weight loss at both elongated infection timepoints (Fig. 5b–e). These immunized mice also cleared vegetative *C. difficile* and spores (Fig. 5k,n,q,t) at both timepoints compared

**Fig. 5 | r.i. vaccination imbues durable colonization and infection protection at 60 and 200 days after boost. a**, Experimental schematic. The diagram was created using BioRender. **b,c**, Survival of WT mice after infection with *C. difficile* R20291 at 60 (**b**) and 200 (**c**) days after boost. **d,e**, Weight loss at 2 d.p.i. (**d**) and 3 d.p.i. (**e**) after infection at 60 and 200 days after boost. **f–h**, Percentages of T cells (CD8$^+$ (**f**), CD8$^+$ T$_{RM}$ (**g**) and CD4$^+$ T$_{RM}$ (**h**) LPLs) in the LPL compartment of the colon. **i–t**, Enumeration of total *C. difficile* R20291 bacteria (vegetative cells and spores; **i–k** and **o–q**) and spores (**l–n** and **r–t**) in the faeces of mice 60 (**i–n**) and 200 (**o–t**) days after boosting, for the dmLT only (**i,l,o,r**), toxins + dmLT (**j,m,p,s**) and toxins + NTAs + dmLT (**k,n,q,t**) groups. For **i–t**, *P* values were calculated relative to the control at the same timepoint. Individual lines correspond to individual mice. The limit of detection is shown by the dotted line: 500 CFU per g faeces. For **d–h**, data are mean ± s.e.m. Statistical significance was calculated using log-rank Mantel–Cox tests (**b** and **c**) and one-way ANOVA with Tukey's correction (**d–t**; for **i–t**, each day was analysed individually). For **b–e**, *n* = 15 per group; **f–h**, *n* = 5 per group; **i–t**, *n* = 10 per group; decreases in *n* were due to animal mortality during infection. Individual datapoints are represented and were pooled from two independent experiments. FACS, fluorescence-activated cell sorting.

with the dmLT-only (Fig. 5i,l,o,r) and toxin + dmLT-only (Fig. 5j,m,p,s) formulas.

To address whether the humoral phenotypes were durable, we measured antigen-specific antibody titres in the sera and faeces at days 14, 74, 134 and 214 (Fig. 5a). There were significant increases in anti-TcdA and anti-TcdB serum IgG, faecal IgA and faecal IgG at 60 and 120 days after boost (day 74 and 134, respectively) (Extended Data Fig. 9a–c). Serum IgG increased against CspC and C40 peptidase 2 at 120 days after boost, and anti-FlgGEK titres trended similarly (Extended Data Fig. 9a). Faecal IgG increased against C40 peptidase 2 and FlgGEK (Extended Data Fig. 9c) at 74 days before waning at 120 days after boost. Thus, there is an expansion of anti-NTA and antitoxin humoral responses, in particular faecal IgG, months after r.i. boost.

We further compared cellular responses at 3 days after infection to determine whether the T$_{RM}$ cells were present at the 60- and 200-day post-r.i.-boost timepoints. Total T cell responses, but not T$_{RM}$ cells, increased in the IEL compartment of vaccinated mice at these elongated timepoints (Extended Data Fig. 10c,e–h). We also found significant increases in CD8$^+$ T cells among LPLs of vaccinated mice (Fig. 5f), as well as CD8$^+$ T$_{RM}$ (Fig. 5g) and CD4$^+$ T$_{RM}$ (Fig. 5h) cells. Taken together, these data highlight the persistence and longevity of r.i.-induced anti-NTA faecal IgG and T$_{RM}$ cells in the lamina propria—responses that correlate with sterilizing immunity against *C. difficile*.

## Discussion

Here we describe the preclinical development of a multivalent, mucosal vaccine comprising inactivated toxins, novel surface-associated NTAs and the dmLT adjuvant, which elicits sterilizing immunity against acute and recurrent CDI. Along with decreasing disease severity, rectal administration of this vaccine reduces tissue damage caused by the *C. difficile* toxins (Extended Data Fig. 3a,b)—a protection not garnered by other preclinical vaccines[11] and antitoxin therapeutics[23,24]. We expect this strategy to have strong translational value in the effort to develop a human vaccine for CDI, as well as other enteric pathogens. We envision deploying an effective r.i. vaccine as an enema, similar to the original administration route of faecal microbiota transplantation. A recent survey highlighted the willingness of the public to receive a *C. difficile* vaccine, regardless of administration route, provided that it is effective[25].

Given that CDI is primarily a disease of the elderly[26], we wanted to evaluate whether the age of immunization or challenge had an impact on the sterilizing immunity inculcated by r.i. We vaccinated mice at both 6- and 12-weeks of age and noted no difference in the responses to *C. difficile* challenge (combined data are represented in Fig. 2). We also examined the longevity of the immune response and challenged mice with *C. difficile* at 14, 60 and 200 days after the final r.i. boost. The observation that mice retain T$_{RM}$ cell responses and the ability to clear the pathogen when challenged 200 days after boost (Fig. 5g,h,q,t) provides a positive indication for clinical translation into older populations. A key next step will be to conduct r.i. immunization trials in aged mice (>18 months)[27].

Previous *C. difficile* toxin vaccines were parenterally administered[6–11]. Our results align with these studies in demonstrating that parenteral vaccination decreases morbidity and mortality (Fig. 2b-e), but not bacterial burden, even with the inclusion of *C. difficile* surface antigens (Fig. 2n,q). Canonical mucosal IgA responses were upregulated in these mice (Fig. 4f–j) but did not correlate with bacterial clearance (Fig. 2n,q). Studies focusing on why the mucosal IgA garnered by parenteral vaccination does not imbue sterilizing immunity may elucidate strategies to overcome these shortcomings in future iterations of preclinical *C. difficile* vaccines.

Our results highlight that r.i., but not i.p., vaccination elicits robust faecal IgG against the *C. difficile* NTAs C40 peptidase 2 and FlgGEK (Fig. 4n,o), which clear the vegetative bacterium (Extended Data Fig. 6f,g) and restrict motility (Extended Data Fig. 6b). Other enteric pathogens have been shown to be eliminated from the gut in a faecal-IgG-dependent manner[28,29]. Although faecal IgG tends to be less abundant than faecal IgA, the concentration does increase in response to enteric infection[30]. Intestinal IgG-secreting plasma cells can home to the bone marrow[31], further suggesting a mechanism for sustained immunological memory. Together, these data implicate a localized anti-surface antigen faecal IgG response, as elicited by r.i., as a key factor in constraining vegetative *C. difficile* colonization.

We included CspC as a vaccine antigen with the intention of targeting spores during infection. r.i. of formulas including CspC reduce the total colonization burden (Fig. 2h,k and Extended Data Fig. 8d–g) faster than the toxin + dmLT formula alone (Fig. 2g). However, the lack of anti-CspC antibody responses (Fig. 4c,h,m) suggests a cellular mechanism of spore clearance. Indeed, CspC-primed BMDCs elicited T$_H$17-cell-response-associated cytokines from CD4$^+$ and CD8$^+$ T$_{RM}$ cells during co-culture (Fig. 4t–v). Notably, IL-17a was lacking from our cytokine assay responses (Extended Data Fig. 7g). Previous reports have highlighted difficulties with low limits of IL-17a quantification in human sera samples using Luminex analysis, which may explain our results[32,33].

Alternatively, a reduction in or lack of IL-17a in a T$_H$17-cell-driven response has been linked to intracellular bacterial infections in the gut[34]. This, coupled with the significant increase in CD8$^+$ T$_{RM}$ in r.i.-vaccinated mice (Fig. 4p,q), was interesting, as *C. difficile* is an extracellular pathogen. Two recent studies have implicated spore internalization by intestinal epithelial cells as a possible mechanism for recurrent CDI[35,36]. This idea is bolstered by the facts that carriage of toxigenic *C. difficile* among otherwise healthy individuals can go undetectable by 16S rRNA stool sequencing[37], and that 83–88% of those who experience recurrent CDI are reinfected by their original strain[38,39]. Taken together, these data may suggest that hidden spore reservoirs in the gut can germinate and reseed infection in susceptible hosts.

As such, we speculate that CD8$^+$ T$_{RM}$ cells are promoting the elimination of hidden spore reservoirs in the host. This theory may explain the oscillation in spore and vegetative cell counts observed in r.i.-vaccinated mice that clear infection (Fig. 2h,k and Fig. 5k,n,q,t), as spores may germinate in response to the clearance of vegetative bacteria until all reservoirs are eliminated. Colonic-resident CD8$^+$ T$_{RM}$ cells are maintained with minimal homeostatic turnover and do not repopulate from circulation[40], which validates the long-lived CD8$^+$ T$_{RM}$ cells (Fig. 5g,h) and clearance phenotypes (Fig. 5k,n,q,t) in our model. Future studies that define the mechanism of spore clearance are important for *C. difficile* immunization endeavours to halt transmission and recurrence.

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

## Methods

### Protein expression and purification

Point mutations were made in WT VPI10463 TcdA and WT TcdB2 as described previously[18–20]. Primer information can be found in Supplementary Table 2. TcdA, TcdB and CspC were recombinantly expressed and purified as previously described[18,41]. Plasmid information for all antigens is provided in Supplementary Table 2. Plasmids encoding NTAs were codon-optimized versions of the candidate NTAs from the *C. difficile* R20291 background which were synthesized at Genscript into a pET47b(+) vector and included a C-terminal 6×-histidine tag for protein purification. C40 peptidase 2, FlgG, FlgE, FlgK, CspC and polysaccharide deacetylase were transformed into *E. coli* BL21 (DE3) STARs (Supplementary Table 2). To express each NTA, 12 l of lysogeny broth medium supplemented with 50 mg l$^{-1}$ kanamycin were inoculated with an overnight culture to an optical density at 600 nm ($OD_{600}$) of 0.1. Cells were grown at 37 °C and 220 rpm. Expression was induced with 1 mM Isopropyl-β-D-1-thiogalactopyranoside (IPTG) once cells reached an $OD_{600}$ of 0.4–0.6. After 4 h, cells were centrifuged and the pellets were resuspended in 20 mM Tris (pH 8.0), 500 mM NaCl, 2% lysis mix (phenylmethylsulfonyl fluoride (0.1 mM), leupeptin (2 mg ml$^{-1}$), pepstatin (2 mg ml$^{-1}$), 2% DNase (2 mg ml$^{-1}$) and 2% lysozyme (10 mg ml$^{-1}$)). Bacterial suspensions were lysed three times using an EmulsiFlex C3 microfluidizer (Avestin) at 15,000 lb in$^{-2}$. Lysates were centrifuged at 40,000*g* for 45 min at 4 °C. NTAs were initially isolated from supernatant using a Ni$^{2+}$-affinity column (HisTrap FastFlow Crude; GE Healthcare). NTA eluents were further purified using an S-200 size-exchange column (GE Healthcare) in 20 mM HEPES (pH 6.9) with 50 mM NaCl on the ÄKTA Pure fast protein liquid chromatography system (Cytiva). All of the samples were treated using an endotoxin removal kit (Thermo Fisher Scientific) and sterile filtered through a 0.22-μm filter before being aliquoted for immunization studies and stored at −80 °C.

The ternary complex of FlgGEK was produced by co-purifying FlgG, FlgE and FlgK. In brief, supernatants of the three proteins were mixed in a 1:1:1 ratio after lysis and centrifugation, before purification by Ni$^{2+}$-affinity and S-200 size-exchange chromatography and subsequent endotoxin removal, filtration and freezing, as described above.

dmLT was provided by PATH (Acknowledgements and Data Availability) in 1× PBS supplemented with 0.05% Tween-20 (0.6 mg ml$^{-1}$).

### Animals and study design

Male and female C57BL/6J mice (Jackson Laboratories, 000664) were used in all studies. Mice were assimilated to the new facility 1 week before immunization. Mice were maintained at Vanderbilt University Medical Center under 12 h–12 h light–dark cycles under an ambient temperature of 23 °C (±3 °C) and 50% humidity (±20%), with ad libitum access to chow pellets and water. These studies were approved by the Institutional Animal Care and Use Committee at Vanderbilt University Medical Center and were performed using protocol M2200087-00. All animals were randomly assigned to experimental groups. Researchers were not blinded to groups throughout the animal experiments to properly monitor individual weight loss and morbidity during *C. difficile* infection according to institutional euthanasia guidelines.

For NTA-cocktail and individual NTA studies, 6-week-old male and female mice were immunized three times over the course of 28 days, with 14 days spanning between injections. Intraperitoneally injected mice received 5 μg of dmLT adjuvant with 5 μg of FlgGEK, C40 peptidase 2, CspC and/or polysaccharide deacetylase in sterile PBS in a total volume of 100 μl per injection. r.i.-treated mice received 25 μg of dmLT adjuvant with 5 μg of FlgGEK, C40 peptidase 2, CspC and/or polysaccharide deacetylase suspended in 200 μl of PBS. Mice were rectally instilled after faecal collection to empty the colon. r.i. occurred under anaesthesia using a sterilized metal ball-end gavage needle that was inserted into the rectum. The vaccine formula was pulsed into the colon, and the rectum was manually squeezed shut for 15 s after administration to prevent leakage, as described previously[42]. All vaccinations were administered within 2 h of antigen thaw. Faecal and serum samples were obtained before each vaccination and challenge. Mice were challenged 14 days after the final boost, as previously described[43,44]. Antibiotic treatment was administered by providing 0.5 mg ml$^{-1}$ cefoperazone in the drinking water ad libitum for 5 days, followed by a 2-day recovery period where normal water was provided before CDI through oral gavage. Two different *C. difficile* strains were used where indicated: WT R20291 and R20291 ΔAΔB[45], both administered at a dose of $1 \times 10^3$ spores per mouse. Mice were monitored daily for survival and weight loss. Animal cages were kept the same (left unchanged) for the entirety of the infection. Mice were humanely euthanized when weight loss exceeded 20% of their original body weight. Faecal samples were obtained daily during challenge for CFU enumeration.

For studies to analyse the toxicity of various TcdA and TcdB point mutants, 6-week-old mice were intraperitoneally injected as described above with 5 μg of dmLT and either 1 or 5 μg of the following toxins/toxin combinations: TcdA$_{GTX}$; TcdB2$_{GTX,L1106K}$; TcdA$_{GTX}$ + TcdB2$_{GTX,L1106K}$; TcdB2$_{GTX,L1106K,D1812G}$; or TcdB2$_{GTX,L1106K}$ + TcdB2$_{GTX,L1106K,D1812G}$. For combination vaccines with two antigens, 1 or 5 μg of each antigen was injected for a combined total antigenic amount of 2 or 10 μg. The mice were monitored for signs of morbidity and mortality for 7 days after injection.

For studies to optimize the amount of dmLT to include in vaccination, 6-week-old mice were either i.p. injected or rectally instilled with varying amounts of dmLT. Mice were intraperitoneally injected twice over 2 weeks with 0, 0.5, 1, 2.5 or 5 μg of dmLT alongside 5 μg of TcdA$_{GTX}$ or rectally instilled twice over 2 weeks with 0, 10, 15, 20 and 25 μg dmLT with 5 μg of TcdA$_{GTX}$. Sera and faeces were collected at days 0 (first dose), 14 (second dose), 28, 58 and 88 for enzyme-linked immunosorbent assays (ELISA) analysis of vaccine-induced humoral immune responses.

For studies comparing vaccination of dmLT adjuvant alone to toxin mutants with dmLT and toxin mutants with the NTA cocktail and dmLT, 2 cohorts of 6- and 12-week-old mice were immunized twice with vaccinations spaced 14 days apart. Intraperitoneally injected mice received 1 μg dmLT; 1 μg dmLT with 5 μg each of TcdA$_{GTX}$, TcdB2$_{GTX,L1106K}$, TcdB2$_{GTX,L1106K,D1812G}$; or 1 μg dmLT alongside 5 μg each of TcdA$_{GTX}$, TcdB2$_{GTX,L1106K}$, TcdB2$_{GTX,L1106K,D1812G}$, CspC, C40 peptidase 2 and FlgGEK. r.i.-treated mice received: 15 μg dmLT; 15 μg dmLT with 5 μg each of TcdA$_{GTX}$, TcdB2$_{GTX,L1106K}$ and TcdB2$_{GTX,L1106K,D1812G}$; or 15 μg dmLT alongside 5 μg each of TcdA$_{GTX}$, TcdB2$_{GTX,L1106K}$, TcdB2$_{GTX,L1106K,D1812G}$, CspC, C40 peptidase 2 and FlgGEK. Serum and faecal samples were collected as described above. Mice were challenged as stated above with WT *C. difficile* R20291. Mice were either euthanized for histopathological analysis and flow cytometry analysis on day 3 after infection or were monitored for 10 (6-week-old mice) or 15 (12-week-old mice) days after infection as described above.

For relapsing infection studies, 6 week-old mice were immunized by r.i. with the same experimental groups and same amounts of adjuvant and antigens as noted above. Then, 2 weeks after the final boost, the mice were challenged as stated above with WT *C. difficile* R20291. Two days after infection, the mice received 0.5 mg ml$^{-1}$ vancomycin ad libitum in the drinking water for 10 days to clear infection, as described elsewhere[46,47]. After receiving vancomycin for 10 days, the mice were returned to regular drinking water and monitored for 30 days to document relapsing infection. Stool samples were scored on a 1–5 scale for colour and composition, similar to our previous work[48], and were determined as follows: 5, normal, well-formed stool; 4, well-formed, slightly moist or slightly discoloured stool; 3, moist and discoloured stool; 2, soft diarrhoea without wet tail; and 1, wet tail, watery diarrhoea and empty rectum. Animal cages, food and water bottles were changed daily for the entirety of the infection to reduce cross-contamination risk.

For passive transfer studies, 6-week-old donor mice were immunized by rectal instillation with either 15 μg dmLT; or 15 μg of dmLT alongside 5 μg each of C40 peptidase 2 and FlgGEK. Then, 2 weeks after boost, faecal samples were collected from donor mice, as were colonic and

caecal contents post-mortem. Faecal and caecal contents were pooled per group and homogenized in 1:1 (w/v) PBS with 2% lysis mix (described above) before centrifugation for 10 min at 10,000*g*. The resultant supernatant was removed, sterile-filtered with 0.22 µm filters and incubated with anti-mouse IgG MicroBeads (Miltenyi Biotech) for extraction of faecal IgG using an LS Column (Miltenyi Biotech) and MidiMACS magnet system (Miltenyi Biotech). A total of 3.8 mg ml$^{-1}$ and 3.3 mg ml$^{-1}$ of faecal IgG were obtained from dmLT-only and C40 peptidase 2- and FlgGEK-vaccinated donors, respectively. Concentrations were obtained on a Nanodrop One C Microvolume UV Spectrophotometer (Thermo Fisher Scientific). Meanwhile, a cohort of 6-week-old recipient mice were r.i.-vaccinated with 15 µg of dmLT alongside 5 µg each of Tcd-A$_{GTX}$, TcdB2$_{GTX,L1106K}$, TcdB2$_{GTX,L1106K,D1812G}$ and CspC. The recipient mice were challenged as stated above with WT *C. difficile* R20291. At days 1, 4 and 7 after infection, mice received a passive transfer of isolated, sterile-filtered faecal IgG from donor mice either by i.p. (100 µl) or r.i. (200 µl). Recipient mice were monitored for 10 days after infection, as noted above. The animal cages were changed daily for the entirety of the infection.

For studies comparing vaccination formulas with and without the inclusion of CspC, 6-week-old mice were rectally instilled with 15 µg dmLT; 15 µg dmLT alongside 5 µg each of TcdA$_{GTX}$, TcdB2$_{GTX,L1106K}$, TcdB2$_{GTX,L1106K,D1812G}$ and CspC; or 15 µg dmLT alongside 5 µg each of TcdA$_{GTX}$, TcdB2$_{GTX,L1106K}$, TcdB2$_{GTX,L1106K,D1812G}$, C40 peptidase 2 and FlgGEK. The mice were challenged as stated above with WT *C. difficile* R20291. The animals were monitored for 10 days after infection as described above, and the cages were changed daily for the entirety of the infection.

For longevity studies, two cohorts of 6-week-old mice were immunized by r.i. with the same experimental groups and same amounts of adjuvant and antigens as noted above. One cohort of mice was challenged as previously stated with WT *C. difficile* R20291 60 days after boost, whereas the other was challenged 200 days after boost. Mice were either euthanized for flow cytometry analysis on day 3 after infection or were monitored for 10 days after infection as described above. Animal cages were changed daily for the entirety of both infections.

### Antigen-specific antibody measurements

ELISAs were performed. Nunc MaxiSorp 384-Well Plates (Thermo Fisher Scientific) were coated with 30 µl per well of 1 µg ml$^{-1}$ recombinant TcdA, TcdB2, CspC, C40 peptidase 2, FlgGEK or polysaccharide deacetylase. After overnight incubation at 4 °C, the plates were washed three times with PBS with 0.1% Tween-20 (PBS-T) (100 µl per well per cycle) and replaced with blocking solution (PBS with 0.1% Tween-20 and 2% (w/v) BSA). The plates were then incubated, rocking for 1 h at room temperature. Faeces were homogenized in 1:1 (w/v) PBS with 2% lysis mix (described above) and centrifuged for 10 min at 10,000*g*. Serum samples were diluted 1:100 in PBS with 2% lysis mix. Plates were washed three times with PBS-T before incubating with 20 µl per well of diluted samples for 2 h, rocking at room temperature. The plates were washed thrice with PBS-T before incubating for an hour with 30 µl per well of horseradish peroxidase (HRP) F(ab')$_2$-specific goat anti-mouse IgG (Jackson ImmunoResearch; 1:2,000) or goat anti-mouse IgA-HRP (Southern Biotech; 1:2,000) in PBS-T with 2% BSA. Plates were washed four times with PBS-T. Then, 30 µl of TMB substrate reagent (Thermo Fisher Scientific) was added to plates and 30 µl of 2 M sulfuric acid was added after 3 min of substrate development. Plates were recorded at wavelengths of 450 nm using a BioTek Cytation 5 plate reader (Agilent). Faecal antibody titres were normalized to milligram of faeces. Graphs were generated using GraphPad Prism v.10.4.2, and statistical differences between groups were assessed using one-way ANOVA and Tukey's HSD test.

### *C. difficile* enumeration

*C. difficile* burden in the stool was quantified by counting CFU from serially diluted stool in 1× PBS (pH 7.4) and plated on taurocholate-cycloserine-cefoxitin-fructose agarose (TCCFA) semi-selective medium[49]. Faeces was heat treated for 20 min at 75 °C to kill vegetative cells before plating on TCCFA to enumerate spores[50]. Similarly, bacterial burden in caecal and colonic tissue were quantified by macerating dissected tissues in 1× PBS until slurries were created and plating on TCCFA agar to enumerate total bacterial and spore burdens. Graphs were generated using GraphPad Prism v.10.4.2, and statistical differences between groups were assessed for each day using one-way ANOVA and Tukey's HSD test.

For PCR analyses, DNA from caecal and colonic tissues (macerated for bacterial burden above) was extracted using the QIAamp Power-Fecal Pro DNA Kit (Qiagen). The *C. difficile*-specific 16S rRNA-encoding gene PCR was used to verify the presence of the bacterium in tissue samples. The PCR set-up was identical to previous published protocols[51,52], with template DNA normalized to 50 ng among all of the samples. The following primers were used (ordered from IDT): forward primer 5′-TTGAGCGATTTACTTCGGTAAAGA-3′ (25 nmol, standard desalt purification); reverse primer 5′-CCATCCTGTACTGGCTCACCT-3′ (25 nmol, standard desalt purification)[53]. PCR products were run on a 1% agarose gel and imaged using a ChemiDoc MP (Bio-Rad).

For qPCR analyses, reactions were set up precisely as outlined elsewhere[53], using the same primers as listed above for PCR and TaqMan Fast Advanced Master Mix for qPCR (Thermo Fisher Scientific). Template DNA extracted above from the caecal and colonic tissues were normalized to 50 ng among all samples. Primer amplifications was identified using a *C. difficile*-specific 16S probe (ordered from IDT): 5′-6-FAM-CGGCGGACGGGTGAGTAACG-MBG-3′ (100 nmol, HPLC purification). Reactions were run on the QuantStudio 6 Flex qPCR system (Thermo Fisher Scientific). Graphs were generated using GraphPad Prism v.10.4.2, and statistical differences between groups were assessed using one-way ANOVA and Tukey's HSD.

### Differential scanning fluorometry

0.1 mg ml$^{-1}$ each of recombinant WT VPI TcdA, VPI TcdA$_{GTX}$, WT TcdB2, TcdB2$_{GTX,L1106K}$ and TcdB2$_{GTX,L1106K,D1812G}$ were loaded into glass capillaries and tested using a Tycho differential scanning fluorometer (NanoTemper) according to the manufacturer's instructions.

### Cell rounding assay

Vero–GFP cells[54] were seeded in a black, clear-bottom 96-well plate at a concentration of 25,000 cells per well and incubated overnight. Recombinant WT TcdB2, TcdB2$_{GTX}$, TcdB2$_{L1106K}$, TcdB2$_{GTX,L1106K}$, TcdB2$_{GTX,L1106K}$ and TcdB2$_{GTX,L1106K,D1812G}$ were serially diluted tenfold in culture medium, and 100 µl of each sample was added to individual wells in technical duplicate. The plates were statically incubated in a BioTek Cytation 5 plate reader (Agilent) at 37 °C under 5% $CO_2$ and imaged under the bright-field and GFP channels every 45 min at ×20 magnification for 24 h. This experiment was performed twice. The normalized number of rounded cells versus the number of total cells per image was calculated and analysed for concentrations of 1 pM at selected timepoints as previously described[54]. Graphs were generated using GraphPad Prism v.10.4.2, and statistical differences between groups were assessed using two-way ANOVA and Tukey's HSD test.

### Histopathology

Caeca and colons were fixed in 10% neutral-buffered formalin, dehydrated in graded ethanol series, cleared with xylenes and embedded in paraffin. Tissue blocks were sectioned at a thickness of 5 µm on the HM 335E microtome (Microm) onto Superfrost Plus microscope slides (Thermo Fisher Scientific). To assess histopathology, caecum and colon sections were stained with haematoxylin and eosin (Vector Labs), and conditions were masked for a board-certified gastrointestinal pathologist and a board-certified veterinary pathologist to separately score oedema, inflammation and epithelial damage based on published criteria ($n = 5$ per treatment)[55,56]. The averages of the scores from the two pathologists were reported. Histological scores were graphed using

GraphPad Prism v.10.4.2, and statistical differences were determined using one-way ANOVA and Tukey's HSD test. Presented images were captured using a BioTek Cytation 5 automated digital image system (Agilent). Whole slides were imaged at ×10 magnification to a resolution of 0.25 μm px$^{-1}$.

## *C. difficile* motility assay

WT *C. difficile* R20291 was grown at 37 °C in BHIS (37 g l$^{-1}$ brain–heart infusion broth supplemented with 5 g l$^{-1}$ yeast extract) anaerobic conditions using a COY anaerobic gas chamber (COY Laboratory Products) until mid-log phase. 1 ml of *C. difficile* was centrifuged at 10,000$g$ for 5 min and the supernatant was removed. The bacterial pellet was either resuspended in 1 ml of 1× PBS (pH 7.4), sterile-filtered dmLT faecal IgG, or sterile-filtered anti-FlgGEK and -C40 peptidase 2 faecal IgG (obtained as noted in the passive transfer mouse experiment, above) and incubated for 30 min. Then, 10 μl of bacterial–faecal IgG mixture was then spotted onto BHIS plates containing 0.3% agar, as previously described[57]. The plates were incubated at 37 °C for 8 h and room temperature for 16 h under anaerobic conditions. Measurements of the diameter of the widest point of bacterial growth were taken for each spotted colony with a ruler. PBS-incubated *C. difficile* grew a lawn on BHIS plates, so measurements were taken until the edge of swimming motility, before lawn growth.

## Spleen and mesenteric lymph node collection

Spleens were collected, macerated into single-cell suspensions, and filtered using 70 μm cell strainers in 1× Hank's buffered saline solution (1× HBSS) (Thermo Fisher Scientific). Red blood cells were lysed using ACK lysis buffer (Thermo Fisher Scientific) to obtain a single-cell suspension. Cells were centrifuged and resuspended in 500 μl 1× HBSS, counted and used immediately. Gut-draining mesenteric lymph nodes were collected and processed as described above (without ACK).

## Preparations of colons to obtain IELs and LPLs

IEL and LPL fractions were obtained and validated as previously described[58,59]. In brief, colons were collected, cleaned of fat residue and faeces, and cut open longitudinally before two sequential washes with cold 1× HBSS. The colons were cut into ~0.7 cm chunks and added to 15 ml conical vials with 2 ml of ice-cold HBSS. To these, 5 ml of DTT mix (1× HBSS supplemented with 20 mM HEPES, pH 8.0, 1 mM sodium pyruvate and 1 mM DTT) was added before a 15 min incubation at 37 °C. The conical vials were then shaken by hand for 2 min. The supernatant was transferred to a new 15 ml conical vial containing 5 ml cRPMI-10% FCS (RPMI-GlutaMax supplemented with 10% FBS and 10 mM HEPES, pH 8.0), whereas the tissue was reserved for LPL extraction (below). IELs in the supernatant were further enriched using a Percoll density gradient (Sigma-Aldrich). IELs were centrifuged for 20 min at 4 °C and 650$g$. The supernatant was aspirated, and IELs were resuspended in 250 μl of 1× PBS supplemented with 0.5% BSA and 1 mM EDTA (constituting PBE buffer). Cells were immediately stained for flow cytometry analyses.

For the LPL compartment, the remaining colonic tissue was kept in the original 15 ml conical vial. Then, 5 ml of EDTA-only mix (1× HBSS supplemented with 20 mM HEPES, pH 8.0, 1 mM sodium pyruvate and 0.5 mM EDTA) was added. The samples were incubated for 10 min at 37 °C. Next, the vials were shaken by hand for 2 min and the supernatant was discarded. Tissue was removed with tweezers and finely minced with scissors into a new 15 ml conical vial containing 3 ml digest mix (1× HBSS supplemented with 20% FBS, 3 mg collagenase D and 0.06 mg DNase I). Samples were then incubated at 37 °C for 30 min. Tubes were shaken by hand for 1 min before pipetting supernatant into new 15 ml conical vials containing 2 ml cRPMI-10% FCS (RPMI-GlutaMax supplemented with 10% FBS and 10 mM HEPES, pH 8.0). LPL cells were centrifuged 20 min at 4 °C and 650$g$. The supernatant was aspirated, and LPLs were resuspended in

250 μl of 1× PBE. Cells were immediately stained for flow cytometry analyses.

## Production of fluorescently labelled recombinant proteins for antigen-specific B cells

Fluorescently labelled recombinant TcdA, TcdB, CspC, C40 peptidase 2, and FlgGEK were prepared by biotinylation using a EZ-Link Sulfo-NHS-LC-Biotinylation kit (Thermo Fisher Scientific, 21435) and conjugated to Streptavidin-linked BV650 (BioLegend, 405231), Alexa Fluor 568 (Thermo Fisher Scientific, S11226), FITC (BioLegend, 405201), APC-Cy7 (BioLegend, 405208) and Alexa Fluor 680 (Thermo Fisher Scientific, S32358). Labelling reactions were calculated to produce a 1:1 ratio of fluorophore:protein. Labelled proteins were flash-frozen in liquid N$_2$ and stored at −80 °C until use to prevent degradation.

## Flow cytometry analysis of B and T cells

Single-cell suspensions were incubated with Zombie Near-IR cell viability dye (BioLegend, 423105) for 30 min at room temperature. Cells were washed with 1× PBE, centrifuged at 650$g$ for 10 min and then resuspended in 1× PBE with 5% normal goat serum (60 mg ml$^{-1}$, Thermo Fisher Scientific) for blocking at room temperature for 30 min. For T cell analysis, cells were stained with anti-CD45 (eFluor 450, 30-F11, 1:600, Thermo Fisher Scientific, 50-112-9409), anti-ΔγTCR (PerCP-Cy5.5, GL3, 1:600, BioLegend, 118117), 5-OP-RU tetramer (PE, 1:1,500, NIH Tetramer Core Facility; Data availability), anti-TCRb (Alexa Fluor 594, H57-597, 1:600, BioLegend, 109238), anti-B220 (FITC, RA3-6B2, 1:600, BioLegend, 103205), anti-CD4 (BV570, RM4-5, 1:150, BioLegend, 100541), anti-CD8 (Alexa Fluor 532, 53-6.7, 1:300, Thermo Fisher Scientific, 58-0081-80), anti-CD69 (APC, H1.2F3, 1:300, BioLegend, 104513), and anti-CD103 (PE-Fire 810, QA17A24, 1:300, BioLegend, 156919) for 30 min at room temperature. For B cell analysis, cells were stained with anti-CD45 (eFluor 450, 30-F11, 1:600, Thermo Fisher Scientific, 50-112-9409), anti-B220 (BV480, RA3-6B2, 1:300, BD Biosciences, 565631), anti-CD27 (BV510, LG.3A10, 1:300, BD Biosciences, 563605), anti-CD138 (BV785, 281-2, 1:300, BioLegend, 142534) and the fluorescently labelled recombinant vaccine antigens noted above for 30 min at room temperature. Cells were washed with 1× PBE and centrifuged at 650$g$ for 10 min before fixation in 4% paraformaldehyde fixation for 30 min at room temperature. Cells were then centrifuged, washed with 1× PBE and resuspended in 250 μl of 1× PBE. Flow cytometry data were acquired on a Cytek Aurora Spectral Flow Cytometer (Cytek) and analysed using SpectroFlow Software (v.3.3.0, Cytek). Before flow cytometry analysis, gating of cell populations was determined using Fluorescence Minus One (FMO) controls. FMO controls were prepared by staining replicate cellular samples and beads (Thermo Fisher Scientific, U20250) with all fluorophore-conjugated antibodies in a panel with the exception of one to be analysed for a given control. This accounted for fluorescence spillover and spread across channels and allowed for the precise determination of the positive and negative populations. The gating strategy is provided in the Supplementary Information. Graphs were generated using GraphPad Prism v.10.4.2, and statistical differences between groups were assessed using one-way ANOVA and Tukey's HSD test.

## Priming of naive dendritic cells with antigens and co-culture with T$_{RM}$ cells

BMDCs were isolated from naive C57BL/6J mice and differentiated using GM-CSF (Thermo Fisher Scientific, 315-03-50UG), IL-4 (Thermo Fisher Scientific, 214-14-50UG) and Flt3L (Thermo Fisher Scientific, 250-31L-50UG) as described previously[60,61]. The day before co-culture with T$_{RM}$ cells, BMDCs were activated with 10 ng ml$^{-1}$ of lipopolysaccharide (LPS, Thermo Fisher Scientific) and 1 μM of individual antigens or LPS alone. BMDCs were incubated with antigens for 24 h at 37 °C and 5% CO$_2$.

On the day of co-culture, colons from mice vaccinated by i.p. or r.i. with dmLT only or dmLT, toxins and NTAs were collected ($n$ = 3 per group).

LPLs were extracted from colons as stated above, pooled within groups, and stained for T cell markers. CD4$^+$ and CD8$^+$ T$_{RM}$ cells were flow-sorted using the Cytek Aurora CS Cell Sorter (Cytek) following the gating strategy provided in the Supplementary Information.

BMDCs were resuspended, washed to remove LPS and growth factors and seeded into 12-well tissue culture plates (Thermo Fisher Scientific) at 5,000 cells per well in 2 ml medium. Based on the counts obtained from the cell sorter, 1,000 CD4$^+$ or CD8$^+$ T$_{RM}$ cells were added to each well of DCs. Cells were incubated together at 37 °C for 72 h before 1 ml of co-culture supernatant was removed and flash-frozen at −80 °C for liquid bead cytokine analysis.

## Liquid bead cytokine array

Two custom 12-plex Mouse Luminex Discovery Assay kits (BioTechne) were used to quantify IL-7, IL-12 p70, IL-1β/IL-1F2, IL-2, IL-4, IL-5, IL-6, IL-10, IL-13, IL-17/IL-17a, IFNγ and IL-1α/IL-1F1. Assays were run in triplicate according to manufacturer's protocol. Acquisition was on a Luminex FlexMap 3D (Luminex). Data were analysed using Millipore Belysa (v.1.0.19) using a four-parameter logistic regression model for calculating concentrations from each standard curve. Graphs were generated using GraphPad Prism v.10.4.2, and statistical differences between groups were assessed using one-way ANOVA and Tukey's HSD test.

## Statistics and reproducibility

Data are presented as mean ± s.e.m. where applicable. Quantitative variables were tested for normal distribution using D'Agostino–Pearson normality tests. If normality was not indicated, then nonparametric statistical tests were used. Statistical tests, parametric or nonparametric, are listed in the figure legends for each experiment and in the corresponding Methods section. Sample variances were also similar between groups unless otherwise mentioned. The range of $n$ within experiments varies based on when samples are taken; for example, lower $n$ values on days 2–3 after infection may be due to animals succumbing to disease or sickness, resulting in an inability to provide faecal samples. Graph schematics were generated using GraphPad Prism v.10.4.2.

## Reporting summary

Further information on research design is available in the Nature Portfolio Reporting Summary linked to this article.

## Data availability

All data supporting the findings of this study are available in the Article and its Supplementary Information. The dmLT adjuvant was provided by PATH under the terms of a materials transfer agreement. PE-labelled 5-OP-RU tetramer was obtained through the NIH Tetramer Core Facility under the terms of a materials transfer agreement. Source data are provided with this paper.

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

**Acknowledgements** We thank J. White, M. Estrada and A. Bzami with the Center for Vaccine Innovation and Access at PATH for providing the dmLT adjuvant; A. Kruse for help with protein validation; and A. Louis Bourgeois, I. Georgiev and E. Skaar for their feedback on this study. dmLT production at PATH is funded by a grant from the Bill & Melinda Gates Foundation (INV-008763). The views expressed here are solely those of the authors and do not necessarily reflect the views of the Bill & Melinda Gates Foundation. The Vanderbilt Flow Cytometry Shared Resource is supported by the Vanderbilt Ingram Cancer Center (P30 CA068485) and the Vanderbilt Digestive Disease Research Center (P30 DK058404). Paraffin embedding of tissue was performed by the Vanderbilt Translational Pathology Shared Resource, which is supported by NCI/NIH Cancer Center Support Grant (P30CA068485) and the Shared Instrumentation Grants S10 OD023475-01A1 (Leica Bond RX), S10 OD016355 (Tissue MicroArray (TMA)) and IS1BX003154 (LCM). Funding was provided by National Institutes of Health grant U19 AI174999 (D.B.L., B.W.S., C.B.C. and M.R.N.).

**Author contributions** A.K.T., F.C.P.-G. and D.B.L. conceptualized this story. A.K.T., B.B.C.L., J.C., R.S., R.C.R., H.K.K., B.W.S. and D.B.L. produced reagents for this work. A.K.T. performed all experiments with assistance from F.C.P.-G., A.G.E. and E.J.B.; A.K.T. and S.M.Y. curated data with guidance from D.O.-V. Formal data analysis was performed by A.K.T. Blinded analysis and scoring of histopathological data were performed by M.K.W. and K.N.G.-C.; A.K.T. visualized all data and created figures. A.K.T. and J.C. validated data. D.B.L. supervised this work. The original draft of this manuscript was written by A.K.T. and reviewed and edited by A.K.T., F.C.P.-G. and D.B.L. with input from all of the authors. Funding was acquired by D.B.L., B.W.S., C.B.C. and M.R.N.

**Competing interests** A.K.T. and D.B.L. are listed as inventors on a patent application filed by Vanderbilt University Medical Center containing data published in this manuscript and covering multiple *C. difficile* vaccines and immunogens (US patent application 18/671,007). D.B.L. has research collaborations with AstraZeneca and Pfizer that are unrelated to this work and serves as a consultant to GSK. B.W.S. is a co-founder and owner at Turkey Creek Biotechnology, which was not involved in this work. C.B.C. receives grant support from Moderna, Pfizer, and Vedanta; serves as a consultant to GSK, Merck, CommenseBio, TDCowen, Guidepoint Global, AstraZeneca and Delbiopharm; and serves on the data and safety monitoring boards for studies sponsored by GSK and Bavarian Nordic—each unrelated to this work. The other authors declare no competing interests.

**Additional information**
**Correspondence and requests for materials** should be addressed to D. Borden Lacy.

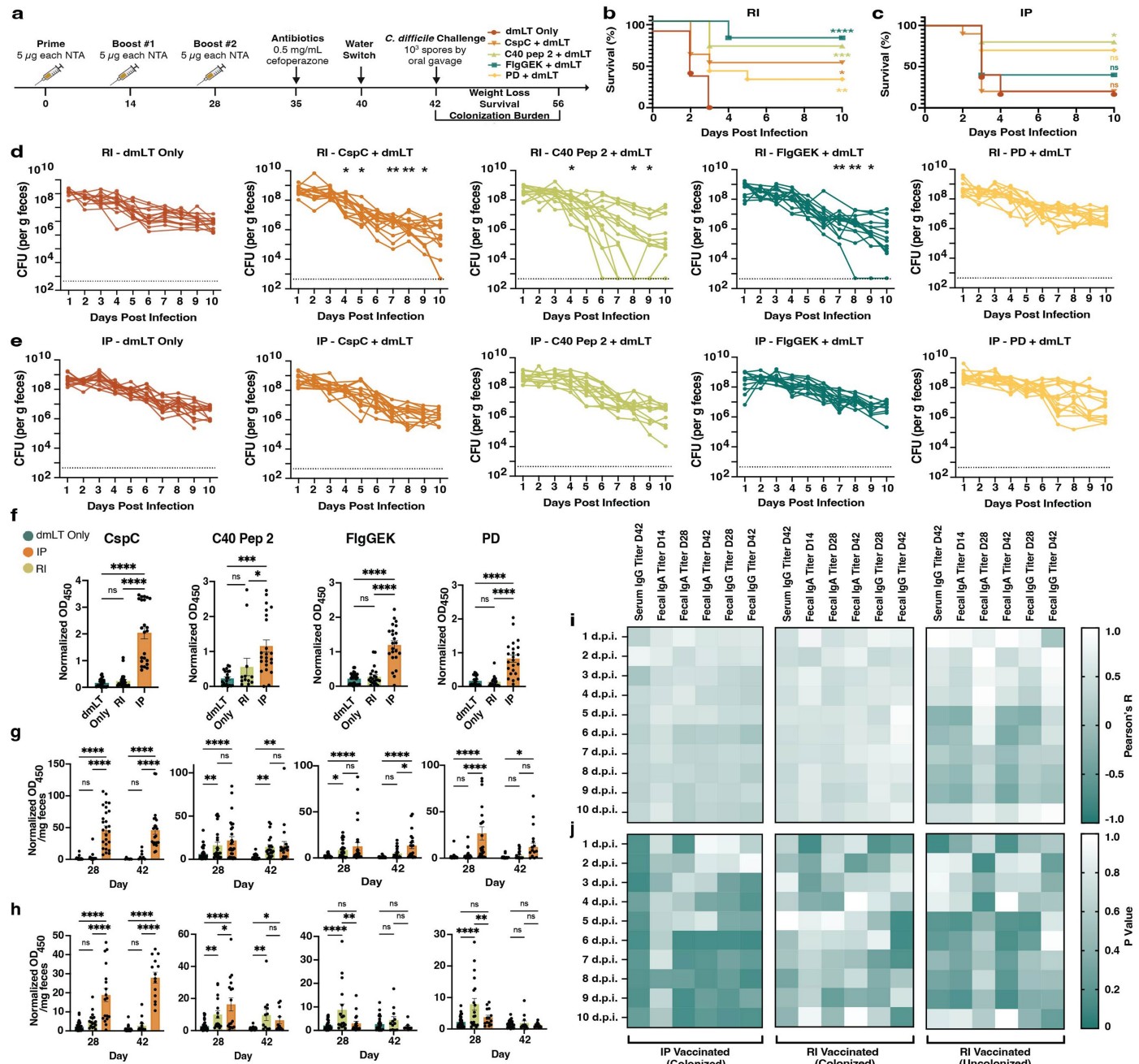

**Extended Data Fig. 1 | RI of CspC, C40 Peptidase 2, and FlgGEK reduce**
*C. difficile* **colonization burden. a**, Experimental schema. The diagram was
created using BioRender. **b,c**, Survival of WT *C. difficile* R20291 infection.
**d,e**, Enumeration of *C. difficile* R20291 ΔAΔB bacteria in faeces. *P* value relative
to dmLT control at the same time point. Individual lines correspond to individual
mice. Limit of detection shown by dotted line, 500 CFU/g faeces. **f–h**, Sera IgG
(**f**), faecal IgA (**g**), and faecal IgG (**h**) responses against antigens at days 28 and 42.
Mean +/− SEM. **i**, Pearson's R correlation coefficient and **j**, corresponding p-value
heat maps comparing colonization burden with humoral titres in IP or RI mice
that were either colonized or uncolonized 10 days post-infection with *C. difficile*.

Negative Pearson's R values correspond to colonization burden decrease.
Statistical significance was calculated by means of [(b-c)] log-rank Mantel-Cox
test, [(d-h), (d-e) each day analysed individually] one-way ANOVA with Tukey's
correction; *$P \leq 0.05$, **$P \leq 0.01$, ***$P \leq 0.001$, ****$P \leq 0.0001$. **1b-e:** n = 10 per
group, decreases in n due to animal mortality during infection; **1f-j:** n = 5-15 per
group, dependent on ability to collect sample. Individual data points are
represented and are pooled from two independent experiments in **1b-e**.
CFU, colony forming units; d.p.i., days post-infection; IP, intraperitoneal
injection; RI, rectal instillation; C40 Pep 2, C40 peptidase 2; PD, polysaccharide
deacetylase.

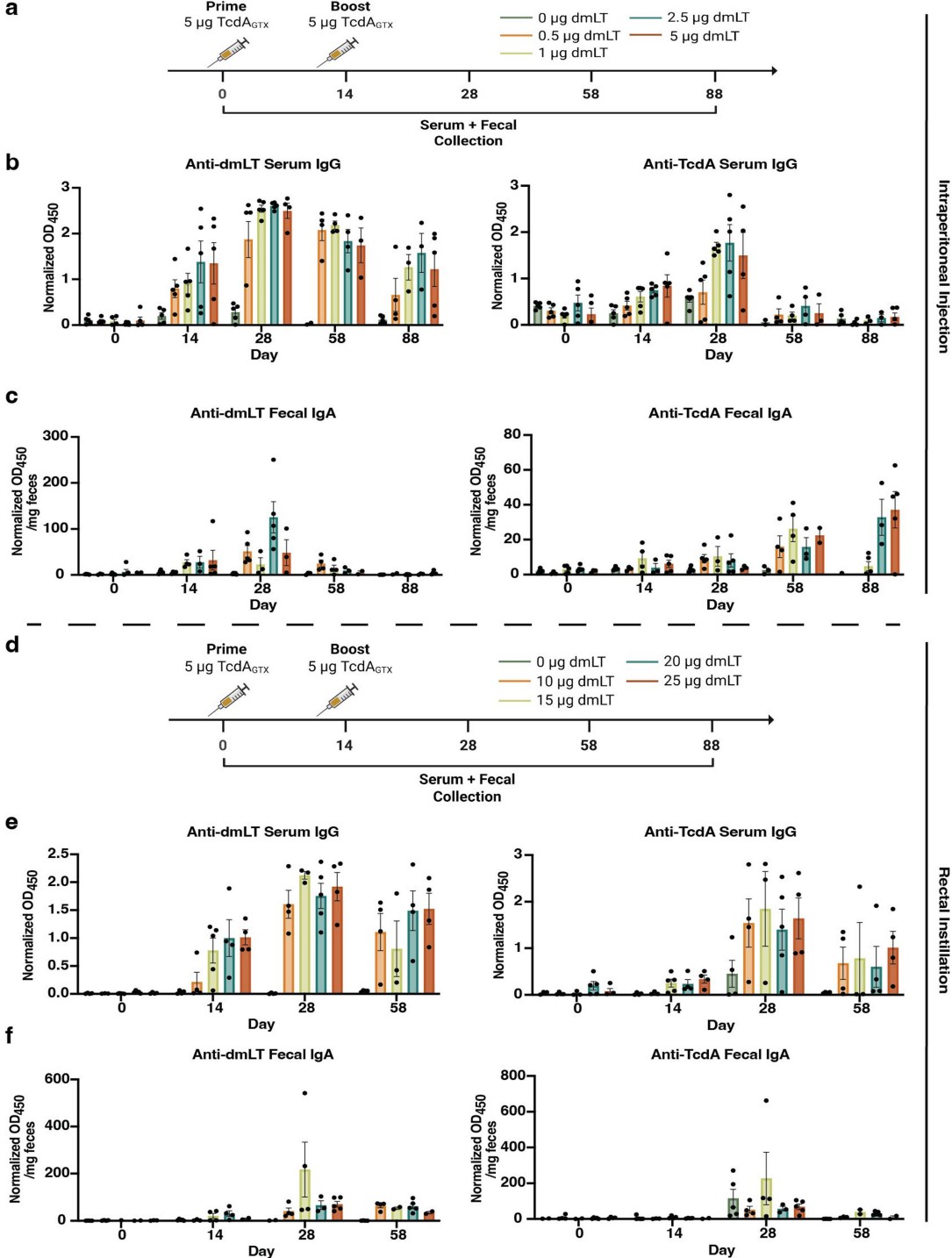

**Extended Data Fig. 2 | Rectal instillation and intraperitoneal injection of TcdA_GTX with either >10 µg or >1 µg of dmLT, respectively, increases anti-TcdA serum IgG and faecal IgA. a**, Experimental schema for intraperitoneal injection. The diagram was created using BioRender. **b,c**, Sera IgG (**b**) and faecal IgA (**c**) titres against TcdA and dmLT in intraperitoneally-injected mice.

**d**, Experimental schema for rectal instillation. The diagram was created using BioRender. **e,f**, Sera IgG (**e**) and faecal IgA (**f**) titres against TcdA and dmLT in rectally-instilled mice. All data shown as mean +/− SEM. n = 3-5 per group. Statistical significance was calculated by means of [(b-c), (e,f)] two-way ANOVA with Tukey's correction; *P ≤ 0.05, **P ≤ 0.01, ***P ≤ 0.001, ****P ≤ 0.0001.

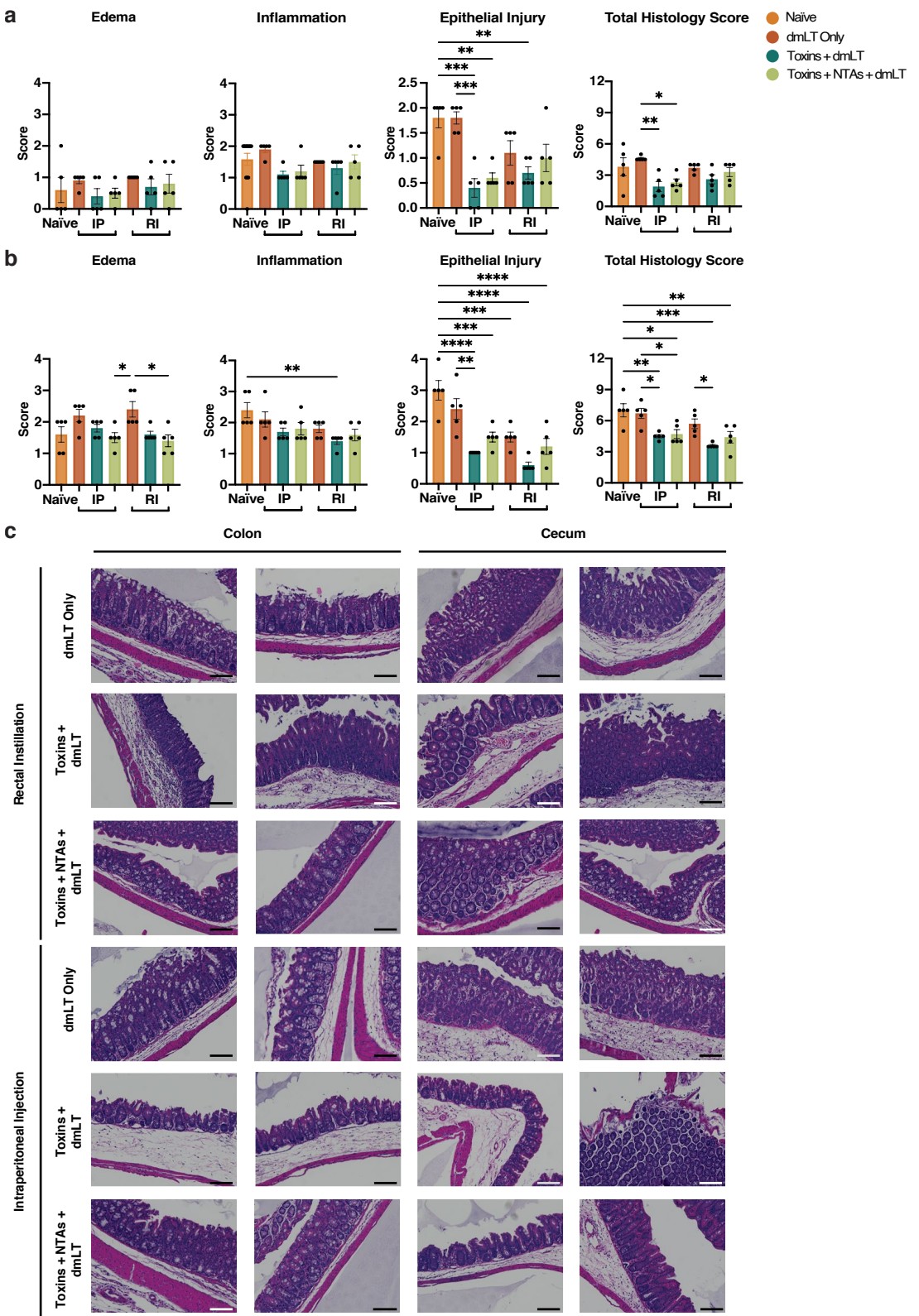

**Extended Data Fig. 3 | Mice vaccinated with toxins and NTAs by RI have less epithelial damage to colons and ceca. a,b,** Histopathology scores of colons (**a**) and caeca (**b**). **c,** Representative haematoxylin & eosin (H&E) images of colons and caeca from each group of vaccinated mice at 3 days post infection with *C. difficile* R20291. Scale bar, 40 μm at 10x magnification. [3a-b] shown as mean +/− SEM. n = 5 per group. Statistical significance was calculated by means of [(a-b)] one-way ANOVA with Tukey's correction; *$P \leq 0.05$, **$P \leq 0.01$, ***$P \leq 0.001$, ****$P \leq 0.0001$. n = 5 per group. Total histology score is the summation of oedema, inflammation, and epithelial injury scores.

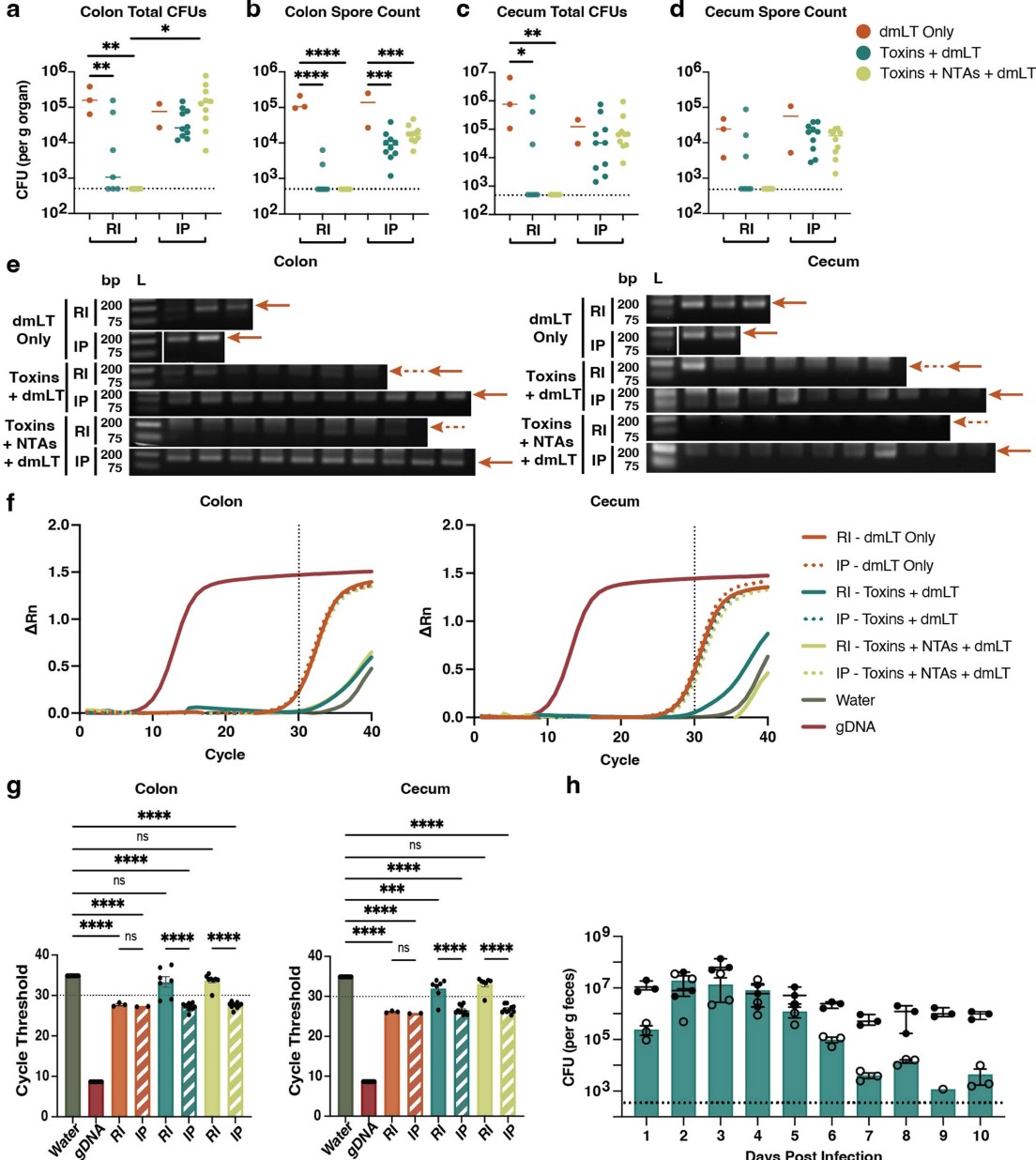

**Extended Data Fig. 4 | RI-vaccination with the toxin and NTA formula eliminates *C. difficile* from tissue. a–d**, Enumeration of total *C. difficile* R20291 bacteria (vegetative cells and spores) (**a**,**c**) and spores (**b**,**d**) from macerated colons and caeca of vaccinated mice fifteen days post-R20291 infection. Limit of detection (dotted line) is 500 CFU/g faeces. **e**, PCR of *C. difficile* 16S gene from bulk DNA extracted from macerated colons and caeca from **a**–**e**, with each lane representing one mouse. Desired amplicon is 175 bp (expected size indicated by red arrow; solid line indicates bands are present whereas dotted shows lack of bands). Distinct bands can be noted for certain mice; smeared bands are background DNA from extracted tissues. **f**, Normalized reporter values from *C. difficile* 16S qPCR of bulk DNA extracted from macerated colons and caeca from **a**–**d**. qPCR was performed in triplicate per sample. Data reported as the mean of all samples' experimental averages, normalized from baseline. Standard threshold (dotted line) is 30 cycles. **g**, Cycle threshold from *C. difficile* 16S qPCR of bulk DNA extracted from macerated colons and caeca from **a**–**d**. qPCR was performed in triplicate per sample. Mean +/− SEM. Standard threshold (dotted line) is 30 cycles. **h**, Total colonization burden (vegetative and spore counts combined; closed circle) versus spore counts (open circle) in faeces of naive, unvaccinated mice infected with WT *C. difficile* R20291. Limit of detection (dotted line) is 500 CFU/g faeces. **4a-g:** n = 2-10 per group, dependent on survival at 15 days post-infection; **4h:** n = 3 per group. CFU, colonization forming units; RI, rectal instillation; IP, intraperitoneal injection; L, ladder; bp, base-pairs; ΔRn, normalized reporter signal above baseline; gDNA, *C. difficile* R20291 genomic DNA. For source gel data, see Supplementary Information.

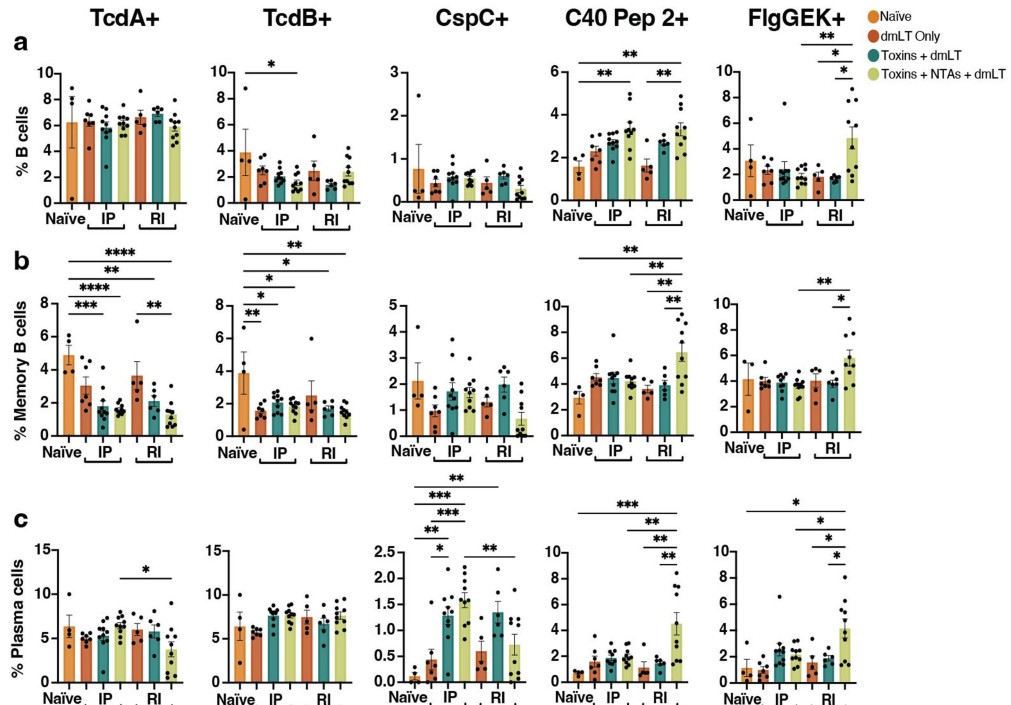

**Extended Data Fig. 5 | Flow cytometric results of vaccinated mesenteric lymph nodes. a–c**, Percent parent of gated mesenteric lymph node total (**a**), memory (**b**, B220+, CD27+), and plasma (**c**, B220+, CD138+) B cells with affinity for each vaccine antigen. All data shown as mean +/− SEM. n = 3–10 per group, dependent on infection survival. Roughly 150,000 events were recorded.

Statistical significance was calculated by means of [(a-c)] one-way ANOVA with Tukey's correction; *$P \le 0.05$, **$P \le 0.01$, ***$P \le 0.001$, ****$P \le 0.0001$. Individual data points are represented and are pooled from two independent experiments. Bmem, memory B cells; C40 Pep 2, C40 peptidase 2.

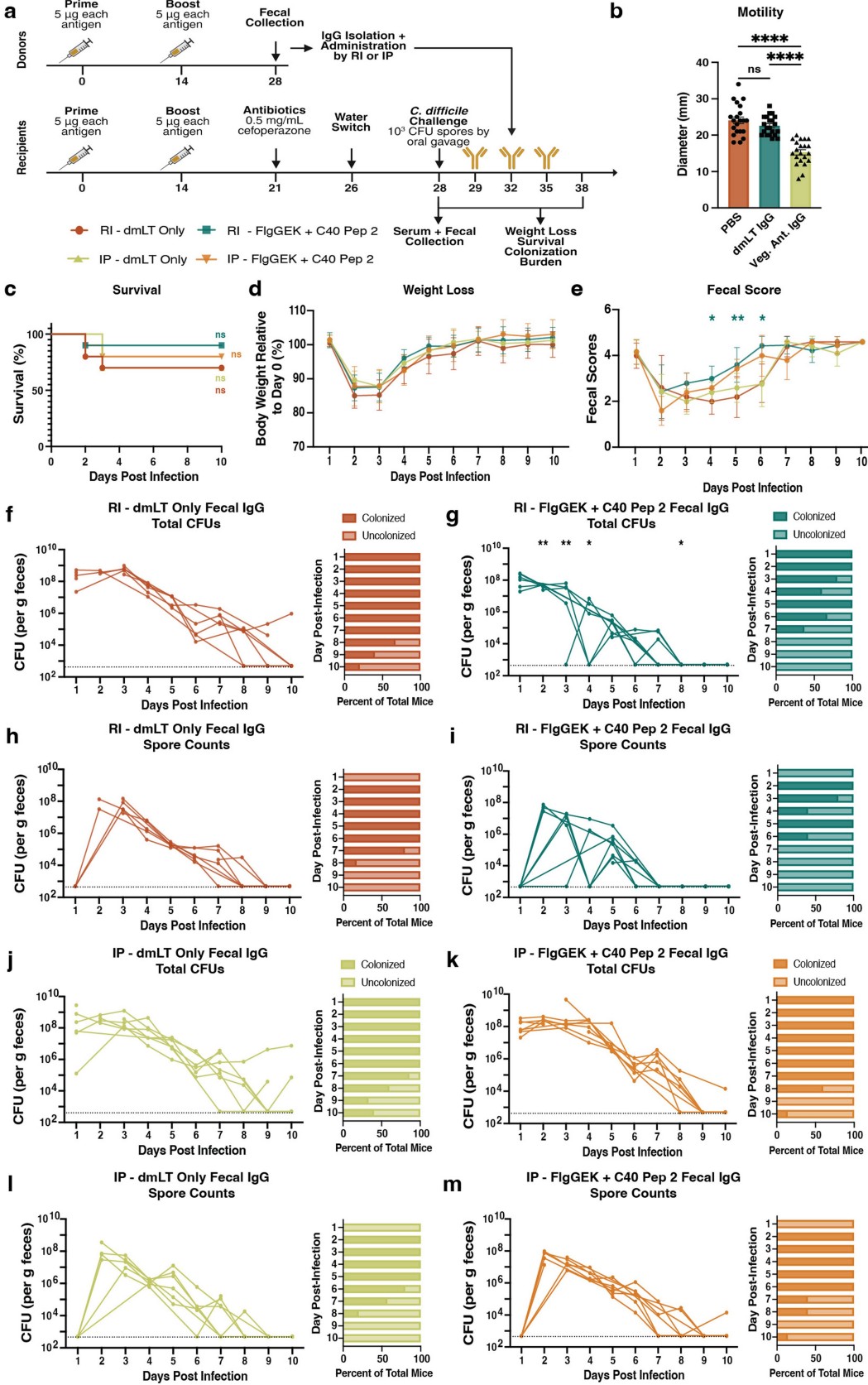

**Extended Data Fig. 6** | See next page for caption.

**Extended Data Fig. 6 | Passive transfer of anti-C40 peptidase 2 and -FlgGEK faecal IgG by RI reduces vegetative bacterial burden. a**, Passive transfer schematic. The diagram was created using BioRender. Recipient mice were RI-vaccinated with inactivated toxins, CspC, and dmLT before undergoing challenge. On days 1, 4, and 7 post-infection, recipient mice were passively transferred faecal IgG by either IP or RI, which was isolated by donor mice that had been RI-vaccinated with either dmLT alone or C40 peptidase 2, FlgGEK, and dmLT. **b**, *C. difficile* R20291 motility with or without faecal IgG pre-incubation. **c**, Survival of WT *C. difficile* R20291 infection. **d**, Weight loss. **e**, Stool scores. **j**–**m**, Enumeration of total *C. difficile* R20291 bacteria (vegetative cells and spores) (**f** to **g**, **j** to **k**) and spores (**h** to **i**, **l** to **m**) in faeces. *P* value relative to control at the same time point. Individual lines correspond to individual mice. Individual lines correspond to individual mice. Limit of detection shown by dotted line, 500 CFU/g faeces. Mean +/− SEM shown for [b-e]. Statistical significance was calculated by means of [c] log-rank Mantel-Cox test, and [(b),(d-e), (f-m, each day analysed individually)] one-way ANOVA with Tukey's correction; *$P \leq 0.05$, **$P \leq 0.01$, ***$P \leq 0.001$, ****$P \leq 0.0001$. **6b:** n = 21 per group; **6c-m:** n = 10 per group, decreases in n due to animal mortality during infection. IP, intraperitoneal injection; RI, rectal instillation; CFU, colony forming units; C40 pep 2, C40 peptidase 2; Veg. Ant., vegetative antigens.

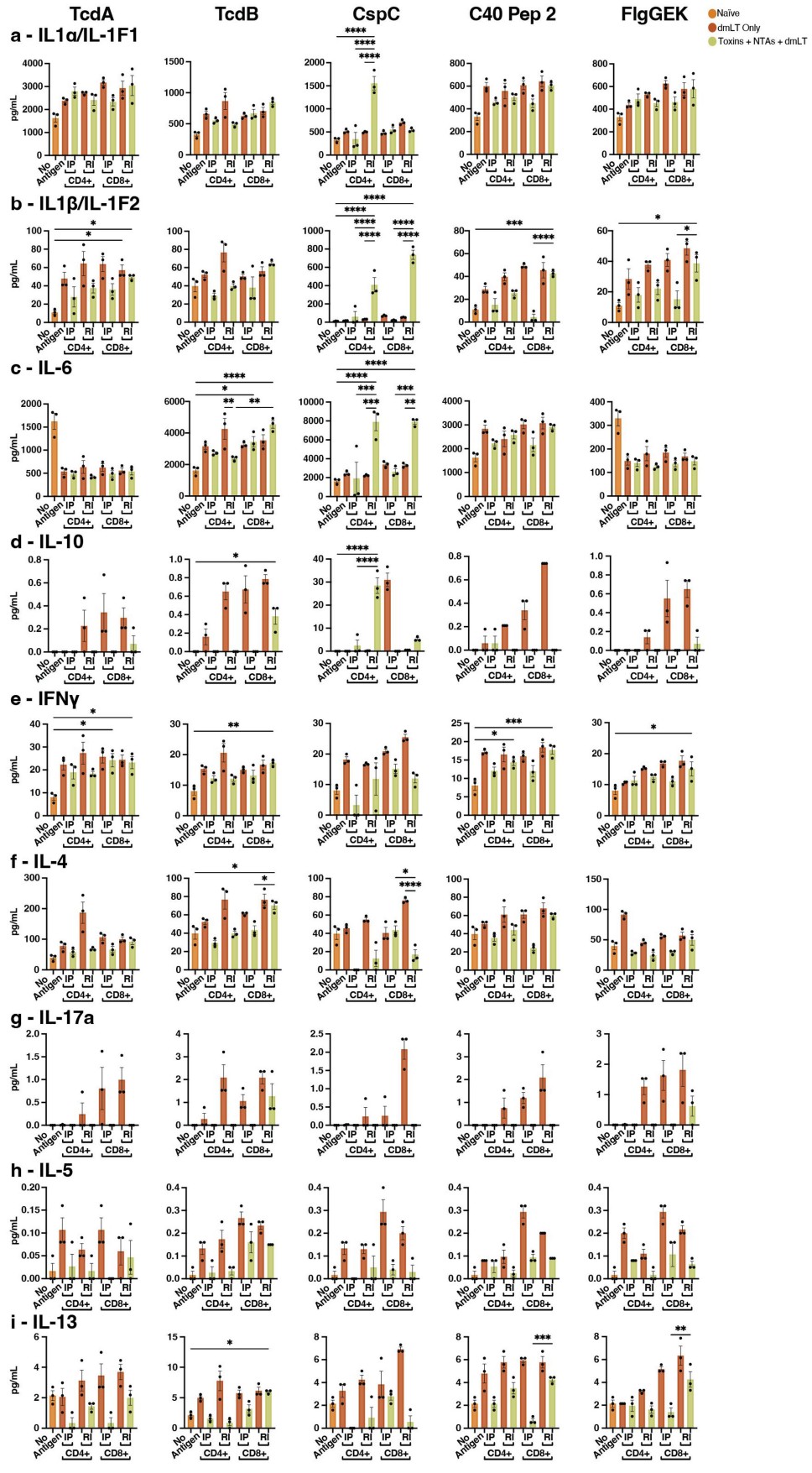

**Extended Data Fig. 7** | See next page for caption.

**Extended Data Fig. 7 | CspC-primed dendritic cells activate Th17-skewed cytokine responses from CD4+ and CD8+ tissue-resident memory T cells from RI-vaccinated mice during co-culture. a–i,** Bone marrow derived dendritic cells were primed individually with TcdA, TcdB, CspC, C40 peptidase 2, or FlgGEK in vitro and incubated for three days with T cells from mice vaccinated by RI or IP with dmLT alone or the toxins, NTAs, and dmLT. Cytokines in the supernatants were quantified by liquid bead array. All data shown as mean +/− SEM. n = 3 per group. Statistical significance was calculated by means of [(a-i)] one-way ANOVA with Tukey's correction; *$P \le 0.05$, **$P \le 0.01$, ***$P \le 0.001$, ****$P \le 0.0001$. n = 3 per group. C40 Pep 2, C40 peptidase 2.

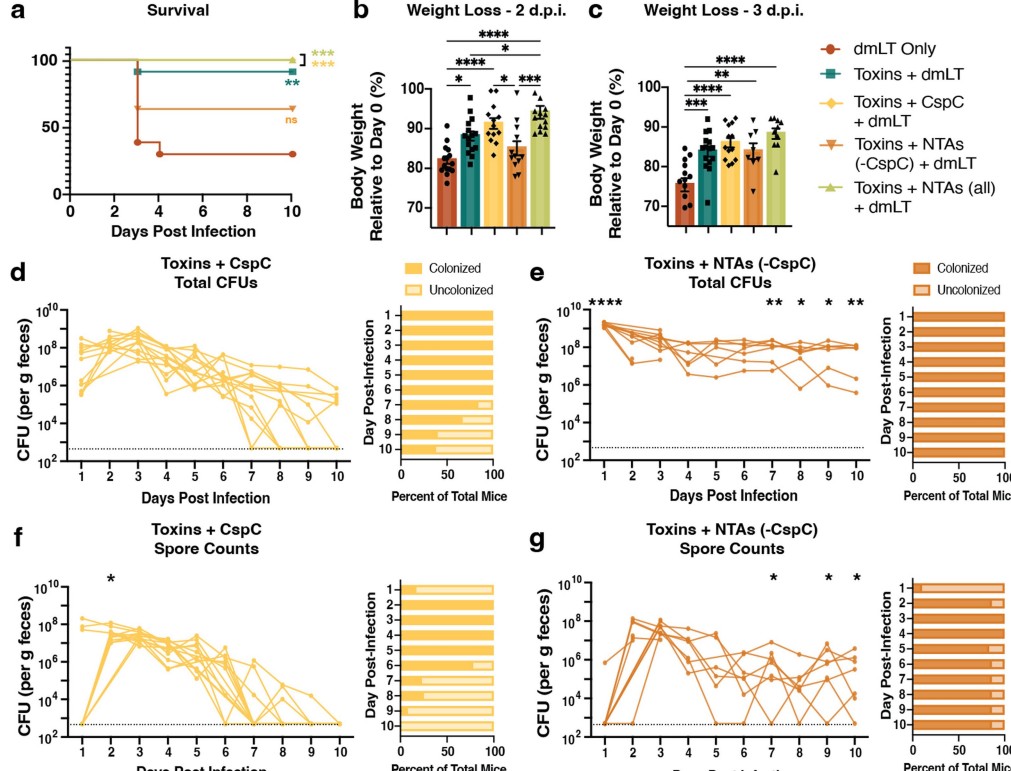

**Extended Data Fig. 8 | Immunization of inactivated toxins and CspC is sufficient to protect against death but does not fully clear *C. difficile*.**
**a**, Survival of WT *C. difficile* R20291 infection. **b**,**c**, Weight loss. Mean +/− SEM. **d**–**g**, Enumeration of total *C. difficile* R20291 bacteria (vegetative cells and spores) (**d**,**e**) and spores (**f**,**g**) in faeces. *P* value relative to RI dmLT-only control at the same time point (shown in Fig. 2). Individual lines correspond to individual mice. Limit of detection shown by dotted line, 500 CFU/g faeces. **8b-g:** n = 10 per group, decreases in n due to animal mortality during infection. Weight loss and survival data for other groups taken from Fig. 2 (n = 10-15 per group). Statistical significance was calculated by means of [(a)] log-rank Mantel-Cox test, and [(b-g), (d-g, each day analysed individually)] one-way ANOVA with Tukey's correction; *$P \le 0.05$, **$P \le 0.01$, ***$P \le 0.001$, ****$P \le 0.0001$. RI, rectal instillation; d.p.i., days post-infection; CFU, colony forming units; NTAs, non-toxin antigens.

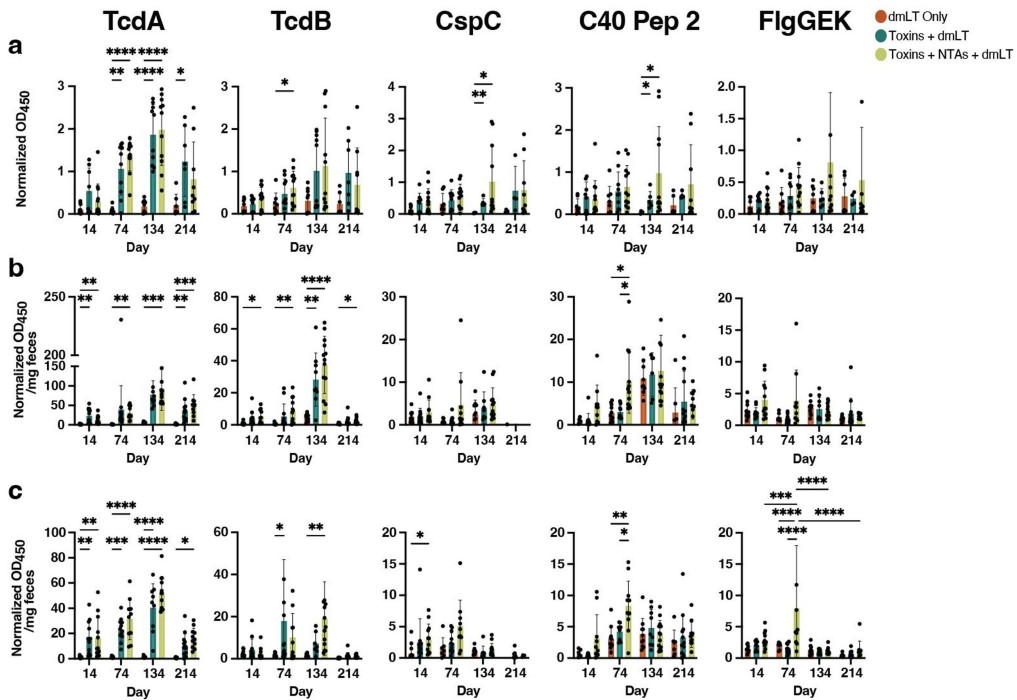

**Extended Data Fig. 9 | Toxin and NTA-vaccinated mice exhibit significantly increased sera and faecal humoral titres against antigens 60- and 120-days post-boost. a–c**, Sera IgG (**a**), faecal IgA (**b**), and faecal IgG (**c**) responses. n = 10-15 per group, based on ability to obtain sample. All data shown as mean +/− SEM. Statistical significance was calculated by means of [(a-c)] two-way ANOVA with Tukey's correction; *$P \le 0.05$, **$P \le 0.01$, ***$P \le 0.001$, ****$P \le 0.0001$.

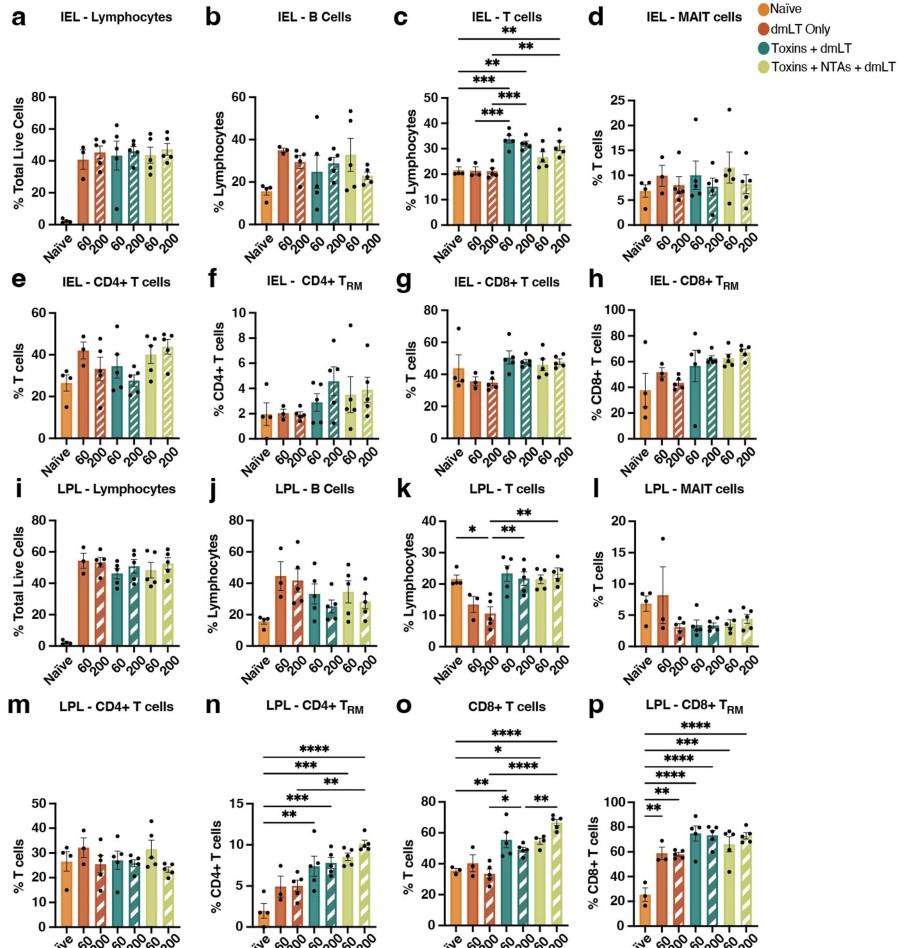

**Extended Data Fig. 10 | Flow cytometric results of colonic intraepithelial and lamina propria lymphocytes 60- and 200-days post-immunization. a–p,** Enumeration and percent parent of gated intraepithelial lymphocyte (**a–h**) and lamina propria lymphocyte (**i–p**) populations in the colon taken at day 3 post-infection. All data shown as mean +/− SEM. n = 3-5 per group, depending on survival at time of harvest. Statistical significance was calculated by means of [(a-p)] one-way ANOVA with Tukey's correction; *$P \le 0.05$, **$P \le 0.01$, ***$P \le 0.001$, ****$P \le 0.0001$. IEL, intraepithelial lymphocytes; LPL, lamina propria lymphocytes; T_{RM}, tissue resident-memory (CD103 + , CD69 + ); MAIT, mucosal-associated invariant T cells.

# Reporting Summary

## Statistics

For all statistical analyses, confirm that the following items are present in the figure legend, table legend, main text, or Methods section.

| n/a | Confirmed | |
|---|---|---|
| ☐ | ☒ | The exact sample size (*n*) for each experimental group/condition, given as a discrete number and unit of measurement |
| ☐ | ☒ | A statement on whether measurements were taken from distinct samples or whether the same sample was measured repeatedly |
| ☐ | ☒ | The statistical test(s) used AND whether they are one- or two-sided<br>*Only common tests should be described solely by name; describe more complex techniques in the Methods section.* |
| ☐ | ☒ | A description of all covariates tested |
| ☒ | ☐ | A description of any assumptions or corrections, such as tests of normality and adjustment for multiple comparisons |
| ☐ | ☒ | A full description of the statistical parameters including central tendency (e.g. means) or other basic estimates (e.g. regression coefficient) AND variation (e.g. standard deviation) or associated estimates of uncertainty (e.g. confidence intervals) |
| ☐ | ☒ | For null hypothesis testing, the test statistic (e.g. *F*, *t*, *r*) with confidence intervals, effect sizes, degrees of freedom and *P* value noted<br>*Give P values as exact values whenever suitable.* |
| ☒ | ☐ | For Bayesian analysis, information on the choice of priors and Markov chain Monte Carlo settings |
| ☒ | ☐ | For hierarchical and complex designs, identification of the appropriate level for tests and full reporting of outcomes |
| ☐ | ☒ | Estimates of effect sizes (e.g. Cohen's *d*, Pearson's *r*), indicating how they were calculated |

*Our web collection on statistics for biologists contains articles on many of the points above.*

## Software and code

Policy information about availability of computer code

| | |
|---|---|
| Data collection | Flowjo V10, GraphPad Prism v.8.0, NIS-Elements Advanced Research software Version 4.50,ImageJ v1.54h |
| Data analysis | Statistical analysis were performed on GraphPad Prism v.8.0. Flowcytometry data were analyzed on FlowJo software package (Flowjo V10). |

For manuscripts utilizing custom algorithms or software that are central to the research but not yet described in published literature, software must be made available to editors and reviewers. We strongly encourage code deposition in a community repository (e.g. GitHub). See the Nature Portfolio guidelines for submitting code & software for further information.

## Data

Policy information about availability of data

All manuscripts must include a data availability statement. This statement should provide the following information, where applicable:
- Accession codes, unique identifiers, or web links for publicly available datasets
- A description of any restrictions on data availability
- For clinical datasets or third party data, please ensure that the statement adheres to our policy

The data that support the findings of this study are available within the paper and its supplementary Information files. RNA-seq data can be accessed through GSE302643. The data that support the findings of this study will be available from the corresponding author upon reasonable request.

# Research involving human participants, their data, or biological material

Policy information about studies with human participants or human data. See also policy information about sex, gender (identity/presentation), and sexual orientation and race, ethnicity and racism.

| | |
|---|---|
| Reporting on sex and gender | Blood samples from healthy male and female individuals were used in human macrophage experiments. There was no specific preference of covariate-relevant population characteristics for these individuals.<br><br>Paraffin blocks of tumor samples from 37 melanoma patients treated with Pembrolizumab at Hospital of University of Pennsylvania were used for immunofluorescence. |
| Reporting on race, ethnicity, or other socially relevant groupings | Information for melanoma is shown in Supplementary Table 2. |
| Population characteristics | Information for melanoma is shown in Supplementary Table 2. |
| Recruitment | Healthy individuals were recruited by the Human Immunology Core at the University of Pennsylvania, with no specific selection basis for blood donation used for human macrophage isolation.<br><br>All melanoma patients were seen at Hospital of University of Pennsylvania during their clinical care. There was no selection bias. |
| Ethics oversight | Blood samples from human healthy donors were collected by the Human Immunology Core at the University of Pennsylvania with the approval from the ethics committee and institutional review board. Written consent was obtained from each healthy donor before blood collection. All experiments involving blood samples from healthy donors were performed in accordance with relevant ethical regulations.<br><br>All studies were conducted under protocols approved by University of Pennsylvania. All melanoma patients or families provided informed consent for research use of biospecimens and clinical data under an institutional approved protocol (IRB #703001). |

Note that full information on the approval of the study protocol must also be provided in the manuscript.

# Field-specific reporting

Please select the one below that is the best fit for your research. If you are not sure, read the appropriate sections before making your selection.

☒ Life sciences  ☐ Behavioural & social sciences  ☐ Ecological, evolutionary & environmental sciences

For a reference copy of the document with all sections, see nature.com/documents/nr-reporting-summary-flat.pdf

# Life sciences study design

All studies must disclose on these points even when the disclosure is negative.

| | |
|---|---|
| Sample size | For mouse studies, the sample size (n≥5 /group) was determined based on our previous experience with the models to provide sufficient statistical power (Chen et al., Nature 2018; Zhong et al. Nature Cancer, 2025;Zhong et al. Cancer Research, 2023; Zhang et al, Dev Cell, 2022). For human studies, the sample size was determined based on previous experience with the models to provide sufficient statistical power (Chen et al., Nature 2018;Moshe Sade-Feldman., Cell 2018 ). |
| Data exclusions | No data were excluded from analysis. |
| Replication | Figure legends describe the number of repeats for the experiments. |
| Randomization | Mice and other samples in experiments were allocated randomly to each treatment group. |
| Blinding | All experiments were performed in a blinded fashion. Downstream analyses of samples (immunofluorescence staining and flow cytometry) were performed in a blinded fashion; the individuals performing the assays were not aware of the treatment groups until data analyses were completed. For experiments other than those involving animals, the investigators were blinded to group allocation during data collection and/or analysis. |

# Reporting for specific materials, systems and methods

We require information from authors about some types of materials, experimental systems and methods used in many studies. Here, indicate whether each material, system or method listed is relevant to your study. If you are not sure if a list item applies to your research, read the appropriate section before selecting a response.

## Materials & experimental systems

| n/a | Involved in the study |
|---|---|
| ☐ | ☒ Antibodies |
| ☐ | ☒ Eukaryotic cell lines |
| ☒ | ☐ Palaeontology and archaeology |
| ☐ | ☒ Animals and other organisms |
| ☐ | ☒ Clinical data |
| ☒ | ☐ Dual use research of concern |
| ☒ | ☐ Plants |

## Methods

| n/a | Involved in the study |
|---|---|
| ☒ | ☐ ChIP-seq |
| ☐ | ☒ Flow cytometry |
| ☒ | ☐ MRI-based neuroimaging |

# Antibodies

| Antibodies used | Anti-human GM-CSFRα Cell Signaling Technology WB IF 1:1000 for WB;<br>1:100 for IP Cat#: 69817<br>Anti-mouse GM-CSFRα Biorbyt WB 1:500 Cat#: orb256474<br>Anti-M-CSFR Cell Signaling Technology WB 1:1000 Cat#: 3152<br>Anti-phospho-JAK2 Cell Signaling Technology WB 1:1000 Cat#: 3771<br>Anti-total JAK2 Cell Signaling Technology WB 1:1000 Cat#: 3230<br>Anti-phospho-STAT5 Cell Signaling Technology WB 1:1000 Cat#: 4322<br>Anti-total STAT5 Cell Signaling Technology WB 1:1000 Cat#: 94205<br>Anti-phospho-NF-κB Cell Signaling Technology WB 1:1000 Cat#: 3033<br>Anti-total NF-κB Cell Signaling Technology WB 1:1000 Cat#: 8242<br>Anti-β-actin Cell Signaling Technology WB 1:1000 Cat#: 4967<br>Anti-HRS Cell Signaling Technology WB 1:1000 Cat#: 15087S<br>Anti-HRS (phospho-Y216) Invitrogen WB 1:500 Cat#: PA5-114579<br>Anti-HRS (phospho-Y216) Biorbyt IF; IP 1:50 Cat#: orb644733<br>Anti-phospho-Src (Tyr416) Cell Signaling Technology WB 1:1000 Cat#: 2101<br>Anti-Src Cell Signaling Technology WB 1:1000 Cat#: 2109<br>Anti- DYKDDDDK (Flag) Cell Signaling Technology WB; IP 1:1000 for WB; 1:50 for IP Cat#: 14793<br>Mouse IgG isotype control BioLegend Blocking 10 µg/ml Cat#: 401404<br>Rat IgG isotype control Bio X Cell Blocking 10 µg/ml Cat#: BE0090<br>Anti-human M-CSFR BioLegend Blocking 10 µg/ml Cat#: 347302<br>Anti-mouse M-CSFR BioLegend Blocking 10 µg/ml Cat#:135502<br>Anti-human GM-CSFR BioLegend Blocking 10 µg/ml Cat#: 305902<br>Anti-mouse GM-CSFR Invitrogen Blocking<br>FCM 10 µg/ml for blocking<br>1:50 for FCM Cat#: MA5-23918<br>Anti-active Caspase-3 BD Biosciences FCM 1: 50 Cat#: 560901<br>Anti-active Caspase-3 BD Biosciences FCM 1: 50 Cat#: 570179<br>Anti-mouse Ki-67 BioLegend FCM 1: 50 Cat#: 652420<br>Anti-mouse Granzyme B eBioscience FCM 1: 50 Cat#: 12-8898-82<br>Anti-mouse CD8 BioLegend FCM 1: 100 Cat#: 126610<br>Anti-mouse CD3 BioLegend FCM 1: 100 Cat#:100228<br>Anti-mouse CD8 Cell Signaling Technology IF 1: 100 Cat#: 98941<br>Anti-mouse CD45.2 BioLegend FCM 1: 100 Cat#: 109835<br>Anti-mouse CD45.1 BioLegend FCM 1: 100 Cat#: 110735<br>Anti-human MHC-II BioLegend FCM 1: 100 Cat#: 327010<br>Anti-human CD86 BioLegend FCM 1: 100 Cat#: 374204<br>Anti-human CD206 BioLegend FCM 1: 100 Cat#: 321124<br>Anti-human CD163 BioLegend FCM 1: 100 Cat#: 333606<br>Anti-human MHC-II BioLegend FCM 1: 100 Cat#: 980414<br>Anti-mouse MHC-II BioLegend FCM 1: 100 Cat#: 107616<br>Anti-mouse CD86 BioLegend FCM 1: 100 Cat#: 105125<br>Anti-mouse CD206 BioLegend FCM 1: 100 Cat#: 141720<br>Anti-mouse CD163 BioLegend FCM 1: 100 Cat#: 111804<br>Anti-human CD63 Abcam WB 1: 1000 Cat#: ab134045<br>Anti-human CD63 Abcam IF 1: 200 Cat#: ab8219<br>Anti-human LAMP1 Cell Signaling Technology IF 1: 200 Cat#: 15665<br>Anti-Mouse IgG, F(ab')2 fragment specific Jackson ImmunoResearch FCM 1: 100 Cat#: 115-475-072<br>Anti-Mouse IgG (FITC), F(ab')2 fragment specific Jackson ImmunoResearch FCM 1: 100 Cat#: 109-096-006<br>Anti-Mouse IgG (APC), F(ab')2 fragment specific Jackson ImmunoResearch FCM 1: 100 Cat#: 115-136-072<br>Anti-EGFP Cell Signaling Technology WB 1: 1000 Cat#: 2956 |
|---|---|
| Validation | All antibodies were verified by the supplier and each lot has been quality tested. All the antibodies used are from commercial sources and have been validated by the vendors. Validation data are available on the manufacturer's website. |

# Eukaryotic cell lines

Policy information about cell lines and Sex and Gender in Research

| | |
|---|---|
| Cell line source(s) | The murine PDAC cell line 4662 cells were obtained as previously described (PMID: 25979873, PMID: 37115855), Melanoma B16-F10 cells (Cat#: CRL-6475);293T (Cat#: CRL-3216 ) were originally obtained from ATCC. WM9 cells, WM35 cells, the human leukocyte antigen (HLA)-matched cytotoxic T cells to WM35 cells and B16-OVA cells were generated as previously described (PMID: 37805922, PMID: 30089911);  4662-OVA cells were generated as previously described (PMID: 27642636). The cells were passaged for less than 1 month. |
| Authentication | A short tandem repeat DNA profiling method was used to authenticate the cell lines and the results were compared with reference database. |
| Mycoplasma contamination | All cells were regularly tested for Mycoplasma using the Mycoplasma Detection Kit (InvivoGen, Cat#: rep-mys-50) before experiments. No mycoplasma contamination was found. |
| Commonly misidentified lines (See ICLAC register) | No commonly misidentified cell lines were used for this study. |

# Animals and other research organisms

Policy information about studies involving animals; ARRIVE guidelines recommended for reporting animal research, and Sex and Gender in Research

| | |
|---|---|
| Laboratory animals | Wide type C57BL/6 mouse line (Cat#:000664) or transgenic C57BL/6 mouse line, with EGFP cDNA  (Cat#: 003291) were ordered from Jackson laboratory and housed in a specific-pathogen-free animal facility at ambient temperature (22 ± 2 °C), air humidity 40%–70% and 12-h dark/12-h light cycle. |
| Wild animals | No wild animal was used in this study. |
| Reporting on sex | Female mice were used in this study |
| Field-collected samples | No samples were collected in Field. |
| Ethics oversight | All animal experiment protocols were reviewed and approved by the institutional animal care and use committee of the University of Pennsylvania. |

Note that full information on the approval of the study protocol must also be provided in the manuscript.

# Clinical data

Policy information about clinical studies

All manuscripts should comply with the ICMJE guidelines for publication of clinical research and a completed CONSORT checklist must be included with all submissions.

| | |
|---|---|
| Clinical trial registration | N/A |
| Study protocol | N/A |
| Data collection | N/A |
| Outcomes | N/A |

# Plants

| | |
|---|---|
| Seed stocks | N/A |
| Novel plant genotypes | N/A |
| Authentication | N/A |

# Flow Cytometry

## Plots

Confirm that:

☒ The axis labels state the marker and fluorochrome used (e.g. CD4-FITC).

☒ The axis scales are clearly visible. Include numbers along axes only for bottom left plot of group (a 'group' is an analysis of identical markers).

☒ All plots are contour plots with outliers or pseudocolor plots.

☒ A numerical value for number of cells or percentage (with statistics) is provided.

## Methodology

| | |
|---|---|
| Sample preparation | Tumor tissues were dissociated using 1 mg/ml type I collagenase with of 50 U/ml RNase and DNase I at 37°C for 40 min. Digestion mixture was passed through 70 µm cell strainers (FALCON) to obtain single-cell suspension. Red blood cells (RBC) were removed using RBC lysis buffer (BD Biosciences). Dead cells in the single-cell suspension were excluded using Live/Dead Fixable Aqua Dead Cell Stain Kit (Life Technologies). |
| Instrument | BD LSR II |
| Software | FlowJo software package (Flowjo V10) |
| Cell population abundance | Sorting was not used in this study |
| Gating strategy | In general, cells were first gated on FSC/SSC. Singlet cells were gated using  FSC-H and FSC-A. Dead cells were then excluded and further surface or intracellular antigen gating was performed on the live cell population (Extended Data Fig. 5 and Extended Data Fig. 7). |

☒ Tick this box to confirm that a figure exemplifying the gating strategy is provided in the Supplementary Information.

