## [Peer Review file · Nature]

Mucosal vaccination clears *Clostridioides difficile* colonization

Corresponding Author: Professor Dana Lacy

Version 0:

Reviewer comments:

Referee #1

(Remarks to the Author)

In their manuscript "Mucosal vaccination clears *Clostridioides difficile* colonization", Thomas and colleagues described the developing and preclinical testing of a novel preventative vaccine for the nosocomial pathogen *C. difficile*. The authors used both the *C. difficile* toxins TcdA and TcdB as well as surface antigens for both the vegetative and spore form of the pathogen in their vaccine formulations. The non-toxin antigens (NTAs) were chosen based on a previously published paper from another group that was trying to identify NTAs for vaccine formulation). Thomas et al. compared the immunogenicity and efficacy of the vaccine candidates when administered via the mucosal (via rectal instillation [RI]) and parenterally (via intraperitoneal [IP] injection).

As has been demonstrated with previous studies, vaccination using the toxin (in this case, inactivated by the generation of defined mutations in the toxin genes to eliminate the glucosyltransferase activity of TcdA and TcdB as well as disrupting pore formation and receptor binding for TcdB) would protect against experimental infection with toxigenic *C. difficile* with a significant increase in survival after challenge. This was true whether the vaccine was administered via the RI or IP route. Interestingly, administration of a vaccine consisting of just the NTAs without toxin also afforded protection from fatal infection, albeit with lower efficacy than vaccination with toxins alone.

Per this reviewer, the real advance reported in this manuscript concerns the ability of RI of vaccine to result in clearance of the pathogen in addition to protection from disease. Elimination of *C. difficile* shedding in feces was only seen when vaccine formulations were administered via RI, not when administered IP. The authors show that immunization via RI with the NTAs alone could result in decreased shedding of *C. difficile*. When NTAs and toxins were administered via RI even greater reductions in fecal shedding was observed.

The authors went on to characterize and compare the immune responses to vaccines administered by the RI and IP routes. IP administration was associated with robust serum IgG compared to the RI route. There was some variation related to each individual antigen examined. Conversely, RI immunization was shown to induce mucosal Th17 responses, associated with the stimulation of tissue-resident memory cells (TRM) cells. The authors speculate that these TRM cells might be responsible for the observed loss of rectal shedding of *C. difficile* observed in animals that receive vaccine via RI. Additionally, the authors present data that fecal IgG against some of the NTAs used in the vaccine was stimulated by RI (but not IP) and this was correlated with clearance of *C. difficile*. The authors finally followed the durability of the immune responses elicited by the various RI vaccines and found that responses were sustained for at least 2.5 months and this was accompanied by sustained protection from experimental *C. difficile* challenge.

Overall, this manuscript describes an exciting advance in the development of a possible novel vaccine for the prevention of *C. difficile*. Importantly, since this not only protects against disease in the immunized individual, since this is also accompanied by a loss of *C. difficile* shedding, this could have important implications for infection prevention. The authors suggest some potential mechanisms for their findings. There are some questions that arise from reading this interesting work that I list below.

Line 116: I have questions about the choice of NTAs that were chosen for vaccine development. For the three at the bottom of Table S1, were the three unable to be expressed, or was there another reason for not choosing these? Similarly, what was the rationale for choosing the specific 4 NTAs for vaccine development?

Line 131: There was equal survival from RI and IP administration, but only IP protected against weight loss. Was there any difference in toxin titers or histopathology between the two groups? Perhaps not expected, but this could be a secondary effect of vaccine induced responses.

Line 160-163 Figure S4: This is an interesting result. I'm not sure what the statistical support for the correlation between titers and *C. difficile* burden. Perhaps I misunderstood this.

Line 174 and Figure S5E: Interesting for TcdB2 that the GTX and pore mutant alone wasn't enough to eliminate toxicity. Would the authors wish to comment on this with regards to insight into toxin function/activity?

Line 182 and Figure 3B and 3C: This is related in part to my following comment. There appears to be a difference in the controls that received dmLT along via IP and RI routes. There seems to be a slight increase survival in the controls that received only dmLT via RI compared to those that received only dmLT via IP. Do the authors wish to comment on this?

Line 197 and Figure 3: Figure 3F and G shows the results of "naïve" mice that were challenged with *C. difficile* after no immunization at all. A few questions about this. No survival or weight loss data is shown for these naïve mice in Figures 3B-E. Were these similar to the mice that received dmLT alone? Also in Figures 3H and 3K, it appears that there were two mice that received dmLT alone by RI that survived and had decrease in both total CFU and spores. Could dmLT alone have stimulated a non *C. difficile*-specific immune response that allowed a subset of animals to survive? Was this not seen at all in fully naïve mice, or was this variation also present in these animals?

Line 194-5: Is this statement based on when the groups first reached "0% colonization" (as judged by the bar charts? If so, wouldn't the difference be Day 9 and Day 10? Regardless, including toxin in the vaccine preparation seems to have an effect on detection of total CFU early, even if animals would "rebound" shedding on subsequent days. Would the authors comment on this? Also, do they think that if they held the animals longer than 10 days after challenge, there might still be some "rebound"? Not sure this matters, but the data suggest that holding animals for more than 10 days could yield some interesting results, in particular comparing the effects of different antigens, as well the comparison between RI and IP.

Line 206-9: There has been a long debate on the role of TcdA in the in vivo pathogenesis of *C. difficile* (the senior author has participated in some of this previous work). Do these responses to TcdA provide any further insight into the potential role of TcdA in pathogenesis? It might be too complex of a system (with too many variables) but perhaps the authors might have an idea.

Line 273: What was the specificity of these "persistent TRM cells? Was it still CspC?

Discussion Lines 321-340: As I noted above, I found the fact that RI could eliminate shedding of *C. difficile* to be very interesting and perhaps one of the most important findings in the study. The discussion raises some mechanistic explanation for this finding and I have some questions based on these possibilities that are suggested. The idea that internalization of *C. difficile* spores by epithelial cells is a source of the pathogen in patients who experience recurrence. In their study, the authors monitor fecal shedding only. At the time of necropsy, was there any attempt to isolate and culture spores from tissue? As alternate approaches, was there an attempt to detect *C. difficile* by PCR on DNA extracted from intestinal tissue at the time of necropsy? There are experimental ways to try and examine this as well. Figure 3 indicates that by 10 days, animals that received vaccine by RI no longer shed *C. difficile* in their feces. Was there ever an attempt to trigger "recurrence" in these animals by repeat administration of antibiotics? There was also a small number (N=1) in the results presented in Figure 3 of animals that were vaccinated by IP that cleared *C. difficile* fecal shedding. Is this a repeatable observation and could antibiotic-triggered recurrence be seen in IP-vaccinated animals that no longer shed the pathogen in their feces? Another experimental approach would be to treat vaccinated animals (both IP and RI) with vancomycin as part of a treatment/recurrence model and determine if one animal vaccinated via RI had recurrence at a lower rate. On line 335-336, the authors propose that CD8+ TRM can promote clearance of this proposed spore reservoir. They suggest that CspC responses (specific to the spore form of the pathogen) could be responsible. Could this be formally tested by vaccine formulations that included the toxins and the NTAs but varied in the inclusion of CspC? The role of the inclusion of CspC may be a critical feature that at least in part explains the ability of RI vaccination to trigger clearance. This of course has to be balanced by the results in Figure 2 that demonstrate that in addition to CspC, vaccination with FlgG_{EC} and C40 alone can result in the loss of fecal shedding of *C. difficile*. I would ask the authors to try and reconcile all of their results in the discussion of what might underlie the apparent clearance of *C. difficile* as judged by loss of detection of the pathogen in feces.

Thanks for giving me the opportunity to review this important work. I hope that the authors find my comments helpful.

Vincent B. Young

Referee #2

(Remarks to the Author)

The manuscript by Thomas, et al. describes a vaccine against *Clostridium difficile* that, in addition to protect mice from mortality and weight loss, is able to protect also against colonization. The results are important because *C. difficile* colonizes hospitalized patients and causes severe disease and mortality, especially after antibiotic therapy.

Although some approved drugs are available, vaccination so far has not been able to confer protection from colonization. This manuscript uses four approaches to improve the outcome of vaccination: first, they use rectal immunization route (RI). This is a very unusual route of immunization, which in this case is proposed as the innovation that may solve the problem. Second, they use surface antigens to prevent from colonization. Third, they use the intact toxins of the bacterium, which are made nontoxic by mutating key amino acids. Fourth, the selected antigens are then delivered using a non-toxicogenic form of *E. coli* enterotoxin as mucosal adjuvant. They show that when non toxin antigens were administered alone, both RI and intraperitoneal (IP) immunization protected mice from challenge with *C. difficile* RT027. A subset of RI immunized mice showed reduced colonization by nontoxicogenic strain challenge, indicating that these surface antigens contribute directly to limiting bacterial burden. Addition of detoxified TcdA and TcdB improved survival. Rectal immunization also induced mesenteric lymph node memory B cells and mucosal Th17 T cells, and protection persisted for at least 60 days following a prime boost regimen.

Overall, this study is innovative, describes results that were not achieved before, and demonstrates that a multivalent vaccine can elicit protective immunity at the site of *C. difficile* infection.

Specific comments:

Despite significant protection via the rectal route, the IP formulation performs comparably, or in some readouts even better, than RI: non toxin antigens alone significantly prevent weight loss when given IP (Fig. 1), and the complete formulation yields improved survival with IP immunization (Fig. 3). The more rapid decolonization observed with inclusion of surface antigens does support the multivalent approach, but additional experiments are needed to establish clear superiority over conventional parenteral vaccination. Specifically,

1. Figures 1 and 2 show that IP immunization with non toxin antigens protects mice and impacts colonization to a degree similar to RI. Given that serum antibodies can transudate into the gut lumen, the authors should consider passive transfer of sera from immunized mice into naïve recipients to clarify the relative contributions of local mucosal versus systemic antibody responses.
2. In Figure 2, differences in CFUs between IP and RI groups are difficult to appreciate. The authors should include CFU counts at a later timepoint (e.g., day 10 post challenge) with statistical comparisons to demonstrate any advantage of RI in reducing bacterial burden.
3. Figure 5 omits the IP immunization arm. To assess whether RI offers longer lasting protection, the authors should compare the durability of immunity (e.g., survival, weight change, CFUs) following both RI and IP routes over the extended period following immunization.

Referee #3

(Remarks to the Author)

Nature review

As a peer reviewer, you will be asked to provide an assessment of the following aspects of the manuscript (where relevant, and not necessarily in this order):

1. Key results. Please summarize what you consider to be the outstanding features of the work.
2. Validity. Does the manuscript have flaws that should prohibit its publication? If so, please provide details.
3. Originality and significance. If the conclusions are not original, please provide relevant references. On a more subjective note, do you feel that the results presented are of immediate interest to many people in your own discipline, or to people from several disciplines?
4. Data & methodology. Is the approach valid? Are the data and presentation of good quality? Please note that we expect our reviewers to review all data, including the Supplementary Information.
5. Appropriate use of statistics and treatment of uncertainties (if applicable). All error bars should be defined in the corresponding figure legends. Please include in your report a specific comment on the appropriateness of any statistical tests, and the accuracy of the description of any error bars and probability values.
6. Conclusions. Are the conclusions and data interpretation robust, valid, appropriate and reliable?
7. Suggested improvements. Please list additional experiments or data that could help strengthen the work in a revision.
8. References. Does the manuscript reference previous literature appropriately?
9. Clarity and context. Is the abstract clear and accessible? Are the abstract and introduction appropriate?
10. Please indicate any particular aspect of the manuscript, data or analyses that you feel is outside the scope of your expertise, or that you were unable to assess fully.

Key results. In this manuscript, the authors have compared mucosal and parental immunization of mice with a novel vaccine containing cleverly designed antigens to clear and prevent infections with *C. diff*. Their strategy targeted several key elements in vaccine design and they measured several useful parameters as correlates of immunity including pathogen clearance. That said, some of the supplemental data are rather descriptive and not necessarily informative. The experiments reflected impressive expertise in some areas but maybe not in others – or the limitations of space may have precluded them from describing issues I sought to understand.

Perhaps the focus should be on the impact of antigen composition and route of immunization rather than including a superficial description of possible effector mechanisms.

Validity. Experiments were repeated only twice and Some potentially interesting results were not validated.

For flow cytometry, I didn't see any mention of isotype controls or, preferably, FMO.

Some of the methods for antigen prep seemed scant and/or lacked sufficient citations.

While IgG may be associated with clearance, no intervention strategy was employed to show cause and effect directly.

For the histopathology, two scorers should be used to validate findings and ideally, one is a veterinary pathologist or they should refer to a previously published paper that describes their scoring criteria and validation. In their images, e.g. SFig 8, the colonic muscularis varies from 10 to 80 microns. That is unusual and may be due to mounting artifacts.

For IEL and LPL isolation, they should refer to methods reported previously that validated the purity of IEL vs LPL as cross contamination is likely, e.g. enterocytes with IEL; B cells mixed into the IEL, etc. For example, due to cells lost during isolation, flow cytometry cannot measure "total" cells, it only assesses percentages. That would require validation with IHC/IF. Further, IHC/IF would also validate the percentage B cells in IEL which appear higher than expected. If they haven't done it before, the validation could be mentioned here. That would enhance the strength of conclusions for data in figure 4 for example. There is no evidence that the apparent differences in IEL or LPL have any impact on immunity induced by the vaccine. The paper could likely be streamlined by paring the descriptive data unless it at least relates to a question many people might be interested in learning about.

Originality and significance. The approach to develop the antigen appeared to be very original and well thought out. Such an approach could have real impact on this disease. The local immunization was also helpful although I remain skeptical of how many subjects would want a rectal immunization.

Data and Methodology including statistics. The experimental design was marginal. For example, it appears only 6-week-old mice were used which does not determine any age effects as required by NIH guidelines. It is unclear if multiple experiments were done to assess inter-experimental variation in all figures. This was mentioned in some figure legends but not all. In some cases, N was 5-15 but it is unclear why they used such a large range or if some responders were deleted from the study. They offer that the variation was depending on the ability to collect samples but no further explanation was evident. As best I could tell, statistics were adequately described but it's hard to determine if they are appropriate if there is no mention of the normal distribution of the variables.

The selection of antigens seems to have been very well thought out.

For the rectal immunization, no details are provided on the colon prep. It wasn't clear why they chose rectal immunization vs oral. Rectal immunization would be a challenging approach to sell in human medicine. That said, adding it to a routine colonoscopy would be logical although that was not the administrative route they tested.

Figure 3C didn't seem to have all three cohorts represented in the figure. There is a symbol that might be suggesting the lines overlap but it should be more clearly specified.

Novel surface antigens

Mutated toxin antigens that induce coverage with broad epitope recognition

E coli adjuvant

Rectal route vs parental

Humoral and cellular correlates of immunity.

Conclusions.

The authors conclude that mucosal immunization is a promising strategy for the prevention of symptoms and clearance of C diff. Their core data support this through the use of an innovative vaccine.

Suggested improvements.

Most are listed above but focusing the manuscript on proven elements, e.g. the functional vaccine and its route of administration vs. vague and indirect correlates of immunity. 23 figures with sometimes over 30 panels reflects today's trend but cannot be meaningfully described in such brief legends and text making them quite superfluous.

Validating key findings.

Expanding the age cohorts. Adding 8 12-16 week old mice per group would give another experimental replicate and address variations by age that can be tracked by ANOVA.

References. The authors know the C diff field much better than I do so I would assume the citations are reasonable for the most part. Some methodological techniques could be cited more thoroughly.

Clarify and Context.

Minor points:

"Filletted open" is not the correct way to describe whatever it is they were doing.

I don't understand if declaring no conflict of interest is consistent with two authors filing a patent based on the data.

Version 1:

Reviewer comments:

Referee #1

(Remarks to the Author)

The revised manuscript, "Mucosal vaccination clears *Clostridioides difficile* colonization" by Thomas et al. (2025-06-14298) describes a novel vaccine formulation and a comparison of the results of vaccination via parenteral (intraperitoneal injection) and mucosal (intrarectal installation) administration. I would like to thank the authors for the additional experimentation designed to address the concerns of the three reviewers who read the initial version of the manuscript. In my opinion, the

authors have done a very good job in addressing all three reviewer's comments. I will focus my comments here on the response to my comments on the first manuscript.

One of my major comments concerned the fact that rectal instillation (RI) of the complete vaccine (including the toxins, the non-toxin antigens and the spore antigen CspC) appeared to result in elimination of *C. difficile* from the gastrointestinal tract while intraperitoneal (IP) administration did not. In the initial version, this was judged by cultivation of feces from immunized animals. I had suggested that examination of tissues at necropsy would be a possible way to further demonstrate that RI does result in the generation of sterilizing immunity. The authors have done this work and using a combination of cultivation and molecular analysis (PCR) they are unable to detect *C. difficile* in the tissues of mice immunized by the RI route, while the pathogen was still detected in tissues of mice immunized by the IP route. While it is obviously difficult to "prove the negative" I think that the additional data provide robust evidence that RI does lead to elimination of *C. difficile* from immunized animals.

Further evidence of the efficacy and sustained protection afforded by RI administration of their novel vaccine is provided by new data that extends the initial period of observation of challenge from 10 to 15 days, demonstrating that there was no rebound of *C. difficile* with a longer period of observation. As an alternate approach to demonstrate the elimination of *C. difficile* in the animals immunized by the RI route, the authors conducted additional experiments whereby they treated animals with vancomycin during the initial *C. difficile* infection and then monitored for recurrence after the discontinuation of the vancomycin. Mice that received the toxin-only vaccine did exhibit recurrence (and mortality) after stopping vancomycin while the animals that received the vaccine containing toxin and non-toxin/CspC antigens did not have a reappearance of *C. difficile* in their feces and did not exhibit any clinical signs of CDI. These additional data greatly strengthen the conclusion that the novel vaccine coupled with mucosal administration represents an advance in the development of a viable *C. difficile* vaccine.

Finally, the authors conducted additional experiments to examine the contributions of the non-toxin antigens FlgGEK, C40 and the spore antigen CspC to the observed complete clearance of *C. difficile* compared to the toxin-only formulation. The results are complex, but the authors do provide evidence that provides insight into the relative contributions of the various non-toxin antigens in their vaccine to the elimination of vegetative cells and spores. Furthermore, they do demonstrate that CspC provides a significant amount of protection against serious/fatal infection in animals.

Overall, I think that the authors have done an excellent job in addressing the previous questions and concerns from the initial review. I do have clarifying questions concerning the versions of inactivated TcdB in the vaccine formulation. The authors note that the residual toxin activity was due to active TcdB2 toxin contaminating the purification column. Was this due to wild type (i.e. no mutations whatsoever) TcdB2? Furthermore, are there still results being reported from using the TcdB2 that contains the L1598A mutation? In lines 149-158 of the revised manuscript, the L1598A mutation is no longer mentioned. Were all of the experiments using this version of the toxin removed from the revised manuscript? I was unable to determine if this is true.

Thank you for giving me the opportunity to revisit this interesting manuscript.
Vincent B. Young

Referee #2

(Remarks to the Author)

The authors addressed the most important questions. The manuscript is suitable for publication.

Referee #3

(Remarks to the Author)

Summary

This manuscript is a resubmission. The authors have exhaustively addressed many comments from the three reviewers and editor. The focus of the title is about the clearance of *C. diff* with mucosal immunization. Given their methods, perhaps they should state rectal immunization in the title and elsewhere instead of what appears to be a conscious attempt to avoid it. I am still concerned that they limited their studies to one age group (young mice) to model a disease that more often affects the elderly. The correlates of immunity remain superficially studied. I would suggest that more focus would help. Compare an older cohort. Focus on IgA and IgG and forget about the T cell studies for now.

Originality and significance

I like the approach to design the antigenic composition of the vaccine and route of immunization. I find these original and significant.

Data and Methods and suggestions for improvement

It is perplexing and I couldn't find a clear explanation, why IP immunization is so immunogenic yet ineffective, particularly since they make a case for IgG being important.

I still think that a single age group ignores the important potential effects of age. 6 weeks is quite a young age for mice (sexually mature but not adult). In addition, they could still have residual passive immunity to who knows which antigens that

could impact colonization, immune responses etc. In contrast, the disease they are trying to prevent most commonly affects the elderly who often have waning immune systems. I do not agree that this age is sufficiently informative to guide future studies in humans. I believe that animal models should attempt to provide a parallel context to the human condition they are studying. This is exactly why the NIH requires all investigators to consider the impact of sex and age on the outcomes they measure in animal models. If this premise is important and correct, then one age group is inadequate, particularly if it is not the age group most affected by the disease one is modeling. It is possible that what they find in this age group has no bearing on most of the humans who get the disease. Perhaps a challenge of the mice 3, 6, 9 months after immunization would suffice. Perhaps immunizing mice at 6, 12, and 24 weeks of age would be better. As they performed the experiments, it is entirely possible that the benefit of this vaccine would not be reproduced in mature subjects receiving a colonoscopy at which time they could be immunized.

They note that rectal immunization was chosen as intranasal is effective for respiratory pathogens, perhaps due to its proximity to the site of infection. Part of the success of intranasal infection is the ability to induce local IgA. Although not tested directly, they feel that IgA did not seem to be a significant mediator of immunity in this study. I feel that rigorously showing that IgG is more important while IgA is not would be a detail that is worth investigating more thoroughly. Many of the responses are associations. While the transfer of IgG conferred some protection, that implies it is sufficient, not necessary. Further, how is it working? Is it secreted into the lumen where the infection is? If so how? Did they demonstrate luminal IgG that wasn't part of contamination with mucosal tissue IgG preps? If so is it protected from digestion? Do they have images of IgG bound to *C. diff*?

Supplemental Fig 7 says many things. It appears IP immunization induced more IgG and IgA yet it wasn't as effective in protection as rectal immunization. I wasn't able to reconcile these observations and I apologize if I missed something. Why wouldn't they also purify and test mucosal IgA, at least as a foil? As reported, it is an incomplete/uncontrolled experiment based on the assumption that IP immunization induced IgA but they weren't protective therefore IgA is meaningless. IP induced IgA may not be protective for many reasons. Perhaps it's monomeric, or the specificity varies due to the site of induction and the variations in certain antigen specific B cell precursor frequencies. Some studies, e.g. <https://doi.org/10.1016/j.micinf.2016.05.001>, suggest fecal IgA is associated with reduced *C. diff*. "Dogma" suggests only 4 or 5% of the B cells in the large intestine produce IgG, thus, ignoring the IgA limits the impact, and the credibility, of the study. There are limited data on the specificity of these antibodies, their origin from serum vs. mucosal B cells. Maybe the IgG is key to reducing *C. diff* colonization, but they didn't really explore this very rigorously. Please don't get me wrong, while I recognize the importance of IgA, it is also true that IgA deficiency isn't as detrimental as one might predict. I also like new theories, but I like them better when they are evidence-based.

Similarly, their investigation of T cell function is purely associative. That said, their findings are quite interesting and merit further investigation.

The differences in pathology appear subtle. Do they have any independent objective correlates that support agreement? Wt loss vs score? TNF less vs. score.

Statistics

The figures are busy. 4 * isn't more impressive than 1. Define the level of significance and if that is 0.05, then indicate it with a single *. 0.00005 means nothing vs. 0.04 if you have defined significance as <0.05, particularly when N is sufficient, but small and different (2-10; 3. Etc). They show individual data points and while unclear, some visually appear to be nonparametric. If the response is robust, their replicate experiments should be included but it was unclear if they were in all cases.

Conclusions

Not all are justified. The mechanisms of protection are studied superficially and since the paper is broad, they are inadequately discussed critically.

References selective but OK

Clarity. Clearly written

Version 2:

Reviewer comments:

Referee #3

(Remarks to the Author)

The responses by the authors are acceptable.

Response to Referees for Nature Submission 2025-06-14298

We would like to thank the Referees and Editor for their insightful comments and suggestions. In response, we have performed additional experiments to provide more mechanistic insights and revised the text, figures, and methods to provide additional clarity and detail. Our revised manuscript includes the following major improvements:

1. Expansion on the achievement of sterilizing immunity by rectal instillation of the vaccine formula when compared to intraperitoneal injection. To further assess this, we monitored vaccinated animals for fifteen days post-challenge to ensure a lack of “rebounding” infection (**new Fig. 2f-q**), macerated colons and ceca from vaccinated animals and plated for hidden *C. difficile* reservoirs (**new SFig. 7a-d**), performed PCR (**new SFig. 7e**) and qPCR (**new SFig. 7f,g**) on the same tissues to test for *C. difficile* undetectable by plating, and tested a recurrent infection model (see Point #2 below; **new Fig. 3**). Crucially, our experiments demonstrate the complete clearance of *C. difficile* and indicate sterilizing immunity imbued by rectal instillation of the toxin and NTA formula.
2. Introduction of a recurrent *C. difficile* infection model that further demonstrates the ability of this vaccine to provide sterilizing immunity to prevent recurrence, as well as protect against death and weight loss (**new Fig. 3**).
3. Passive transfer of fecal IgG from fully-vaccinated, rectally-instilled mice into naïve animals to highlight the role of anti-vegetative bacterial fecal IgG in *C. difficile* clearance (**new Ext. Fig. 2**). The effects of fecal IgG against vegetative bacteria to reduce bacterial motility were further verified *in vitro*.
4. Characterization of the role of CspC in vaccine-induced spore clearance (**new Ext. Fig. 3**). Importantly, inclusion of CspC in the vaccine is essential to clear *C. difficile* vegetative cells and spores from the colon. In addition, immunization with CspC and the inactivated toxins alone is sufficient to clear *C. difficile* spores from the host, albeit on a slower timeframe than the formula that includes the vegetative antigens.
5. Validation of RI- and IP-mediated protective and clearance effects in an older cohort of mice. To test whether the age of the mice represented an experimental variable, we repeated the immunization and challenge experiments with an older cohort of mice that were 12-weeks old at the time of the initial immunization. (In our original Figure 3, all animals were 5-weeks old at the time of the initial immunization.) Notably, we did not discern differences in phenotypes between younger and older mice by ANOVA analysis. All rectally-instilled mice given the toxin and NTA formula cleared the bacteria and were protected against acute infection at both ages. The combined results are represented in the **new Figure 2**.
6. Further demonstration of vaccine response durability at 200 days post-boost, in addition to the 60 days post-boost data that were originally reported (**new Fig. 5**). Importantly, protection against, and clearance of, lethal *C. difficile* infection holds at this elongated timepoint.

In addition to these exciting new data, we have addressed each of the Referees’ comments and concerns in a comprehensive point-by-point response below. We feel the manuscript is greatly improved thanks to the insightful comments from the Referees and look forward to your further consideration.

Referee #1:

In their manuscript “Mucosal vaccination clears Clostridioides difficile colonization”, Thomas and colleagues described the developing and preclinical testing of a novel preventative vaccine for the nosocomial pathogen C. difficile. The authors used both the C. difficile toxins TcdA and TcdB as well as surface antigens for both the vegetative and spore form of the pathogen in their vaccine formulations. The non-toxin antigens (NTAs) were chosen based on a previously published paper from another group that was trying to identify NTAs for vaccine formulation). Thomas et al. compared the immunogenicity and efficacy of the vaccine candidates when administered via the mucosal (via rectal instillation [RI]) and parenterally (via intraperitoneal [IP] injection).

As has been demonstrated with previous studies, vaccination using the toxin (in this case, inactivated by the generation of defined mutations in the toxin genes to eliminate the glucosyltransferase activity of TcdA and TcdB as well as disrupting pore formation and receptor binding for TcdB) would protect against experimental infection with toxigenic C. difficile with a significant increase in survival after challenge. This was true whether the vaccine was administered via the RI or IP route. Interestingly, administration of a vaccine consisting of just the NTAs without toxin also afforded protection from fatal infection, albeit with lower efficacy than vaccination with toxins alone.

Per this reviewer, the real advance reported in this manuscript concerns the ability of RI of vaccine to result in clearance of the pathogen in addition to protection from disease. Elimination of C. difficile shedding in feces was only seen when vaccine formulations were administered via RI, not when administered IP. The authors show that immunization via RI with the NTAs alone could result in decreased shedding of C. difficile. When NTAs and toxins were administered via RI even greater reductions in fecal shedding was observed.

The authors went on to characterize and compare the immune responses to vaccines administered by the RI and IP routes. IP administration was associated with robust serum IgG compared to the RI route. There was some variation related to each individual antigen examined. Conversely, RI immunization was shown to induce mucosal Th17 responses, associated with the stimulation of tissue-resident memory cells (TRM) cells. The authors speculate that these TRM cells might be responsible for the observed loss of rectal shedding of C. difficile observed in animals that receive vaccine via RI. Additionally, the authors present data that fecal IgG against some of the NTAs used in the vaccine was stimulated by RI (but not IP) and this was correlated with clearance of C. difficile. The authors finally followed the durability of the immune responses elicited by the various RI vaccines and found that responses were sustained for at least 2.5 months and this was accompanied by sustained protection from experimental C. difficile challenge.

Overall, this manuscript describes an exciting advance in the development of a possible novel vaccine for the prevention of C. difficile. Importantly, since this not only protects against disease in the immunized individual, since this is also accompanied by a loss of C. difficile shedding, this could have important implications for infection prevention. The authors suggest some potential mechanisms for their findings. There are some questions that arise from reading this interesting work that I list below.

Author’s Response:

We thank the Referee for the thoughtful overview and insightful suggestions. We have provided specific responses to each concern below; briefly, we have experimentally addressed key concerns about sterilizing immunity, relapse models, and the role of CspC in our revised manuscript. In addition, we have elaborated on points that were previously unclear.

Line 116: I have questions about the choice of NTAs that were chosen for vaccine development. For the three at the bottom of Table SI, were the three unable to be expressed, or was there another reason for not choosing these? Similarly, what was the rationale for choosing the specific 4 NTAs for vaccine development?

Author’s Response:

The three NTAs at the bottom of Table SI were unable to be expressed, and that is why we did not move forward with them. We have added clarifying language to the Table SI legend to reflect this: “Y, yes,

able to express; N, no, unable to express”. As for the rationale for choosing the specific NTAs for vaccine development, we selected them based on overall yield and solubility of the protein and/or complex, as well as whether they were predicted to be on vegetative cells or spores, as we wanted to incorporate antigens that targeted both forms of *C. difficile*. We have modified the text as follows: “We recombinantly expressed and purified thirteen of these proteins and selected several for antigenicity testing based on the overall yield and solubility of each protein or protein complex, as well as prioritizing antigens on both vegetative and spore forms of *C. difficile* (**STable I**).”

Line 131: There was equal survival from RI and IP administration, but only IP protected against weight loss. Was there any difference in toxin titers or histopathology between the two groups? Perhaps not expected, but this could be a secondary effect of vaccine induced responses.

Author’s Response:

For initial cocktail testing of the NTAs by RI and IP, we did not test for differences in toxin titers nor histopathology. That said, the disparity in weight loss between cocktail-vaccinated groups could certainly be a result of vaccine-induced secondary effects.

Line 160-163 Figure S4: This is an interesting result. I’m not sure what the statistical support for the correlation between titers and C. difficile burden. Perhaps I misunderstood this.

Author’s Response:

Thank you for pointing out this discrepancy to us. We have since amended Figure S4 to include the P values (**new SFig. 3b**) associated with the Pearson’s R values (**new SFig. 3a**) in our heatmaps correlating colonization burden reduction and humoral responses.

Line 174 and Figure S5E: Interesting for TcdB2 that the GTX and pore mutant alone wasn't enough to eliminate toxicity. Would the authors wish to comment on this with regards to insight into toxin function/activity?

Author’s Response:

We are very grateful for this question as it was surprising to us, as well. It prompted us to go back and re-sequence and reevaluate our stocks and, in so doing, we found two errors that we have now corrected in the revised manuscript. The first is that the protein we described as TcdB2 (D286N/D288N, L1106K, L1598A) was in fact TcdB2 (D286N/D288N, L1106K). The second is that the TcdB2 (D286N/D288N, L1106K) protein is completely non-toxic. We have tracked the unexpected cell rounding phenotype of the prior preparation to column contamination with active TcdB2 toxin. This is unfortunate as it means we could have done all experiments with TcdB2 (D286N/D288N, L1106K) from the beginning. Fortunately, we have the chance to correct this now (**new SFig. 4**), and none of our experimental results are impacted.

Line 182 and Figure 3B and 3C: This is related in part to my following comment. There appears to be a difference in the controls that received dmLT along via IP and RI routes. There seems to be a slight increase survival in the controls that received only dmLT via RI compared to those that received only dmLT via IP. Do the authors wish to comment on this?

Author’s Response:

We believe that this difference is due to dmLT-induced effects at the local mucosal tissue, which may be partially protective against *C. difficile* infection. Previous studies of dmLT-adjuvanted vaccines have noted that mucosal administration of the adjuvant alone induces a Th1/Th17 response (PMID: 34054818, PMID: 25786687, PMID: 38594380), and data in this manuscript and others (PMID: 31990686, PMID: 33253683) suggest Th17 responses may mitigate infection (although other Th17-*C. difficile* interplay data is paradoxical, see PMID: 31003940). As such, it is logical to believe that even RI vaccinated dmLT and its Th1/Th17 responses may be sufficient to protect some mice against lethality. Because these explanations are speculative and the effect is modest, we have decided not to comment on them in the manuscript.

Line 197 and Figure 3: Figure 3F and G shows the results of “naïve” mice that were challenged with C. difficile after no immunization at all. A few questions about this. No survival or weight loss data is shown for these naïve mice in Figures 3B-E. Were these similar to the mice that received dmLT along?

Author’s Response:

We apologize for the lack of clarity. The experiments to evaluate the tissue were performed in a separate cohort of mice where all animals were sacrificed on day 3 post infection. To better distinguish this as a separate experiment, we have removed these panels from new Figure 2 and discuss them only in the context of SFig. 6. The weight loss observed in the naïve animals was very similar to what was observed in the dmLT only controls, but we have focused the figure only on the histology in the interest of conserving space.

Also in Figures 3H and 3K, I appears that there were two mice that received dmLT alone by RI that survived and had decrease in both total CFU and spores. Could dmLT alone have stimulated a non C. difficile-specific immune response that allowed a subset of animals to survive? Was this not seen at all in fully naïve mice, or was this variation also present in these animals?

Author’s Response:

Very rarely do our naïve mice survive WT *C. difficile* R20291 infection past day three; but on occasion, they may survive due to a higher weight at the beginning of infection (PMID: 35176123) or previously encountering *C. difficile*. As noted above in our response to the Referee’s Line 182 comment above, we believe that this difference could reflect dmLT-induced Th1/Th17 effects at the local mucosal tissue, which may be partially protective against *C. difficile* infection (PMID: 31990686, PMID: 33253683).

Line 194-5: Is this statement based on when the groups first reached “0% colonization” (as judged by the bar charts? If so, wouldn’t the difference be Day 9 and Day 10? Regardless, including toxin in the vaccine preparation seems to have an effect on detection of total CFU early, even if animals would “rebound” shedding on subsequent days. Would the authors comment on this? Also, do they think that if they held the animals longer than 10 days after challenge, there might still be some “rebound”? Not sure this matters, but the data suggest that holding animals for more than 10 days could yield some interesting results, in particular comparing the effects of different antigens, as well the comparison between RI and IP.

Author’s Response:

We appreciate the suggestion to hold vaccinated animals longer than ten days post-infection. We have repeated the experiment from the original Figure 3 (**new Fig. 2a**) with ten additional mice per group and have monitored these mice for fifteen days post-infection. No differences were discerned in survival (**new Fig. 2b,c**) and weight loss (**new Fig. 2d,e**) between the previous and new experiments. Similar patterns of bacterial burden were also noted (**new Fig. 2f-q**). For mice RI-vaccinated with the toxin + NTA formula, there were no rebounding effects of total and spore CFU burden past 9 and 8 days, respectively (**new Fig. 2h,k**). As noted, mice vaccinated by RI with the toxin formula had a significant reduction of CFU burden compared to dmLT-only vaccinated controls (**new Fig. 2f,g,i,j**). However, while most of these mice cleared spores around day nine post-infection (**new Fig. 2j**), they had oscillating vegetative bacterial burdens for fifteen days after infection (**new Fig. 2g**).

Line 206-9: There has been a long debate on the role of TcdA in the in vivo pathogenesis of C. difficile (the senior author has participated in some of this previous work). Do these responses to TcdA provide any further insight into the potential role of TcdA in pathogenesis? It might be too complex of a system (with too many variables) but perhaps the authors might have an idea.

Author’s Response:

At this point we can only speculate, but prior publications do indicate a role for TcdA in contributing to the overall severity of *C. difficile* infection in mice (PMID: 20844489; PMID: 19252482; PMID: 35176123). We plan to conduct future studies where we examine the impact of the TcdA antigen on the overall immune response and vaccine efficacy.

Line 273: What was the specificity of these “persistent TRM cells? Was it still CspC?

Author’s Response:

Unfortunately, we do not have tetramers against the toxins nor NTAs, so determining T_{RM} specificity by flow cytometry was not possible. However, given our results in Fig. 4t-v and SFig. 10 demonstrating CspC-specific cytokine responses when colon tissue CD4⁺ and CD8⁺ cells are incubated with CspC-primed dendritic cells, we infer that these are CspC-specific T_{RM} cells.

Discussion Lines 321-340: As I noted above, I found the fact that RI could eliminate shedding of C. difficile to be very interesting and perhaps one of the most important findings in the study. The discussion raises some mechanistic explanation for this finding and I have some questions based on these possibilities that are suggested. The idea that internalization of C. difficile spores by epithelial cells is a source of the pathogen in patients who experience recurrence. In their study, the authors monitor fecal shedding only. At the time of necropsy, was there any attempt to isolate and culture spores from tissue? As alternate approaches, was there an attempt to detect C. difficile by PCR on DNA extracted from intestinal tissue at the time of necropsy? There are experimental ways to try and examine this as well.

Author’s Response:

We agree that the finding that RI clears *C. difficile* infection is the most important finding of our study. We originally monitored fecal shedding but have now included experiments to elaborate on these findings as suggested. We repeated the experiment in the original Figure 3 (**new Fig. 2**), and at day fifteen post-infection (experimental endpoint), we harvested ceca and colons during necropsy to isolate and culture *C. difficile* from the tissue. No *C. difficile* spores nor vegetative bacteria were isolated from macerated ceca and colons of RI mice given the toxin and NTA formula (**new SFig. 7a-d**). However, *C. difficile* was isolated from IP mice, and from some RI mice given the toxin-only formula.

To further validate these findings, which implicate rectal instillation in achieving sterilizing immunity against *C. difficile*, we performed both PCR and qPCR on DNA extracted from the same macerated tissues to probe for a validated 16S rRNA gene sequence specific to *C. difficile*, thereby increasing our sensitivity in detecting the bacterium. By PCR, we qualitatively verified the lack of the 16S product (175 bp) in the colons and ceca of mice that had received the toxin and NTA formula by RI (**new SFig. 7e**). Distinct PCR product bands were visualized in mice that had been vaccinated by IP, as well as some mice vaccinated by RI with the toxin-only formula. To even further clarify the potential elimination of the pathogen from these mice, we performed TaqMan-based qPCR using a probe within the 16S amplicon region in the colons and ceca from vaccinated and challenged mice. Notably, there were no significant differences in cycle threshold nor normalized reporter values between RI-vaccinated ceca and colons and the negative control, where nuclease-free water was used instead of template DNA (no-template control) (**new SFig. 7f,g**). The normalized reporter values and cycle threshold in the colonic DNA isolated from mice RI-vaccinated with the toxin-only formula were also not significantly different from the negative control. However, dmLT-only vaccinated mice by either route of administration, as well as IP-vaccinated mice regardless of formula, had significantly lower cycle thresholds and higher normalized reporter values compared to the negative control (**new SFig. 7f,g**). Taken together, the organ plating, PCR, and qPCR results extend the original findings obtained by fecal plating and indicate the achievement of sterilizing immunity against *C. difficile* in mice vaccinated by RI with the toxin and NTA formula.

Figure 3 indicates that by 10 days, animals that received vaccine by RI no longer shed C. difficile in their feces. Was there ever an attempt to trigger “recurrence” in these animals by repeat administration of antibiotics? There was also a small number (N=1) in the results presented in Figure 3 of animals that were vaccinated by IP that cleared C. difficile fecal shedding. Is this a repeatable observation and could antibiotic-triggered recurrence be seen in IP-vaccinated animals that no longer shed the pathogen in their feces? Another experimental approach would be to treat vaccinated animals (both IP and RI) with vancomycin as part of a treatment/recurrence model and determine if one animals vaccinated via RI had recurrence at a lower rate.

Author’s Response:

We appreciate the suggestion to test the vaccine in a murine model of *C. difficile* recurrence. We did not observe other IP-vaccinated mice clearing *C. difficile* burden upon repeating the original Figure 3 experiments (**new Fig. 2n,q**), suggesting that the original animal that had previously cleared the bacteria was an outlier. As such, we did not perform the recurrence model with IP-vaccinated mice. After RI of the vaccine formulas, we challenged mice as we had previously and then administered 0.5 mg/mL vancomycin *ad libitum* in drinking water for ten days (from days two to twelve post-infection) to treat *C. difficile* infection and trigger relapse after the antibiotics were removed (PMID: 23147742) (**new Fig. 3a**). Mice administered with the toxin-only and toxin and NTA formulas were significantly protected against death (**new Fig. 3b**), weight loss (**new Fig. 3c**), and diarrhea (**new Fig. 3d**) during reinfection. That said, mice that were immunized against the inactivated toxins only were still susceptible to relapse and only protected against early effects of recurrence; these animals all succumbed by day 17 post-infection (five days after being taken off vancomycin treatment). The mice given the toxin and NTA formula also fully cleared primary infection while on vancomycin and never relapsed in terms of total CFU burden and spore counts (**new Fig. 3i,j**). The prevention of recurrence is even more striking when compared to the bacterial burdens of the RI controls (**new Fig. 3e-h**), which rebloom one day after being taken off vancomycin (day thirteen post-infection). Combined, these data suggest the ability of the toxin and NTA formula to imbue sterilizing immunity in the host to completely clear *C. difficile* and any potential hidden spore reservoirs.

On line 335-336, the authors propose that CD8+ TRM can promote clearance of this proposed spore reservoir. They suggest that CspC responses (specific to the spore form of the pathogen) could be responsible. Could this be formally tested by vaccine formulations that included the toxins and the NTAs but varied in the inclusion of CspC? The role of the inclusion of CspC may be a critical feature that at least in part explains the ability of RI vaccination to trigger clearance. This of course has to be balanced by the results in Figure 2 that demonstrate that in addition to CspC, vaccination with FlgGEK and C40 alone can result in the loss of fecal shedding of C. difficile. I would ask the authors to try and reconcile all of their results in the discussion of what might underlie the apparent clearance of C. difficile as judged by loss of detection of the pathogen in feces.

Author's Response:

Thank you for this suggestion. To determine if the inclusion of CspC in the vaccine formula was necessary for the clearance of spores, we RI-vaccinated mice with formulas that either contained the inactivated toxins and CspC (without the vegetative antigens FlgGEK and C40 peptidase 2) or the inactivated toxins, FlgGEK, and C40 peptidase 2 (without CspC). Vaccination of the toxins and CspC formula protected mice against death (**new Ext. Fig. 3a**) and weight loss (**new Ext. Fig. 3b,c**) as effectively as the original toxin and NTA formula. These mice were able to clear *C. difficile* spore burden by 10 days post-infection (**new Ext. Fig. 3f**) and had oscillating clearance of vegetative bacteria (**new Ext. Fig. 3d**), similar to the toxin-only formula administered by RI (**new Fig. 2i,l**). Conversely, vaccination of the formula without CspC had a 40% mortality rate (**new Ext. Fig. 3a**) and significant weight loss (**new Ext. Fig. 3b,c**) compared to the other formulas tested. Mice vaccinated without CspC had extremely high total CFU burden (**new Ext. Fig. 3e**) and varied spore clearance (**new Ext. Fig. 3g**). Altogether, these results imply that the inactivated toxins, with and without CspC, are sufficient to clear spore burden but are unable to clear vegetative bacteria unless co-administered with the vegetative antigens FlgGEK and C40 peptidase 2.

Thanks for giving me the opportunity to review this important work. I hope that the authors find my comments helpful.

Vincent B. Young

Author's Response:

We greatly appreciate your helpful suggestions and insights. We believe the changes implemented bolster our original findings and strengthen the overall quality of this manuscript.

Referee #2:

The manuscript by Thomas, et al. describes a vaccine against Clostridium difficile that, in addition to protect mice from mortality and weight loss, is able to protect also against colonization. The results are important because C. difficile colonizes hospitalized patients and causes severe disease and mortality, especially after antibiotic therapy.

Although some approved drugs are available, vaccination so far has not been able to confer protection from colonization. This manuscript uses four approaches to improve the outcome of vaccination: first, they use rectal immunization route (RI). This is a very unusual route of immunization, which in this case is proposed as the innovation that may solve the problem. Second, they use surface antigens to prevent from colonization. Third, they use the intact toxins of the bacterium, which are made nontoxic by mutating key amino acids. Fourth, the selected antigens are then delivered using a non-toxigenic form of E. coli enterotoxin as mucosal adjuvant. They show that when non toxin antigens were administered alone, both RI and intraperitoneal (IP) immunization protected mice from challenge with C. difficile RT027. A subset of RI immunized mice showed reduced colonization by nontoxigenic strain challenge, indicating that these surface antigens contribute directly to limiting bacterial burden. Addition of detoxified TcdA and TcdB improved survival. Rectal immunization also induced mesenteric lymph node memory B cells and mucosal Th17 T cells, and protection persisted for at least 60 days following a prime boost regimen.

Overall, this study is innovative, describes results that were not achieved before, and demonstrates that a multivalent vaccine can elicit protective immunity at the site of C. difficile infection.

Author's Response:

We thank the Referee for their overview, kind praise, and discerning recommendations. We have provided specific responses to each concern below; briefly, we have experimentally addressed key concerns about the superiority of RI over IP in achieving sterilizing immunity and passive transfer of fecal IgG to clarify contributions of local versus systemic immune responses in our revised manuscript. We believe these changes have significantly strengthened our manuscript.

Specific Comments:

Despite significant protection via the rectal route, the IP formulation performs comparably, or in some readouts even better, than RI: non toxin antigens alone significantly prevent weight loss when given IP (Fig. 1), and the complete formulation yields improved survival with IP immunization (Fig. 3). The more rapid decolonization observed with inclusion of surface antigens does support the multivalent approach, but additional experiments are needed to establish clear superiority over conventional parenteral vaccination.

Author's Response:

While there have been many pre-clinical studies showing that parenteral administration of toxins, toxin fragments, or toxoids can confer protection against *C. difficile* infection-induced weight loss and death, the translation of these strategies into human clinical trials has been unsuccessful to date.

Our goal was to craft a vaccine that significantly reduces *C. difficile* colonization burden while protecting against mortality and weight loss. Clearance of the bacterium from the colon is an important prerequisite for preventing *C. difficile* transmission and recurrence, which occurs in up to 30% of patients (PMID: 38577028). During the NTA cocktail experiments, IP-vaccinated mice did not reduce colonization burden (**Fig. 1g**) like RI-vaccinated mice did (**Fig. 1f**). Furthermore, we were unable to achieve bacterial clearance in IP-vaccinated mice administered the toxin + NTA formula (**new Fig. 2n,q**), but were able to achieve clearance via RI (**new Fig. 2h,k**). By this metric, we have established the superiority of RI over conventional IP vaccination for *C. difficile* clearance. In addition, during the NTA cocktail experiments, IP vaccination prevented weight loss (**Fig. 1c-e**), but IP-vaccinated mice performed worse in terms of survival (**Fig. 1b**).

We further demonstrated the ability to achieve sterilizing immunity against *C. difficile* with RI, but not IP, vaccination by performing three new experiments. First, we vaccinated mice by RI or IP with the

same formulas (dmLT only controls, dmLT + toxins, dmLT + toxins + NTAs), challenged them with WT *C. difficile* R20291, and held mice for fifteen days to ensure there was no colonization “rebounding” and that our previous results (taken to ten days post-infection in the original Fig. 3) were replicable. In the new Fig. 2, we validated that the clearance phenotype holds for RI-vaccinated mice but not IP-vaccinated mice (**new Fig. 2f-q**).

Second, we harvested ceca and colons during necropsy (day fifteen post-infection; experimental endpoint) to isolate and culture *C. difficile* from the tissues, and we performed PCR and qPCR on extracted DNA from the organs. No *C. difficile* spores nor vegetative bacteria were isolated from macerated ceca and colons of RI mice vaccinated with the toxin and NTA formula (**new SFig. 7a-d**). However, *C. difficile* was cultured from the tissues of IP mice, and from some RI mice given the toxin-only formula. To further validate these findings, which implicate RI in achieving sterilizing immunity against *C. difficile*, we performed both PCR and TaqMan qPCR on DNA extracted from the same macerated tissues to detect a *C. difficile*-specific 16S rRNA gene sequence. This approach provided a more sensitive detection method compared to our limit of detection with CFU plating. By PCR, we qualitatively verified the lack of the 16S product (175 bp) in the colons and ceca of mice that had received the toxin and NTA formula by RI (**new SFig. 7e**). Distinct PCR product bands were visualized in mice that had been vaccinated by IP, as well as some mice vaccinated by RI with the toxin-only formula. To even further clarify the potential elimination of the pathogen from these mice, we performed qPCR, again probing for the *C. difficile*-specific 16S rRNA gene sequence in the colons and ceca of vaccinated and challenged animals. Notably, there were no significant differences in cycle threshold or normalized reporter values between RI-vaccinated ceca and colons and the negative control, which used nuclease-free water instead of template DNA (no template control) (**new SFig. 7f,g**). The normalized reporter values and cycle threshold in the colonic DNA isolated from RI-vaccinated mice with the toxin-only formula were also not significantly different from the negative control. However, dmLT-only vaccinated mice by either route of administration, as well as IP-vaccinated mice regardless of formula, had significantly lower cycle thresholds and higher normalized reporter values compared to the negative control (**new SFig. 7f,g**). Taken together, the organ plating, PCR, and qPCR results indicate that RI vaccination with the toxin and NTA formula confers sterilizing immunity against *C. difficile* in mice, whereas IP vaccination does not.

Lastly, we performed a relapsing *C. difficile* infection in RI-vaccinated mice to ensure true clearance of the bacteria. We vaccinated mice by RI with various experimental formulas (dmLT only controls, dmLT + toxins, dmLT + toxins + NTAs) before treatment with vancomycin. In this model, mice taken off vancomycin rebound with infection, as shown previously (PMID: 21576341; PMID: 23147742) (**new Fig. 3a**). We demonstrated that RI of the complete formulation (dmLT + toxins + NTAs) cleared *C. difficile* infection during vancomycin treatment, and that mice did not develop CDI recurrence (**new Fig. 3b-d,i,j**). Mice administered with the toxin-only and toxin and NTA formulas were significantly protected against death (**new Fig. 3b**), weight loss (**new Fig. 3c**), and diarrhea (**new Fig. 3d**) during reinfection. That said, animals that were immunized against the inactivated toxins only were still susceptible to relapse and were only protected against early effects of recurrence; these animals all succumbed by day 17 post-infection (five days after being taken off vancomycin treatment). Not only were the animals given the toxin and NTA formula protected against disease severity and death, but they also cleared primary infection rapidly while on vancomycin and never relapsed in-terms of total CFU burden and spore counts (**new Fig. 3i,j**). The prevention of recurrence is even more striking when compared to the bacterial burdens of the RI controls (**new Fig. 3e-h**), which rebloom one day after being taken off vancomycin (day thirteen post-infection). Combined, these data suggest that the toxin and NTA formula, administered by RI, can confer sterilizing immunity in the host, clearing *C. difficile* and any potential hidden spore reservoirs. We believe we have highlighted that RI is the most effective route of administration, providing sterilizing immunity, clearing the recalcitrant pathogen, and protecting against morbidity, mortality, and relapse.

Specifically,

1. Figures 1 and 2 show that IP immunization with non toxin antigens protects mice and impacts colonization to a degree similar to RI. Given that serum antibodies can transudate into the gut lumen, the authors should consider passive transfer of sera from immunized mice into naïve recipients to clarify the relative contributions of local mucosal versus systemic antibody responses.

Author's Response:

We thank the Referee for the suggestion of a passive transfer experiment. We agree that clarification of the relative contributions of local versus systemic antibody responses is key to this manuscript, as did Referee #3. Since serum IgG can transudate into the gut lumen, one would anticipate that IP-vaccinated mice, which we found have higher titers of antitoxin and anti-NTA serum IgG than their RI-vaccinated counterparts (**Fig. 4a-e**), would also clear the bacterium. However, we did not observe clearance of *C. difficile* in IP-vaccinated mice (**new Fig. 2n,q**). As such, we believe that mucosal immunization induces a specific, local immune response that cannot be achieved by serum IgG transudation.

To clarify the contributions of local and systemic antibody responses to the clearance phenotype observed by RI, we passively transferred fecal IgG against vegetative *C. difficile* by both RI and IP administration routes into mice naïve to those antigens. We chose to highlight the effects of fecal IgG, as opposed to serum IgG or fecal IgA, as this is what we have demonstrated (**Fig. 4k,n,o**) correlates with bacterial clearance in RI-vaccinated mice (**new Fig. 2h,k; SFig. 3**). Moreover, despite similarities between sera and fecal IgG, glycosylation of the Fc-region of IgG can differ based on where the antibody in question is produced (PMID: 25004930) and has profound impacts on effector functions (PMID: 36189205; PMID: 21653738; PMID: 22305040; PMID: 25895110; PMID: 26211613; PMID: 29309774). In our study, since fecal IgG correlates with *C. difficile* clearance and serum IgG does not, we passively-transferred fecal IgG by both routes of administration to assess the outcomes on CDI and colonization.

We isolated and sterile-filtered fecal IgG from donor mice RI-vaccinated with dmLT alone or dmLT + the vegetative NTAs (C40 peptidase 2 and FlgGEK). Donor fecal IgG were administered by either IP or RI throughout *C. difficile* challenge to recipient mice that were RI-vaccinated against the toxins, CspC, + dmLT (since CspC does not induce fecal IgG nor IgA by RI; **Fig. 4h,m; new Ext. Fig. 2a**). We did not discern differences in survival (**new Ext. Fig. 2c**) nor weight loss (**new Ext. Fig. 2d**) between groups administered dmLT or vegetative antigen fecal IgG by either route. However, recipient mice that had received fecal IgG via RI had significantly less diarrhea on days 4, 5, and 6 post-infection than those that had received control mucosal IgG (**new Ext. Fig. 2e**). These mice also had significantly decreased total *C. difficile* burden on days 2, 3, 4, and 8 post-infection when compared to their respective RI dmLT-only controls, coinciding with administration of fecal IgG (**new Ext. Fig. 2f,g**). None of the groups had statistical differences in *C. difficile* spore counts, suggesting that any notable differences in total CFU burden are due to vegetative bacteria (**new Ext. Fig. 2h,i,l,m**). Differences in total CFU burden were not present in mice that had been IP-injected with either group's fecal IgG (**new Ext. Fig. 2j,k**). These results demonstrate that anti-vegetative fecal IgG administered by IP did not significantly reduce colonization burden, suggesting that the contribution of systemic circulating IgG to *C. difficile* clearance is minimal. Conversely, RI-administered anti-C40 peptidase 2 and -FlgGEK fecal IgG specifically reduces vegetative *C. difficile*, highlighting fecal IgG as a driving force behind vegetative bacterial clearance.

2. In Figure 2, differences in CFUs between IP and RI groups are difficult to appreciate. The authors should include CFU counts at a later timepoint (e.g., day 10 post challenge) with statistical comparisons to demonstrate any advantage of RI in reducing bacterial burden.

Author's Response:

We thank the Referee for this suggestion. We attempted to address this suggestion in **Figure R1** with day 10 post-infection colonization data from the new **Ext. Fig. 1d-m** (previously Figure 2d-m). While there is no statistical significance between nor within groups at this timepoint, RI-vaccinated mice that have received the NTAs trend towards lower CFU/g feces compared to IP-vaccinated mice. This further highlights the few RI-vaccinated mice in the CspC, C40 peptidase 2, and FlgGEK groups that have cleared *C. difficile* to the limit of detection. Our goal was to determine which NTAs contributed to the clearance phenotype in the mice RI-vaccinated with the NTA cocktail in Figure 1 with the hope of selecting only colonization-burden-reducing NTAs for inclusion in the complete formula with the toxins and dmLT in the new Figure 2. Extended Figure 1d-m emphasizes colonization burden trends in individual mice, as well as those that cleared infection.

3. Figure 5 omits the IP immunization arm. To assess whether RI offers longer lasting protection, the authors should compare the durability of immunity (e.g., survival, weight change, CFUs) following both RI and IP routes over the extended period following immunization.

Author's Response:

The longevity of parenteral *C. difficile* vaccines has been well documented (PMID: 39361752, PMID: 39180325, PMID: 32946836). Our goal for Figure 5 was therefore not to compare RI versus IP but rather to determine if the protective and clearance phenotypes of RI-vaccination held at elongated timepoints. We have shown that the protective and clearance phenotypes hold at 60- and 200- days post-boost (**new Fig. 5k,n,q,t**). Since IP-vaccinated mice did not reduce infection burden at two-weeks post-boost (**new Fig. 2n,q**), which was our primary endpoint in the creation of a preclinical immunization, we decided against testing its durability, as we anticipated that there would be no clearance phenotype.

Figure R1. RI of CspC, C40 peptidase 2, and FlgGEK reduce *C. difficile* colonization burden at Day 10 post-infection. Enumeration of *C. difficile* R20291 $\Delta\Delta\Delta B$ bacteria in feces. Limit of detection shown by dotted line, 500 CFU/g feces. Statistical significance was assessed using a one-way ANOVA with Tukey's post-hoc test; however, none of the comparisons were significant. Individual data points are represented and are pooled from two independent experiments. CFU, colony forming units. PD, polysaccharide deacetylase; C40 pep 2, C40 peptidase 2.

Referee #3:

Key results:

*In this manuscript, the authors have compared mucosal and parental immunization of mice with a novel vaccine containing cleverly designed antigens to clear and prevent infections with *C. diff*. Their strategy targeted several key elements in vaccine design and they measured several useful parameters as correlates of immunity including pathogen clearance. That said, some of the supplemental data are rather descriptive and not necessarily informative. The experiments reflected impressive expertise in some areas but maybe not in others – or the limitations of space may have precluded them from describing issues I sought to understand. Perhaps the focus should be on the impact of antigen composition and route of immunization rather than including a superficial description of possible effector mechanisms.*

Author's Response:

We thank the Referee for their overview and insightful suggestions. We have provided specific responses to each concern below; briefly, we have experimentally addressed key concerns about the contributions of fecal IgG to overall bacterial clearance and interpretation of flow cytometric analyses, and we added necessary wording and citations for methodology. We believe suggestions from Referee #3 significantly improved our manuscript.

Validity:

Experiments were repeated only twice and some potentially interesting results were not validated. For flow cytometry, I didn't see any mention of isotype controls or, preferably, FMO.

Author's Response:

While some experiments were only repeated twice, we were intentional in the effort to validate all of the primary results. In some cases, we did not properly document these validation efforts, so we appreciate the Referee bringing these issues to our attention. For example, we have amended the “Flow Cytometric Analysis of B and T cells” section in the Methods with the following language: “Prior to flow cytometric analysis, accurate gating of cell populations was determined using Fluorescence Minus One (FMO) controls. FMO controls were prepared by staining replicate cellular samples and beads (ThermoFisher, U20250) with all fluorophore-conjugated antibodies in the panel with the exception of one to be analyzed for a given control. This accounted for fluorescent spillover and spread across channels and allowed for the precise determination of positive and negative populations. Gating strategy is provided in SFig. 13.”

Some of the methods for antigen prep seemed scant and/or lacked sufficient citations.

Author's Response:

We appreciate the Referee's comment and apologize for the scant “Protein Expression and Purification” section in the Methods. Due to word constraints, we did not elaborate on the manufacturing of point mutations in WT VPI TcdA and WT R20291 TcdB2 backgrounds, but we now cite studies performed by our laboratory and others describing in-detail how we generate mutants (PMID: 27456833; PMID: 24567384; PMID: 34145250; PMID: 35303428). Primer information is provided in **STable II**. We also reference publications for the protein expression for WT VPI TcdA, WT R20291 TcdB2, and CspC, as they have been described in-detail elsewhere, including by our group (PMID: 27456833; PMID: 31276487). The plasmid information for each antigen can be found in **STable II** as well. We elaborate on the purification of the NTAs, as these had not been previously reported.

While IgG may be associated with clearance, no intervention strategy was employed to show cause and effect directly.

Author's Response:

We thank the Referee for this comment. We agree that clarification of the relative contributions of fecal IgG is key to this manuscript, as did Referee #2. To clarify the contributions of local and systemic antibody responses to the clearance phenotype observed by RI, we passively transferred fecal IgG against vegetative *C. difficile* by both RI and IP administration routes into mice naïve to those antigens. We chose to highlight the effects of fecal IgG, as opposed to serum IgG or fecal IgA, as this is what we

have demonstrated (**Fig. 4k,n,o**) correlates with bacterial clearance in RI-vaccinated mice (**Fig. 3j,m**). Moreover, despite similarities between sera and fecal IgG, glycosylation of the Fc-region of IgG can differ based on where the antibody in question is produced (PMID: 25004930) and has profound impacts on effector functions (PMID: 36189205; PMID: 21653738; PMID: 22305040; PMID: 25895110; PMID: 26211613; PMID: 29309774). In our study, since fecal IgG correlates with *C. difficile* clearance and serum IgG does not, we passively transferred fecal IgG by both routes of administration to define the impact on disease outcomes and pathogen clearance.

As such, we isolated and sterile-filtered fecal IgG from donor mice RI-vaccinated with dmLT alone or dmLT + the vegetative NTAs (C40 peptidase 2 and FlgGEK). We hypothesized that these anti-C40 peptidase 2 and anti-FlgGEK fecal IgG would clear vegetative *C. difficile*. Donor fecal IgG was administered by either IP or RI throughout *C. difficile* challenge to recipient mice RI-vaccinated against the toxins, CspC, + dmLT (since CspC does not induce fecal IgG nor IgA by RI, **Fig. 4h,m**) (**new Ext. Fig. 2a**). We did not discern differences in survival (**new Ext. Fig. 2c**) nor weight loss (**new Ext. Fig. 2d**) between groups administered dmLT or vegetative antigen mucosal IgG by either route. However, recipient mice that had received mucosal IgG via RI had significantly less diarrhea on days 4, 5, and 6 post-infection than those that had received control mucosal IgG (**new Ext. Fig. 2e**). These mice also had significantly decreased total *C. difficile* burden on days 2, 3, 4, and 8 post-infection when compared to their respective RI dmLT-only controls, coinciding with administration of fecal IgG (**new Ext. Fig. 2f,g**). None of the groups had statistical differences in *C. difficile* spore counts, suggesting that any notable differences in total CFU burden are due to vegetative bacteria (**new Ext. Fig. 2h,i,l,m**). Differences in total CFU burden were not present in mice that had been IP-injected with either groups' fecal IgG (**new Ext. Fig. 2j,k**). These results may suggest that anti-vegetative IgG administered by IP did not transudate into the colon from the serum. Still, they indicated that IP-delivered fecal IgG does not reduce the colonization burden. Conversely, RI-administered anti-C40 peptidase 2 and -FlgGEK fecal IgG specifically reduces vegetative *C. difficile*, highlighting fecal IgG as a driving force behind vegetative bacterial clearance. We further confirmed the ability of anti-C40 peptidase 2 and -FlgGEK fecal IgG to disrupt bacterial swimming motility (**new Ext. Fig. 2b**), providing evidence of direct interaction with *C. difficile*.

For the histopathology, two scorers should be used to validate findings and ideally, one is a veterinary pathologist or they should refer to a previously published paper that describes their scoring criteria and validation.

Author's Response:

We have amended the "Histopathology" section of the methods to include the following references (PMID: 22198617; PMID: 37699522). PMID: 22198617 is the original paper describing the scoring criteria we used and is standard in the *C. difficile* field. PMID: 37699522 is a 2023 *Nature* publication on *C. difficile* that describes the same scoring criteria and validates each metric (for example, what constitutes a "1" versus a "3" in edema, alongside images of these differences) in their Extended Figure #1a-e. Additionally, we enlisted a board-certified veterinary pathologist, Dr. Katherine Gibson-Corley, to blindly score the histopathology slides using the same criteria as our gastrointestinal pathologist, Dr. M. Kay Washington. We averaged scores from Drs. Gibson-Corley and Washington and have reported those new scores (**new SFig. 6a,b**). We have also amended the Methods section to reflect this change: "To assess histopathology, cecum and colon sections were stained with hematoxylin & eosin (H&E; Vector Labs), and conditions were masked for both a board-certified gastrointestinal pathologist and veterinary pathologist to separately score edema, inflammation, and epithelial damage based on published criteria ($n = 5$ per treatment)^{52,53}. Averages of the scores from the two pathologists were reported."

In their images, e.g. SFig 8, the colonic muscularis varies from 10 to 80 microns. That is unusual and may be due to mounting artifacts.

Author's Response:

While we acknowledge the possibility of mounting artifacts, we think it unlikely since the tissues were mounted by our translational pathology core and evaluated by professional pathologists. However, since we don't see the thickening throughout the tissue, and muscularis size does not factor into the histopathological scoring criteria (PMID: 22198617; PMID: 37699522), we have selected new images from the same tissue (**new SFig. 6**) that allow the viewer to better focus on the relevant histopathological differences (**new SFig. 6c**).

For IEL and LPL isolation, they should refer to methods reported previously that validated the purity of IEL vs LPL as cross contamination is likely, e.g. enterocytes with IEL; B cells mixed into the IEL, etc. For example, due to cells lost during isolation, flow cytometry cannot measure "total" cells, it only assesses percentages. That would require validation with IHC/IF. Further, IHC/IF would also validate the percentage B cells in IEL which appear higher than expected. If they haven't done it before, the validation could be mentioned here. That would enhance the strength of conclusions for data in figure 4 for example.

Author's Response:

We have now updated our "Methods" section under the "Preparations of Colons to Obtain Intraepithelial and Lamina Propria Lymphocytes" header to include the published reports that validate the purity of the IEL and LPL cells isolated by this protocol. (PMID: 23334789; PMID: 28783695). However, as a way to streamline the manuscript and reduce descriptive content, we have removed discussion of total cellular counts and have removed most of these graphs from the supplemental figures.

There is no evidence that the apparent differences in IEL or LPL have any impact on immunity induced by the vaccine. The paper could likely be streamlined by paring the descriptive data unless it at least relates to a question many people might be interested in learning about.

Author's Response:

We thank the Referee for the suggestion of how to better streamline the manuscript. In response, we have removed much of the descriptive data regarding cellular counts and splenic B cell responses. However, we have retained the report of the CD4⁺ and CD8⁺ T_{RM} cells in the IEL and LPL compartments at 2 weeks post-boost (**Fig. 4p-s**), because our new longevity data now show the retention of increased CD8⁺ and CD4⁺ T_{RM} (**Fig. 5g,h**) in the LPL of vaccinated mice. These differences can be attributed to the rapid turnover rate of IELs, as well as longevity of LPL T_{RM}, in the gut (PMID: 20156972; PMID: 31337737). Given our prior observation that CspC alone induces a Th17 skew in the T_{RM} response (**Fig. 4t-v; SFig. 10**), and our new data showing that CspC-induced responses are necessary to protect against death and weight loss (**new Ext. Fig. 3a-c**) and reduce colonization burden (**new Ext. Fig. 3e,g**), we think these results will be of exceptional interest. As the first pre-clinical vaccine to fully eliminate *C. difficile* from the host, having positive correlates of spore clearance will advance the next stages of optimization.

Originality and Significance:

The approach to develop the antigen appeared to be very original and well thought out. Such an approach could have real impact on this disease. The local immunization was also helpful although I remain skeptical of how many subjects would want a rectal immunization.

Author's Response:

We thank the Referee for their generous words and understand their hesitations with intrarectal instillation. That said, surveys of healthy adults and *C. difficile* infected patients have shown promise for positive public reception of an enema-based immunization. In a survey of 1216 adults, 56% said they would "opt-in" for *C. difficile* vaccine over "opt-out", with 75% stating they would receive a *C. difficile* vaccine if the out-of-pocket cost were \$0. Those surveyed indicated high relative importance (RI) regarding vaccine efficacy (RI = 17.7) and disease severity (RI = 10.3), compared to injection site (RI = 6.4) and dosing (RI = 2.2) (PMID: 39217776). This survey highlights the willingness of the public to receive a *C. difficile* vaccine so long as it is effective, regardless of route of administration. Although the enema route of immunization seems taboo, so too did the utilization of fecal microbiota transplantation (FMT) via enema to treat *C. difficile* infection before FDA approval of Vowst and Rebyota. In a 2020

survey of patients with *C. difficile* infection, only 31% of patients reported reservations about receiving FMT either by colonoscopy or enema, citing “yuck factor” as their greatest reservation. Despite this, all patients still opted for FMT and 96% recommended it as a treatment for *C. difficile* infection (PMID: 33102769). These sentiments were also documented by other groups, especially with patients aware of the physical and psychological burden of *C. difficile* infection and recurrence (PMID: 37395984). While FMT as a treatment and a mucosal vaccine as a preventative measure are very different, we believe that potential aesthetic reservations about receiving an enema could be addressed with education and marketing.

Data and Methodology Including Statistics:

The experimental design was marginal. For example, it appears only 6-week-old mice were used which does not determine any age effects as required by NIH guidelines. It is unclear if multiple experiments were done to assess inter-experimental variation in all figures. This was mentioned in some figure legends but not all.

Author’s Response:

We appreciate the Referee’s comments and apologize for the lack of clarity. We did perform multiple experiments to assess inter-experimental variation. In particular, we have performed multiple iterations of the same experiments for the most important figures of the manuscript but did not for some of our initial exploratory experiments. We have clarified this in the figure legends. We have addressed the Referee’s comments regarding age effects below (data amended onto **new Fig. 2**), underneath the Referee’s “Validating Key Findings” sub-header.

In some cases, N was 5-15 but it is unclear why they used such a large range or if some responders were deleted from the study. They offer that the variation was depending on the ability to collect samples but no further explanation was evident.

Author’s Response:

We apologize to the Referee for our lack of clarity. The range of n in experiments varied depending on when the samples were taken (before, during, or after vaccination and challenge) and whether the animals could provide samples. For example, at 2-3 days post-infection with *C. difficile* R20291, animals rarely produce enough stool to quantify CFU due to severe diarrhea, resulting in lower n in experiments requiring fecal samples. Many animals, especially dmLT-only and naïve controls, succumb to infection at 2-3 days post-infection and are found dead, preventing us from obtaining samples for flow cytometry or fecal plating. Similarly, it is difficult to obtain sera and fecal samples from some animals before challenge. To prevent confusion, we amended the “Statistical Analysis” section of the Methods to more clearly state: “The range of n within experiments varies based on when samples are taken; for example, lower n on days 2-3 post-infection may be due to animals succumbing to disease or sickness and the inability to provide fecal samples.”

As best I could tell, statistics were adequately described but it’s hard to determine if they are appropriate if there is no mention of the normal distribution of the variables. The selection of antigens seems to have been very well thought out.

Author’s Response:

We thank the Referee for this comment and apologize for our lack of transparency; this was not intentional. We added the following language to our “Statistical Analysis” sub-header in the “Methods” section of the manuscript: “Quantitative variables were tested for normal distribution via D’Agostino-Pearson normality tests. If normality was not indicated, then non-parametric statistical tests were used. Statistical tests, parametric or non-parametric, are listed for each experiment in the figure legends and underneath the corresponding Methods sub-header. Sample variances were also similar between groups unless otherwise mentioned”.

For the rectal immunization, no details are provided on the colon prep.

Author’s Response:

We thank the Referee for this note and apologize for the lack of clarity. We have added text to describe colon prep to our Methods section under “Animals and Study Design”. Specifically, we now state: “Mice were rectally instilled after fecal collection to empty the colon. Rectal instillation occurred under anesthesia using a sterilized metal ball-end gavage needle that was inserted into the rectum. The vaccine formula was pulsed into the colon, and the rectum was manually squeezed shut for 15 seconds after administration to prevent leakage, as described previously³⁹.”

It wasn't clear why they chose rectal immunization vs oral. Rectal immunization would be a challenging approach to sell in human medicine. That said, adding it to a routine colonoscopy would be logical although that was not the administrative route they tested.

Author's Response:

We thank the Referee for the question. We chose to administer our vaccine rectally as opposed to orally for a few reasons. First, previous oral preclinical vaccines against *C. difficile*, using nontoxicogenic or manufactured strains, protected animals against death. However, these oral vaccines did not induce high titers of anti-toxin nor anti-surface protein fecal IgA and serum IgG, mice were challenged with less clinically relevant and hypervirulent strains, and colonization burden either did not decrease or was not tested (PMID: 30150259, PMID: 35583336). These results gave us pause as to whether oral administration would elicit robust humoral and cellular responses necessary to clear infection. Secondly, oral immunization with the enteropathogenic *Escherichia coli* (ETEC) ACE527 vaccine with the adjuvant dmLT reduced ETEC strain shedding by 10-fold in human participants (PMID: 30797634), but animals remained notably colonized post-vaccination. This may be because researchers did not immunize the intestinal mucosa, where ETEC infects, but the upper gastrointestinal tract. Preclinical nasal vaccines against influenza (PMID: 34344825), *Bordetella pertussis* (PMID: 37275891), and SARS-CoV-2 (PMID: 36302057) all significantly reduced colonization burden or cleared infection outright, likely due to the enhanced local mucosal immune response in the respiratory tract and lungs, where these infections occur. There also remain questions about whether oral vaccination, even after providing subjects with sodium bicarbonate slurries to neutralize stomach acid, would alter antigen conformations and thus neutralizing epitopes. Given these data, we believed immunizing the local mucosa would be the best route of administration to prevent the adverse effects of stomach acid/bile acids on the conformation of the antigens, as well as initiate more robust *C. difficile* clearance at the site of infection.

As we noted to the Referee in a previous response above, we understand the hesitations around intrarectal instillation but believe education and marketing would reduce any taboo surrounding an enema-based *C. difficile* vaccine. Previously, our concluding paragraph in the manuscript mentioned adding an immunization to a routine colonoscopy. We apologize to the Referee for the confusion, as we did not perform colonoscopies but enemas. We have since adjusted our language to reflect this, stating: “We envision deploying an effective RI vaccine as an enema, similar to the original route of administration for fecal microbiota transplantation therapeutics. This strategy could be supplemented with a parenteral prime or boost, an approach that has shown promise in other preclinical vaccine studies²³. A recent survey highlighted the willingness of the public to receive a *C. difficile* vaccine so long as it is effective, regardless of administration route, which supports the potential for an enema-based vaccine³⁷.” We believe marketing an enema-based vaccine after a colonoscopy would reduce the “yuck factor” as previously mentioned (PMID: 33102769), as one would not travel to the hospital just for the vaccine but for a routine procedure, and would be preferable, as patients would be sedated under anesthesia and far more comfortable.

Figure 3C didn't seem to have all three cohorts represented in the figure. There is a symbol that might be suggesting the lines overlap but it should be more clearly specified.

Author's Response:

We thank the Referee for pointing this out. We have updated the language of the original Figure 3 (**new Fig. 2**) legend to reflect that the cohort lines overlap: “**(B to C)** Survival of WT *C. difficile* R20291

infection. Bracket on C refers to both the lines of both the toxins + dmLT and toxins + NTAs + dmLT cohorts.”

Conclusions:

The authors conclude that mucosal immunization is a promising strategy for the prevention of symptoms and clearance of C diff. Their core data support this through the use of an innovative vaccine.

Author’s Response:

We thank the Referee for agreeing that our work is innovative and that our data support our core desire to design a preclinical vaccine against *C. difficile* that clears infection while protecting against mortality and morbidity.

Suggested Improvements:

Most are listed above but focusing the manuscript on proven elements, e.g. the functional vaccine and its route of administration vs. vague and indirect correlates of immunity. 23 figures with sometimes over 30 panels reflects today’s trend but cannot be meaningfully described in such brief legends and text making them quite superfluous.

Author’s Response:

We thank the Referee for this comment and, in response, have condensed and removed unnecessary supplemental figures; it was not our intention to be superfluous. However, given the potential impact of our findings, and the importance of both transparency and making research data publicly accessible, we did retain a lot of multi-paneled figures. We have streamlined the narrative to focus on the most important data and are open to working with *Nature* editors to potentially further reduce figure count if requested.

Validating Key Findings:

Expanding the age cohorts. Adding 8 12-16 week old mice per group would give another experimental replicate and address variations by age that can be tracked by ANOVA.

Author’s Response:

We thank the Referee for this suggestion and expanded our original Figure 3 (**new Fig. 2a**) to include a cohort of mice that underwent vaccination starting at 12-weeks of age. We did not observe any age-related differences by ANOVA and post-hoc tests, so we combined these experimental replicates with our previous data (an additional n=10 per experimental formula and route of administration; now combined in new Figure 2). The aged mice were monitored for fifteen days post-infection to further probe potential reblooming effects, as suggested by Referee #1. No differences were discerned in survival (**new Fig. 2b,c**) or weight loss (**new Fig. 2d,e**) between the previous and new experiments. Similar patterns of bacterial burden were also noted (**new Fig. 2f-q**). For mice RI-vaccinated with the toxin + NTA formula, there were no rebounding effects of total and spore CFU burden past 9 and 8 days, respectively (**new Fig. 2h,k**). IP-vaccinated mice with the toxin + NTA formula remained colonized with *C. difficile* throughout the experiment (**new Fig. 2n,q**).

References:

The authors know the C diff field much better than I do so I would assume the citations are reasonable for the most part. Some methodological techniques could be cited more thoroughly.

Author’s Response:

We appreciate the Referee’s comment and have addressed the lack of citations in the Methods section as was noted in their previous suggestions. We thank the Referee for directing us to where we lacked adequate citations.

Clarify and Context:

Minor points: “Filletted open” is not the correct way to describe whatever it is they were doing. I don’t understand if declaring no conflict of interest is consistent with two authors filing a patent based on the data.

Author’s Response:

We apologize for our confusing language and have adjusted it. In the “Preparations of Colons to Obtain Intraepithelial and Lamina Propria Lymphocytes” section of the Methods, we have altered the language from “filleted open” to “cut open longitudinally”. We have also omitted the language declaring no conflict of interest in the “Competing Interest Declaration” to reflect the Referee’s statements.

Response to Referees for Nature Submission 2025-06-14298A

We would like to thank the Referees and Editor for their comments and the opportunity to address them. In response, we have clarified points of concern and confusion in the Discussion of the revised manuscript, particularly in response to Referees #1 and #3. Each of the Referees' comments and concerns have been addressed in a point-by-point response below.

Moreover, we have introduced edits to the revision to fit the *Nature* guidelines on length. We have removed redundancies from the introduction of our Main Text and omitted our Abstract in place of a sole Summary Paragraph. To reduce the size and number of printed figures, we have reassigned previous Figure 5 as an Extended Figure (**new Ext. Fig. 4**). Additionally, we have included all Source Data and Supplementary Information for the gels in SFig. 7e.

We feel our revised manuscript is improved in terms of clarity and content thanks to the comments from the Referees, and we look forward to your continued consideration.

Referee #1:

The revised manuscript, “Mucosal vaccination clears Clostridioides difficile colonization” by Thomas et al. (2025-06-14298) describes a novel vaccine formulation and a comparison of the results of vaccination via parenteral (intraperitoneal injection) and mucosal (intrarectal installation) administration. I would like to thank the authors for the additional experimentation designed to address the concerns of the three reviewers who read the initial version of the manuscript. In my opinion, the authors have done a very good job in addressing all three reviewer’s comments. I will focus my comments here on the response to my comments on the first manuscript.

One of my major comments concerned the fact that rectal instillation (RI) of the complete vaccine (including the toxins, the non-toxin antigens and the spore antigen CspC) appeared to result in elimination of C. difficile from the gastrointestinal tract while intraperitoneal (IP) administration did not. In the initial version, this was judged by cultivation of feces from immunized animals. I had suggested that examination of tissues at necropsy would be a possible way to further demonstrate that RI does result in the generation of sterilizing immunity. The authors have done this work and using a combination of cultivation and molecular analysis (PCR) they are unable to detect C. difficile in the tissues of mice immunized by the RI route, while the pathogen was still detected in tissues of mice immunized by the IP route. While it is obviously difficult to “prove the negative” I think that the additional data provide robust evidence that RI does lead to elimination of C. difficile from immunized animals.

Further evidence of the efficacy and sustained protection afforded by RI administration of their novel vaccine is provided by new data that extends the initial period of observation of challenge from 10 to 15 days, demonstrating that there was no rebound of C. difficile with a longer period of observation. As an alternate approach to demonstrate the elimination of C. difficile in the animals immunized by the RI route, the authors conducted additional experiments whereby they treated animals with vancomycin during the initial C. difficile infection and then monitored for recurrence after the discontinuation of the vancomycin. Mice that received the toxin-only vaccine did exhibit recurrence (and mortality) after stopping vancomycin while the animals that received the vaccine containing toxin and non-toxin/CspC antigens did not have a reappearance of C. difficile in their feces and did not exhibit any clinical signs of CDI. These additional data greatly strengthen the conclusion that the novel vaccine coupled with mucosal administration represents an advance in the development of a viable C. difficile vaccine.

Finally, the authors conducted additional experiments to examine the contributions of the non-toxin antigens FlgG_{EC}, C40 and the spore antigen CspC to the observed complete clearance of C. difficile compared to the toxin-only formulation. The results are complex, but the authors do provide evidence that provides insight into the relative contributions of the various non-toxin antigens in their vaccine to the elimination of vegetative cells and spores. Furthermore, they do demonstrate that CspC provides a significant amount of protection against serious/fatal infection in animals.

Overall, I think that the authors have done an excellent job in addressing the previous questions and concerns from the initial review.

Author’s Response:

We thank the Referee for their helpful feedback.

I do have clarifying questions concerning the versions of inactivated TcdB in the vaccine formulation. The authors note that the residual toxin activity was due to active TcdB2 toxin contaminating the purification column. Was this due to wild type (i.e. no mutations whatsoever) TcdB2? Furthermore, are there still results being reported from using the TcdB2 that contains the L1598A mutation? In lines 149-158 of the revised manuscript, the L1598A mutation is no longer mentioned. Were all of the experiments using this version of the toxin removed from the revised manuscript? I was unable to determine if this is true.

Author's Response:

We apologize for the confusion. Our previous description of a TcdB2 L1598A mutation was erroneous. All experiments were conducted with a mixture of TcdB2_{GTX,L1106K} and TcdB2_{GTX,L1106K,D1812G}.

Thank you for giving me the opportunity to revisit this interesting manuscript.

Vincent B. Young

Referee #2:

The authors addressed the most important questions. The manuscript is suitable for publication.

Author's Response:

We appreciate the Referee's continued support for this manuscript and belief that it is suitable for publication in *Nature*. We thank the Referee for their guidance during initial review and resubmission.

Referee #3:

Summary

This manuscript is a resubmission. The authors have exhaustively addressed many comments from the three reviewers and editor. The focus of the title is about the clearance of C diff with mucosal immunization. Given their methods, perhaps they should they state rectal immunization in the title and elsewhere instead of what appears to be a conscious attempt to avoid it. I am still concerned that they limited their studies to one age group (young mice) to model a disease that more often affects the elderly. The correlates of immunity remain superficially studied. I would suggest that more focus would help. Compare an older cohort. Focus on IgA and IgG and forget about the T cell studies for now.

Author's Response:

We thank the Referee for this feedback.

- We did not consciously avoid a reference to our method of “rectal immunization”, as it is a critical aspect of our approach. While we were not able to create a title that both included these words and stayed within the 75 character limit, we have now made explicit reference to the rectal immunization approach in the Summary Paragraph.
- We agree that CDI is a disease that primarily affects the elderly and would like to call your attention to two experiments that we included in the last revision to address your previously stated concern. At your suggestion, we added experiments where we immunized 12-week old mice. We also added an experiment where we waited 200-days after the final boost before challenging the mice with *C. difficile*.
- We will clarify our thinking on the IgA and IgG responses below but have retained the T cell results with guidance from the Editor.

Originality and significance

I like the approach to design the antigenic composition of the vaccine and route of immunization. I find these original and significant.

Author's Response:

Thank you.

Data and Methods and Suggestions for Improvement

It is perplexing and I couldn't find a clear explanation, why IP immunization is so immunogenic yet ineffective, particularly since they make a case for IgG being important.

Author's Response:

We agree! It is certainly not the result we expected, and unfortunately, we do not have a satisfying explanation as to why the IP immunization was ineffective. Rather than focus on that question, our goal was to better understand the immune correlates of the rectal immunization that was effective.

I still think that a single age group ignores the important potential effects of age. 6 weeks is quite a young age for mice (sexually mature but not adult). In addition, they could still have residual passive immunity to who knows which antigens that could impact colonization, immune responses etc. In contrast, the disease they are trying to prevent most commonly affects the elderly who often have waning immune systems. I do not agree that this age is sufficiently

informative to guide future studies in humans. I believe that animal models should attempt to provide a parallel context to the human condition they are studying. This is exactly why the NIH requires all investigators to consider the impact of sex and age on the outcomes they measure in animal models. If this premise is important and correct, then one age group is inadequate, particularly if it is not the age group most affected by the disease one is modeling. It is possible that what they find in this age group has no bearing on most of the humans who get the disease. Perhaps a challenge of the mice 3, 6, 9 months after immunization would suffice. Perhaps immunizing mice at 6, 12, and 24 weeks of age would be better. As they performed the experiments, it is entirely possible that the benefit of this vaccine would not be reproduced in mature subjects receiving a colonoscopy at which time they could be immunized.

Author's Response:

We thank the Referee for this suggestion, which they had previously mentioned in their comments on our initial submission (received in July): “*Expanding the age cohorts. Adding 8 12-16 week old mice per group would give another experimental replicate and address variations by age that can be tracked by ANOVA.*” Please note that in our revised manuscript from October, we did include a cohort of mice that began vaccination at 12-weeks of age to Fig. 2. We did not observe any age-related differences by ANOVA and post-hoc tests, so we combined these experimental replicates with our previous data (an additional n=10 per experimental formula and route of administration; now combined in Figure 2). Moreover, we did challenge mice 60- and 200-days post-rectal-immunization (2 and ~7 months, respectively) and found no attrition in the ability of the vaccine to protect and clear mice (Ext. Fig. 4). Overall, we agree that testing in aged mice is important and hope to test even older mice in greater depth going forward. We have now added dialogue about this to the Discussion.

They note that rectal immunization was chosen as intranasal is effective for respiratory pathogens, perhaps due to its proximity to the site of infection. Part of the success of intranasal infection is the ability to induce local IgA. Although not tested directly, they feel that IgA did not seem to be a significant mediator of immunity in this study. I feel that rigorously showing that IgG is more important while IgA is not would be a detail that is worth investigating more thoroughly. Many of the responses are associations. While the transfer of IgG conferred some protection, that implies it is sufficient, not necessary. Further, how is it working? Is it secreted into the lumen where the infection is? If so how? Did they demonstrate luminal IgG that wasn't part of contamination with mucosal tissue IgG preps? If so is it protected from digestion? Do they have images of IgG bound to C. diff?

Author's Response:

We thank the Referee for these comments. We agree that clarification of the relative contributions of fecal IgG/IgA is key to this manuscript, as did Referee #2 in response to our initial submission. Part of what, we believe, makes our manuscript novel and of interest to broad readership is the fact that rectal instillation does not imbue the canonical humoral correlates of protection that other routes of mucosal administration do (PMID: 34312520). Significantly, this includes a lack of fecal IgA responses in our rectally-instilled mice.

Intraperitoneally-injected mice did induce an antitoxin and anti-CspC fecal IgA response (**Fig. 4f-h**), but were unable to clear infection (**Fig. 2m,n,p,q**). Intraperitoneal injection

also did not garner an anti-C40 peptidase 2 nor -FlgGEK fecal IgA titers (**Fig. 4i,j**), suggesting that there was minimal, if any, IgA response against the vegetative antigens. Conversely, the only fecal IgA response observed in rectally immunized mice was one against TcdA (**Fig. 4f**). However, mice administered with the inactivated toxin formula (including TcdA and TcdB, but without NTAs) were unable to clear infection (**Fig. 2g,j**), whereas those given the formula that included the NTAs did (**Fig. 2h,k**). Taking all these data into account, we made the logical connection that any fecal IgA garnered against TcdA and/or CspC did not contribute to *C. difficile* clearance. We have rewritten parts of our text to better clarify this reasoning.

Rather than focus on the shortcomings of intraperitoneal injection and why it did not clear *C. difficile* colonization (**Fig. 2m,n,p,q**) despite inducing a systemic serum IgG (**Fig. 4a-e**) and a mucosal fecal IgA (**Fig. 4f-h**) humoral response, we chose to highlight the correlates that *did* associate with sterilizing immunity – namely, those imbued by rectal instillation. The one humoral correlate of bacterial clearance that was starkly different between rectally-instilled and intraperitoneally-injected mice was that of fecal IgG. As opposed to serum IgG or fecal IgA, fecal IgG (**Fig. 4k,n,o**; **SFig. 2j,k**) correlates with bacterial clearance in rectally-vaccinated mice (**Fig. 2h,k**; **Fig. 3i,j**). Importantly, there is an expansion of anti-vegetative surface antigen fecal IgG at both 60- and 120-days post-RI boost (**SFig. 11n,o**), implying longevity of these responses. Anti-C40 peptidase 2 and -FlgGEK fecal IgG significantly reduced vegetative bacterial burden when passively transferred by rectal instillation (**Ext. Fig. 2g**), but not by parenteral injection (**Ext. Fig. 2k**), suggesting that any systemic anti-vegetative IgG do not transudate into the colon (PMID: 12965913; PMID: 12163559) and are instead produced locally. We therefore believe that anti-vegetative fecal IgG, as produced by local, colonic plasma cells and memory B cells in the mesenteric lymph nodes (which we have shown to be significantly increased in rectally instilled mice only, **SFig. 8o,p,r,s**) is secreted into the lumen during infection to clear the bacterium. Similar mechanisms have been demonstrated in other bacterial infection models (PMID: 25936799; PMID: 26944199; PMID: 30686564). We have now included dialogue in the Discussion that acknowledges the potential for protective fecal IgA responses moving forward.

We hypothesize that these fecal IgG could be agglutinating or enchaining the bacterium (PMID: 28405025), allowing for antibody-mediated phagocytosis (PMID: 36653078) or other effector functions (PMID: 29063907). We have shown that the binding of these fecal IgG influences the swimming motility of *C. difficile* (**Ext. Fig. 2b**), but a deeper understanding of the mechanisms that prevent motility are warranted for future studies.

Lastly, luminal IgG was not a contamination risk for our flow cytometric results as we did not look at mucosal tissue responses, only the mesenteric lymph nodes (**SFig. 8o,p,r,s**) and spleen (during initial review, this Referee suggested we omit splenic B cell data from our Supplemental Figures, as it complicated our manuscript and added unnecessary length; however, these data can be found in our initial submission). As such, there would not have been any luminal IgG from the colon present in those tissues.

Supplemental Fig 7 says many things. It appears IP immunization induced more IgG and IgA yet

*it wasn't as effective in protection as rectal immunization. I wasn't able to reconcile these observations and I apologize if I missed something. Why wouldn't they also purify and test mucosal IgA, at least as a foil? As reported, it is an incomplete/uncontrolled experiment based on the assumption that IP immunization induced IgA but they weren't protective therefore IgA is meaningless. IP induced IgA may not be protective for many reasons. Perhaps it's monomeric, or the specificity varies due to the site of induction and the variations in certain antigen specific B cell precursor frequencies. Some studies, e.g. <https://doi.org/10.1016/j.micinf.2016.05.001>, suggest fecal IgA is associated with reduced *C. diff*. "Dogma" suggests only 4 or 5% of the B cells in the large intestine produce IgG, thus, ignoring the IgA limits the impact, and the credibility, of the study. There are limited data on the specificity of these antibodies, their origin from serum vs. mucosal B cells. Maybe the IgG is key to reducing *C diff* colonization, but they didn't really explore this very rigorously. Please don't get me wrong, while I recognize the importance of IgA, it is also true that IgA deficiency isn't as detrimental as one might predict. I also like new theories, but I like them better when they are evidence-based.*

Author's Response:

We apologize to the Referee for our lack of clarity and have rectified this in the revised text. We chose to not purify and test mucosal IgA alongside IgG for Ext. Fig. 2 due to the reasons we listed in our response above: there was no correlation between fecal IgA from either route of administration, especially intraperitoneal injection, and colonization burden reduction; the goal of this manuscript is to exhibit the unique correlates of sterilizing immunity that we did identify, like fecal IgG, not investigate the shortcomings of a fecal IgA response that was not associated with clearance. Our initial objective for Ext. Fig. 2 was not to ask why mucosal IgA, which is not correlated with *C. difficile* clearance, does not reduce colonization burden; it was to determine whether the one humoral correlate of clearance we had, fecal IgG, was sufficient to clear *C. difficile*. For these reasons, we do not believe that our experiments are ill-controlled. Our data, as outlined in our response above, demonstrate that the anti-vegetative fecal IgG we observe post-rectal instillation are from the gut-draining mesenteric lymph nodes. We recognize that the percentage of IgG-producing mesenteric lymph node B cells is low; however, other studies have shown that these cells expand upon infection (PMID: 25601863) and fecal IgG clears bacterial pathogens (PMID: 25936799; PMID: 26944199; PMID: 30686564).

Similarly, their investigation of T cell function is purely associative. That said, their findings are quite interesting and merit further investigation.

Author's Response:

The fact that the findings are interesting and merit further investigation is our rationale for including them in the manuscript.

The differences in pathology appear subtle. Do they have any independent objective correlates that support agreement? Wt loss vs score? TNF less vs. score.

Author's Response:

While the histopathological differences are subtle, the scores do correlate with the weight loss (**Fig. 2d,e**) and survival data (**Fig. 2b,c**). Further, there is a well-documented history of antitoxin immune responses not reducing that damage *in vivo*. A recent preclinical multivalent *C. difficile* mRNA-lipid nanoparticle vaccine (PMID: 39361752) and anti-

TcdB monoclonal therapeutic (PMID: 41183070; PMID: 36045589) were not found to reduce histopathological damage. However, these antitoxin immune responses do protect the host from dying. As such, we chose to retain SFig. 6 in our revision to emphasize the capabilities of our mucosal vaccination model in reducing epithelial damage, even modestly. Our data, juxtaposed to previous studies, imply that an antitoxin humoral response is not sufficient to protect against toxin-induced epithelial damage, but that the other unique correlates of protection/clearance imbued by mucosal vaccination may be. We have now addressed this in the Discussion of our revised manuscript.

Statistics

*The figures are busy. 4 * isn't more impressive than 1. Define the level of significance and if that is 0.05, then indicate it with a single *. 0.00005 means nothing vs. 0.04 if you have defined significance as <0.05, particularly when N is sufficient, but small and different (2-10; 3. Etc). They show individual data points and while unclear, some visually appear to be nonparametric. If the response is robust, their replicate experiments should be included but it was unclear if they were in all cases.*

Author's Response:

We appreciate the Referee's perspective regarding the number of significance stars, but we feel that they provide visual interpretative insight into the raw data. We chose to retain how our significance is displayed in our figures, as it meets *Nature's* guidelines for statistics. Other articles published in *Nature* also delineate between one and multiple significance stars in their figures, and if not, individual *p*-values are listed (which we think would make our figures even busier). We are trying to balance full transparency with figures that are easy to understand. Similarly, we chose to show individual data points to highlight the natural variation that occurs with immune responses, *C. difficile* infections, etc. All replicate experiments have been included upon initial resubmission of the manuscript in October, and all nonparametric data have been noted in individual figure legends or the Methods.

Conclusions

Not all are justified. The mechanisms of protection are studied superficially and since the paper is broad, they are inadequately discussed critically.

Author's Response:

Our goal with this manuscript is to relay the critical and novel correlates associated with sterilizing immunity against *C. difficile* that are garnered upon administration of a mucosal vaccine. We believe we have investigated fecal IgG adequately to confirm its role in the clearance of vegetative *C. difficile*, which was suggested upon initial submission by both this Referee and Referee #2 (**Ext. Fig. 2**). More in-depth comments and responses to the Referee can be found in previous responses above. That said, we have rewritten and amended our Discussion to critically examine these correlates of protection and clearance and draw upon potential mechanisms that we plan to follow up on in the future.

References

Selective but OK.

Author's Response:

We credit this Referee's comments during initial resubmission that guided us to add necessary references to our Methods section to reinforce our work.

Clarity

Clearly written.

Author's Response:

We appreciate the Referee's comment that this manuscript is clearly written.

Response to Editor for Nature Submission 2025-06-14298B

We would like to thank the Editor for his clarification and comments, as well as for the opportunity to revise our manuscript for publication. In response, we have made significant adjustments to the format of our manuscript. Our revision includes the following improvements:

1. We have reformatted the Extended Data figures and Supplementary Information per *Nature* requirements and guidelines. We apologize for not correctly formatting these sections previously and appreciate the added clarification. Per the Editor's request, we have added another Figure to our original four, so the manuscript now contains a total of five main-text Figures (see new Figure 5). We also removed all pertinent data from our previous Supplementary Information document and have reformatted these data as Extended Data Figures. Some of these data/previous figures have been combined and re-formatted to meet the requirement of having only ten Extended Data Figures. We have since remade the Extended Data Figures in Adobe Illustrator to ensure continuity of formatting with our main-text Figures. We have provided all source data for the Extended Data Figures (and our entire manuscript), as well. All language and callbacks to the Extended Data Figures and Supplementary Information have since been corrected and spelled-out in the text. Our new Supplementary Information PDF contains only antigen quality control and safety data, source gels, and flow cytometric gating schemes, as *Nature* guidelines states.
2. All main-text Figures have been supplied as editable Adobe Illustrator files, and all text within them has been verified to be at least size 5. Sizing was also double-checked for Extended Data figures. If requested by the Editor, we can provide Adobe Illustrator files for the Extended Data Figures.
3. We have amended our Figure legends with the statistics asked. We have more specific notes on these changes in the Related Manuscript File entitled "Thomas Editorial Requests", which was sent to us by the Editor alongside our acceptance email.
4. We have since completed and/or updated copies of the following documents: Biology Editorial Checklist, Manuscript Checklist, Reporting Summary, and Third-Party Rights Table. These documents are listed as Related Manuscript Files in the *Nature* submission portal. Of note, we thank the Editor for drawing our attention to the Third-Party Rights Table and Licenses for subfigures created in BioRender. We have now supplied both as Related Manuscript Files. The BioRender subfigure licenses are in a Word document entitled "Thomas *et al.* BioRender Licenses".
5. We have fixed all in-text subheader lengths to be forty characters or less with spaces. These have since been changed in-text.
6. We have since amended our "Conflicts of Interest" statement to cover all authors.

We are grateful for the opportunity from the Editor to improve our manuscript for publication. We look forward to working together in the next editorial steps.